# More Exotic Field Theories in 3+1 Dimensions

Pranay Gorantla[1], Ho Tat Lam[1], Nathan Seiberg[2], and Shu-Heng Shao[2]

[1]Physics Department, Princeton University, Princeton, NJ, USA

[2]School of Natural Sciences, Institute for Advanced Study, Princeton NJ, USA

## Abstract

We continue the exploration of nonstandard continuum field theories related to fractons in $3 + 1$ dimensions. Our theories exhibit exotic global and gauge symmetries, defects with restricted mobility, and interesting dualities. Depending on the model, the defects are the probe limits of either fractonic particles, strings, or strips. One of our models is the continuum limit of the plaquette Ising lattice model, which features an important role in the construction of the X-cube model.

Friday 21st August, 2020

# 1 Introduction

Fractons are novel lattice models that do not admit conventional quantum field theory descriptions in the continuum limit. (For reviews, see e.g., [1, 2] and references therein.) Their main characteristics include immobile massive particle excitations, large ground state degeneracy that grows in the system size subextensively, and exotic global and gauge symmetries.

In this paper, we continue the exploration in [3–5] of exotic continuum field theory in $3 + 1$ dimensions and discuss new examples. As in [3–5], our theories are not Lorentz invariant. In fact, they are not even rotationally invariant. The spacetime symmetry of our field theories consists only of continuous translations and spatial rotations generated by 90 degree rotations. In $3 + 1$ dimensions, the latter finite rotation group is isomorphic to $S_4$. In addition, our models also have parity and time reversal symmetries.

Since many aspects of our discussion are similar to [3–5], we will be brief and we refer the reader to these papers for further details.

As in [6], we will investigate these nonstandard quantum field theories by following their exotic global symmetries. Similar to the theories in [3–5], discontinuous field configurations play an important role in our analysis of the charged spectra under these global symmetries.

We will then consider the $U(1)$ and the $\mathbb{Z}_N$ gauge theories associated with the exotic global symmetries. The $\mathbb{Z}_N$ gauge theories are gapped and have defects exhibit restricted mobility, i.e., fractons. Depending on the model, the fracton can be a point, a string, or a strip in space. The gapped $\mathbb{Z}_N$ theories in this paper are not robust (in the sense of [3]) if we do not impose any global symmetry. In the language of continuum field theory, it means that certain local operators can be added to the Lagrangian and destabilize the theory.

## 1.1 Outline and Summary

This paper has two parts. Part one of the paper, which consists of Sections 2, 3, and 4, studies the $3 + 1$-dimensional generalizations of the models in [3]. In Section 2 we study a $3 + 1$-dimensional XY-model with interactions around a cube. We will refer to this lattice model as the XY-cube model. Its continuum limit is described by a circle-valued scalar field $\varphi \sim \varphi + 2\pi$ with continuum Lagrangian [7],

$$\mathcal{L} = \frac{\mu_0}{2}(\partial_0\varphi)^2 - \frac{1}{2\mu}(\partial_x\partial_y\partial_z\varphi)^2 \ . \tag{1.1}$$

This $\varphi$-theory has two exotic global symmetries, which we refer to as momentum and winding. We analyze the charged spectra of these global symmetries. Similar to the ordinary relativistic compact boson in $1+1$ dimensions and to the $\phi$ theory of [3] in $2+1$ dimensions,

this theory also enjoys a self-duality. We compare these models in Table 1.

In Section 3, we study the $U(1)$ gauge theory associated with the momentum global symmetry in the $\varphi$-theory. This gauge theory has been previously studied in [7, 8]. The temporal and spatial components of the gauge fields $(B_0, B_{xyz})$ are in the $(\mathbf{1}, \mathbf{1}')$ representations of the spatial $S_4$ group. (See Appendix A for the representation theory of $S_4$ and our conventions.) Their gauge transformations are

$$B_0 \sim B_0 + \partial_0 \alpha, \qquad B_{xyz} \sim B_{xyz} + \partial_x \partial_y \partial_z \alpha, \tag{1.2}$$

with $\alpha \sim \alpha + 2\pi$. The gauge invariant field strength is

$$E_{xyz} = \partial_0 B_{xyz} - \partial_x \partial_y \partial_z B_0, \tag{1.3}$$

and there is no magnetic field. The Lorentzian Lagrangian is

$$\mathcal{L} = \frac{1}{g_e^2} E_{xyz}^2 + \frac{\theta}{2\pi} E_{xyz}, \tag{1.4}$$

with a $2\pi$-periodic $\theta$-angle. This theory has no propagating degrees of freedom and is similar to the ordinary $1 + 1$-dimensional $U(1)$ gauge theory and to the $2 + 1$-dimensional $U(1)$ gauge theory of $A$ in [4]. We compare these models in Table 2.

In Section 4, we Higgs this $U(1)$ gauge theory of $B$ to $\mathbb{Z}_N$. The $\mathbb{Z}_N$ theory admits a $BF$-type Lagrangian

$$\mathcal{L} = \frac{N}{2\pi} \varphi^{xyz} E_{xyz}, \tag{1.5}$$

where $\varphi^{xyz}$ is a circle-valued field in the $\mathbf{1}'$ of $S_4$. We also present two lattice models that lead to this continuum model at long distances. One of them is a $\mathbb{Z}_N$ lattice gauge theory and the other is the $\mathbb{Z}_N$ version of the XY-cube model in Section 2, which will be referred to as the cube Ising model. We compare the $\mathbb{Z}_N$ $B$-theory with the ordinary $1+1$-dimensional $\mathbb{Z}_N$ gauge theory and the $2 + 1$-dimensional $\mathbb{Z}_N$ gauge theory of $A$ in [3] in Table 3.

Both the $U(1)$ and the $\mathbb{Z}_N$ tensor gauge theories of $B$ have defects that are the probe limits of fracton excitations. These fractons exhibit restricted mobility: while a single fracton cannot move by itself, four of them, forming a fracton quadrupole, can move collectively. The $\mathbb{Z}_N$ theory also has large ground state degeneracy. When we regularize the $\mathbb{Z}_N$ tensor gauge theory on a lattice with $L^i$ sites in the $x^i$ direction, the theory has $N^{L^x L^y + L^y L^z + L^x L^z - L^x - L^y - L^z + 1}$ states of zero energy. The ground state degeneracy becomes infinite in the continuum limit. Similar to the ordinary $1+1$-dimensional $\mathbb{Z}_N$ gauge theory and the $2 + 1$-dimensional $\mathbb{Z}_N$ tensor gauge theory of [3], this $\mathbb{Z}_N$ tensor gauge theory is not robust if we do not impose any global symmetry.

Part two of the paper, which consists of Sections 5 - 8, investigates two $U(1)$ and $\mathbb{Z}_N$

| | $(1+1)d$ compact scalar | $(2+1)d$ $\phi$ theory | $(3+1)d$ $\varphi$ theory |
|---|---|---|---|
| lattice | XY-model | XY-plaquette model | XY-cube model |
| Lagrangian | $\frac{R^2}{4\pi}(\partial_0\Phi)^2 - \frac{R^2}{4\pi}(\partial_x\Phi)^2$ | $\frac{\mu_0}{2}(\partial_0\phi)^2 - \frac{1}{2\mu}(\partial_x\partial_y\phi)^2$ | $\frac{\mu_0}{2}(\partial_0\varphi)^2 - \frac{1}{2\mu}(\partial_x\partial_y\partial_z\varphi)^2$ |
| global symmetry | momentum $\partial_0 J_0 = \partial_x J^x$  $J_0 = \frac{R^2}{2\pi}\partial_0\Phi$ $J^x = \frac{R^2}{2\pi}\partial^x\Phi$  winding $\partial_0 J_0^x = \partial^x J$  $J_0^x = \frac{1}{2\pi}\partial^x\Phi$ $J = \frac{1}{2\pi}\partial_0\Phi$ | momentum dipole $\partial_0 J_0 = \partial_x\partial_y J^{xy}$  $J_0 = \mu_0\partial_0\phi$ $J^{xy} = -\frac{1}{\mu}\partial^x\partial^y\phi$  winding dipole $\partial_0 J_0^{xy} = \partial_x\partial_y J^{xy}$  $J_0^{xy} = \frac{1}{2\pi}\partial^x\partial^y\phi$ $J = \frac{1}{2\pi}\partial_0\phi$ | momentum quadruple $\partial_0 J_0 = \partial_x\partial_y\partial_z J^{xyz}$  $J_0 = \mu_0\partial_0\varphi$ $J^{xyz} = \frac{1}{\mu}\partial^x\partial^y\partial^z\varphi$  winding quadruple $\partial_0 J_0^{xyz} = \partial_x\partial_y\partial_z J$  $J_0^{xyz} = \frac{1}{2\pi}\partial^x\partial^y\partial^z\varphi$ $J = \frac{1}{2\pi}\partial_0\varphi$ |
| duality | T-duality $R \leftrightarrow 1/R$ | Self-duality $4\pi^2\mu_0 \leftrightarrow \mu$ | Self-duality $4\pi^2\mu_0 \leftrightarrow \mu$ |

Table 1: Analogy between the ordinary $1+1$-dimensional compact scalar theory, the $2+1$-dimensional $\phi$-theory of [3], and the $3+1$-dimensional $\varphi$ theory of Section 2.

|  | $(1+1)d\ U(1)$ gauge theory | $(2+1)d\ U(1)$ gauge theory of $A$ | $(3+1)d\ U(1)$ gauge theory of $B$ |
|---|---|---|---|
| gauge fields | $A_\mu \sim A_\mu + \partial_\mu \alpha$ | $A_0 \sim A_0 + \partial_0 \alpha$ <br> $A_{xy} \sim A_{xy} + \partial_x \partial_y \alpha$ | $B_0 \sim B_0 + \partial_0 \alpha$ <br> $B_{xyz} \sim B_{xyz} + \partial_x \partial_y \partial_z \alpha$ |
| field strengths | $E_x = \partial_0 A_x - \partial_x A_0$ | $E_{xy} = \partial_0 A_{xy} - \partial_x \partial_y A_0$ | $E_{xyz} = \partial_0 B_{xyz} - \partial_x \partial_y \partial_z B_0$ |
| Lagrangian | $\mathcal{L} = \frac{1}{g^2} E_x^2 + \frac{\theta}{2\pi} E_x$ | $\mathcal{L} = \frac{1}{g_e^2} E_{xy}^2 + \frac{\theta}{2\pi} E_{xy}$ | $\mathcal{L} = \frac{1}{g_e^2} E_{xyz}^2 + \frac{\theta}{2\pi} E_{xyz}$ |
| EoM | $\partial_0 E_x = 0$ | $\partial_0 E_{xy} = 0$ | $\partial_0 E_{xyz} = 0$ |
| Gauss law | $\partial_x E_x = 0$ | $\partial_x \partial_y E_{xy} = 0$ | $\partial_x \partial_y \partial_z E_{xyz} = 0$ |
| $U(1)$ global symmetry | electric one-form <br> $\partial_0 J_0^x = 0$ <br> $\partial_x J_0^x = 0$ <br><br> $J_0^x = \frac{2}{g^2} E_x + \frac{\theta}{2\pi}$ | electric tensor <br> $\partial_0 J_0^{xy} = 0$ <br> $\partial_x \partial_y J_0^{xy} = 0$ <br><br> $J_0^{xy} = \frac{2}{g_e^2} E_{xy} + \frac{\theta}{2\pi}$ | electric tensor <br> $\partial_0 J_0^{xyz} = 0$ <br> $\partial_x \partial_y \partial_z J_0^{xyz} = 0$ <br><br> $J_0^{xyz} = \frac{2}{g_e^2} E_{xyz} + \frac{\theta}{2\pi}$ |

Table 2: Analogy between the $1+1$-dimensional $U(1)$ gauge theory, the $2+1$-dimensional $U(1)$ gauge theory of $A$ in [3], and the $3+1$-dimensional $U(1)$ gauge theory of $B$ in Section 3.

|  | $(1+1)d\ \mathbb{Z}_N$ gauge theory | $(2+1)d\ \mathbb{Z}_N$ gauge theory of $A$ | $(3+1)d\ \mathbb{Z}_N$ gauge theory of $B$ |
|---|---|---|---|
| lattice | $\mathbb{Z}_N$ lattice gauge theory or $\mathbb{Z}_N$ Ising model | $\mathbb{Z}_N$ lattice gauge theory of $A$ or $\mathbb{Z}_N$ plaquettte Ising model | $\mathbb{Z}_N$ lattice gauge theory of $B$ or $\mathbb{Z}_N$ cube Ising model |
| fields | $B \sim B + 2\pi$ $A_\mu \sim A_\mu + \partial_\mu \alpha$ | $\phi^{xy} \sim \phi^{xy} + 2\pi$ $A_0 \sim A_0 + \partial_0 \alpha$ $A_{xy} \sim A_{xy} + \partial_x \partial_y \alpha$ | $\phi^{xyz} \sim \phi^{xyz} + 2\pi$ $B_0 \sim B_0 + \partial_0 \alpha$ $B_{xyz} \sim B_{xyz} + \partial_x \partial_y \partial_z \alpha$ |
| Lagrangian | $\mathcal{L} = \frac{N}{2\pi} B E_x$ | $\mathcal{L} = \frac{N}{2\pi} \phi^{xy} E_{xy}$ | $\mathcal{L} = \frac{N}{2\pi} \phi^{xyz} E_{xyz}$ |
| $\mathbb{Z}_N$ global symmetry | electric one-form $\exp[iB]$ ordinary zero-form $\exp[i \oint dx A_x]$ | electric tensor $\exp[i\phi^{xy}]$ dipole $\exp[i \int_{x_1}^{x_2} dx \oint dy A_{xy}]$ $\exp[i \oint dx \int_{y_1}^{y_2} dy A_{xy}]$ | electric tensor $\exp[i\phi^{xyz}]$ quadrupole $\exp[i \int_{x_1}^{x_2} dx \oint dy \oint dz B_{xyz}]$ $\exp[i \oint dx \int_{y_1}^{y_2} dy \oint dz B_{xyz}]$ $\exp[i \oint dx \oint dy \int_{z_1}^{z_2} dz B_{xyz}]$ |
| defect | probe particles $\exp[i \int_{\mathcal{C}} (dt A_0 + dx A_x)]$ | fractons $\exp[i \int_{-\infty}^{\infty} dt A_0]$ | fractons $\exp[i \int_{-\infty}^{\infty} dt B_0]$ |
| ground state degeneracy on a torus | $N$ | $N^{L^x + L^y - 1}$ | $N^{L^x L^y + L^y L^z + L^x L^z - L^x - L^y - L^z + 1}$ |

Table 3: Comparison between the ordinary $1+1$-dimensional $\mathbb{Z}_N$ gauge theory, the $2+1$-dimensional $\mathbb{Z}_N$ gauge theory of $A$ in [3], and the $3+1$-dimensional $\mathbb{Z}_N$ gauge theory of $B$ in Section 4.

| global symmetry | current conservation & differential condition | gauge fields |
|---|---|---|
| $(\mathbf{1}, \mathbf{3}')$ <br> dipole symmetry | $\partial_0 J_0 = \frac{1}{2}\partial_i\partial_j J^{ij}$ | $(A_0, A_{ij})$ of [4] |
| $(\mathbf{2}, \mathbf{3}')$ <br> tensor symmetry | $\partial_0 J_0^{[ij]k} = \partial^i J^{jk} - \partial^j J^{ik}$ | $(\hat{A}_0^{[ij]k}, \hat{A}^{ij})$ of [4] |
| $(\mathbf{3}', \mathbf{2})$ <br> tensor symmetry | $\partial_0 J_0^{ij} = \partial_k(J^{[ki]j} + J^{[kj]i})$ <br> $\partial_i\partial_j J_0^{ij} = 0$ | $(C_0^{ij}, C^{[ij]k})$ of Section 5 |
| $(\mathbf{3}', \mathbf{1})$ <br> dipole symmetry | $\partial_0 J_0^{ij} = \partial^i\partial^j J$ <br> $\partial^i J_0^{jk} = \partial^j J_0^{ik}$ | $(\hat{C}_0^{ij}, \hat{C})$ of Section 7 |

Table 4: The four exotic $U(1)$ global symmetries of [4] and their associated gauge fields. Each global symmetry is labeled by the $S_4$ representations $(\mathbf{R}_{\text{time}}, \mathbf{R}_{\text{space}})$ for the temporal and spatial components of its conserved current. The temporal components of the currents for the third and the fourth global symmetries obey a differential condition in addition to the current conservation equation.

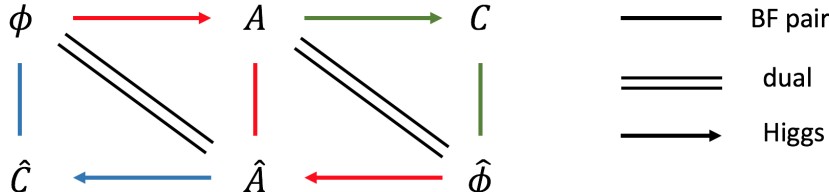

Figure 1: The relation between the two $U(1)$ gauge theories $A, \hat{A}$ of [4], the two non-gauge theories $\phi, \hat{\phi}$ of [4], and the two $U(1)$ gauge theories $C, \hat{C}$. The solid lines mean that there is a $\mathbb{Z}_N$ gauge theory whose $BF$ Lagrangian uses these two fields. The double lines mean that the two theories are dual to each other. The arrows, for example, $\phi \to A$, mean that the former is the Higgs field of the latter. Each solid line and arrow gives a gapped $\mathbb{Z}_N$ theory, with certain equivalences. We use the same color for the same $\mathbb{Z}_N$ gauge theory. In total, there are three different $\mathbb{Z}_N$ gauge theories. $\mathbb{Z}_N$ $A$ or $\hat{A}$-theory in [5]: $(\phi \to A) = (\hat{\phi} \to \hat{A}) = (A - \hat{A})$, $\mathbb{Z}_N$ $C$-theory of Section 6: $(A \to C) = (\hat{\phi} - C)$, $\mathbb{Z}_N$ $\hat{C}$-theory of Section 8: $(\hat{A} \to \hat{C}) = (\phi - \hat{C})$.

gauge theories related to the models studied in [4].

In Sections 5 and 7, we consider $U(1)$ gauge theories, denoted by $C$ and $\hat{C}$, associated with two of the exotic global symmetries in [4] (see Table 4).[1] The currents of these two exotic global symmetries obey a differential condition in addition to the current conservation equation. Consequently, the gauge parameters in the associated gauge theory have more components and have their own gauge transformations. Furthermore, the $U(1)$ $C$ and $\hat{C}$ theories have no propagating degrees of freedom. These two properties make the $C$ and $\hat{C}$ theories similar to the ordinary two-form gauge theory in $2 + 1$ dimensions. We summarize these two $U(1)$ gauge theories in Table 5 and their relations to the theories of [4] in Figure 1.

In Sections 6 and 8, we Higgs the $U(1)$ gauge group of the $C$ and $\hat{C}$ theories to $\mathbb{Z}_N$ using the gauge fields $A$ and $\hat{A}$ of [4], respectively. The two gapped $\mathbb{Z}_N$ gauge theories admit a $BF$-type Lagrangian by pairing up either with the $\hat{\phi}$ or the $\phi$ fields discussed in [4]. The $\mathbb{Z}_N$ gauge theory of $\hat{C}$ is the continuum limit of the $\mathbb{Z}_N$ plaquette Ising model featured in [9] (see [10] for a review and earlier references). Both gauge theories have fractonic defects, which are either strips or strings in space. The two $\mathbb{Z}_N$ gauge theories are however not robust if no symmetry is imposed. We summarize these two $\mathbb{Z}_N$ gauge theories in Table 6.

---

[1]The gauge theories associated with the other two exotic global symmetries of [4], which are denoted as $A$ and $\hat{A}$, were already discussed in that reference. For more detail of these global symmetries, see Table 3 and 4 of [4] and Table 1 of [5].

| | $(3+1)d\ U(1)\ C$-theory | $(3+1)d\ U(1)\ \hat{C}$-theory |
|---|---|---|
| gauge fields | $C_0^{ij} \sim C_0^{ij} + \partial_0 \alpha^{ij} - \partial^i \partial^j \alpha_0$ <br> $C^{[ij]k} \sim C^{[ij]k} - \partial^i \alpha^{jk} + \partial^j \alpha^{ik}$ | $\hat{C}_0^{ij} \sim \hat{C}_0^{ij} + \partial_0 \hat{\alpha}^{ij} - \partial_k \hat{\alpha}_0^{k(ij)}$ <br> $\hat{C} \sim \hat{C} + \frac{1}{2} \partial_i \partial_j \hat{\alpha}^{ij}$ |
| gauge transformations of gauge parameters | $\alpha_0 \sim \alpha_0 + \partial_0 \gamma$ <br> $\alpha^{ij} \sim \alpha^{ij} + \partial^i \partial^j \gamma$ | $\hat{\alpha}_0^{i(jk)} \sim \hat{\alpha}_0^{i(jk)} + \partial_0 \hat{\gamma}^{i(jk)}$ <br> $\hat{\alpha}^{ij} \sim \hat{\alpha}^{ij} + \partial_k \hat{\gamma}^{k(ij)}$ |
| field strengths | $E^{[ij]k} = \partial_0 C^{[ij]k} + \partial^i C_0^{jk} - \partial^j C_0^{ik}$ | $\hat{E} = \partial_0 \hat{C} - \frac{1}{2} \partial_i \partial_j \hat{C}_0^{ij}$ |
| Lagrangian | $\mathcal{L} = \frac{1}{2g_e^2} E_{[ij]k} E^{[ij]k}$ | $\mathcal{L} = \frac{1}{\hat{g}_e^2} \hat{E}^2 + \frac{\hat{\theta}}{2\pi} \hat{E}$ , |
| EoM | $\partial_0 E^{[ij]k} = 0$ | $\partial_0 \hat{E} = 0$ |
| Gauss law | $\partial_k E^{k(ij)} = 0$ | $\partial_i \partial_j \hat{E} = 0$ |
| $U(1)$ global symmetry | $\partial_0 J_0^{k(ij)} = 0, \ \ \partial_k J_0^{k(ij)} = 0$ <br> $J_0^{k(ij)} = \frac{2}{g_e^2} E^{k(ij)}$ | $\partial_0 J_0 = 0, \ \ \partial_i \partial_j J_0 = 0$ <br> $J_0 = \frac{2}{\hat{g}_e^2} \hat{E} + \frac{\hat{\theta}}{2\pi}$ |

Table 5: The $U(1)$ gauge theories of $C$ in Section 5 and of $\hat{C}$ in Section 7.

|  | $(3+1)d$ $\mathbb{Z}_N$ $C$-theory | $(3+1)d$ $\mathbb{Z}_N$ $\hat{C}$-theory |
|---|---|---|
| lattice | $\mathbb{Z}_N$ lattice gauge theory of $C$ or $\mathbb{Z}_N$ lattice model for $\hat{\phi}$ | $\mathbb{Z}_N$ lattice gauge theory of $\hat{C}$ or $\mathbb{Z}_N$ plaquettte Ising model |
| Lagrangian | $\mathcal{L} = \frac{N}{2(2\pi)}\hat{\phi}_{[ij]k}E^{[ij]k}$ | $\mathcal{L} = \frac{N}{2\pi}\phi\hat{E}$ |
| $\mathbb{Z}_N$ global symmetry | electric tensor $\exp[i\hat{\phi}^{k(ij)}]$  $(\mathbf{2},\mathbf{3'})$ tensor $\exp\left[i\int_{x_1^i}^{x_2^i} dx^i \oint dx^j \oint dx^k C^{[jk]i}\right]$ | electric $\exp[i\phi]$  $(\mathbf{1},\mathbf{3'})$ dipole $\exp\left[i\int_{x_1^i}^{x_2^i} dx^i \oint dx^j \oint dx^k \hat{C}\right]$ |
| defect | fractonic strips $\exp\left[i\int_{-\infty}^{\infty} dt \int_{x_1^k}^{x_2^k} dx^k \oint dx^j C_0^{jk})\right]$ | fractonic strings $\exp\left[i\int_{-\infty}^{\infty} dt \oint dx^i \hat{C}_0^{jk}\right]$ |
| ground state degeneracy on a torus | $N^{L^x+L^y+L^z-1}$ | $N^{L^x+L^y+L^z-2}$ |

Table 6: The $\mathbb{Z}_N$ gauge theories of $C$ in Section 6 and of $\hat{C}$ in Section 8. The fields $\hat{\phi}_{[ij]k}$ and $\phi$ are discussed in [4].

## 1.2 Relation to the X-Cube Model

We conclude with a discussion on the relation between these two $\mathbb{Z}_N$ theories and the X-cube model [9], one of the simplest gapped fracton models. See [5] for a general characterization of the various exotic $\mathbb{Z}_N$ symmetries in these models.

The global and the gauge symmetries of the $\mathbb{Z}_N$ $\hat{C}$-theory are related to those of the X-cube model in the continuum as follows. The $\mathbb{Z}_N$ $\hat{C}$ theory has a $\mathbb{Z}_N$ $(\mathbf{1}, \mathbf{3}')$ dipole *global* symmetry. This is the continuum limit of the planar subsystem symmetry in the plaquette Ising model of [9]. The X-cube model can be interpreted as the pure *gauge* theory of this symmetry, i.e., as a $\mathbb{Z}_N$ gauge theory of $A$ (see Section 2 of [5]).[2]

We can repeat the same analysis for the $\mathbb{Z}_N$ $C$-theory. The $\mathbb{Z}_N$ $C$-theory has a $\mathbb{Z}_N$ $(\mathbf{2}, \mathbf{3}')$ tensor *global* symmetry. The X-cube model can be interpreted alternatively as the pure *gauge* theory of this symmetry, i.e., as a $\mathbb{Z}_N$ gauge theory of $\hat{A}$ (see Section 3 of [5]). We have thus seen that the global symmetry of the $\mathbb{Z}_N$ $C$ and $\hat{C}$-theories are the gauge symmetries of the X-cube model.

Conversely, the global symmetry of the X-cube model is the gauge symmetry of either the $\mathbb{Z}_N$ $C$-theory or the $\mathbb{Z}_N$ $\hat{C}$-theory. The X-cube model has a $\mathbb{Z}_N$ $(\mathbf{3}', \mathbf{2})$ dipole *global* symmetry and a $\mathbb{Z}_N$ $(\mathbf{3}', \mathbf{1})$ tensor *global* symmetry. Indeed, the $\mathbb{Z}_N$ $C$-theory is the pure *gauge* theory of the $(\mathbf{3}', \mathbf{2})$ dipole symmetry. Similarly, the $\mathbb{Z}_N$ $\hat{C}$-theory is the pure *gauge* theory of the $(\mathbf{3}', \mathbf{1})$ tensor symmetry. We summarize these global and gauge symmetries of the three $\mathbb{Z}_N$ theories in Table 7.

*An analogy to standard $2 + 1d$ theories*

We would like to end the introduction by drawing an analogy with some ordinary relativistic theories in $2 + 1$ dimensions.

Let $\phi^{(0)}$ be a real, circle-valued scalar field and $c^{(2)}$ a two-form gauge field. Consider the $2 + 1$-dimensional $BF$ theory

$$\mathcal{L}_{\phi c} = \frac{N}{2\pi} \phi^{(0)} \wedge dc^{(2)} . \tag{1.6}$$

It describes a spontaneously broken zero-form $\mathbb{Z}_N$ global symmetry. Let us also consider the ordinary $2 + 1$-dimensional $\mathbb{Z}_N$ gauge theory

$$\mathcal{L}_{b\Phi} = \frac{1}{2\pi} b^{(2)} \wedge (d\Phi^{(0)} - N a^{(1)}) , \tag{1.7}$$

where $b^{(2)}$ is a two-form Lagrange multiplier, $\Phi^{(0}$ is a scalar, and $a^{(1)}$ is an ordinary $U(1)$

---

[2]See also [11, 5] for a $BF$-type Lagrangian for the X-cube model in the continuum.

|  | $\mathbb{Z}_N$ $C$-theory | $\mathbb{Z}_N$ $\hat{A}$-theory $=$ $\mathbb{Z}_N$ $A$-theory | $\mathbb{Z}_N$ $\hat{C}$-theory |
|---|---|---|---|
| global symmetry | $\mathbb{Z}_N$ $(\mathbf{2}, \mathbf{3}')$ | $\mathbb{Z}_N$ $(\mathbf{3}', \mathbf{2})$ and $\mathbb{Z}_N$ $(\mathbf{3}', \mathbf{1})$ | $\mathbb{Z}_N$ $(\mathbf{1}, \mathbf{3}')$ |
| pure gauge theory of | $\mathbb{Z}_N$ $(\mathbf{3}', \mathbf{2})$ | $\mathbb{Z}_N$ $(\mathbf{2}, \mathbf{3}')$ or $\mathbb{Z}_N$ $(\mathbf{1}, \mathbf{3}')$ | $\mathbb{Z}_N$ $(\mathbf{3}', \mathbf{1})$ |

Table 7: Global and gauge symmetries of three $\mathbb{Z}_N$ gauge theories. The $\mathbb{Z}_N$ symmetries are labeled by the $S_4$ representations $(\mathbf{R}_{\text{time}}, \mathbf{R}_{\text{space}})$ of the temporal and spatial currents for their $U(1)$ versions. See Table 1 of [5] for more detail. The $\mathbb{Z}_N$ $A$ and $\hat{A}$ theories are dual to each other, and they are the continuum limit of the X-cube model [5]. The $\mathbb{Z}_N$ $\hat{C}$-theory is the continuum limit of the plaquette Ising model in [9]. Furthermore, the $\mathbb{Z}_N$ $(\mathbf{1}, \mathbf{3}')$ dipole global symmetry is the continuum version of the planar subsystem symmetry of the plaquette Ising model. We did not include all the global symmetries of these models in the table above.

gauge field. Equivalently, we can consider its dual $BF$-type presentation [12–15]

$$\mathcal{L}_{a\hat{a}} = \frac{N}{2\pi} a^{(1)} \wedge d\hat{a}^{(1)} \ , \tag{1.8}$$

where both $a^{(1)}$ and $\hat{a}^{(1)}$ are one-form gauge fields.

Our $\mathbb{Z}_N$ gauge theory of $C$ (or of $\hat{C}$) is analogous to $\mathcal{L}_{\phi c}$, while the continuum limit of the X-cube model is analogous to $\mathcal{L}_{b\Phi}$ or $\mathcal{L}_{a\hat{a}}$. Indeed, the global symmetry of the $\mathcal{L}_{\phi c}$ theory is the gauge symmetry of $\mathcal{L}_{a\hat{a}}$, and vice versa. Specifically, the $\mathcal{L}_{\phi c}$ theory has an ordinary $\mathbb{Z}_N$ global symmetry generated by $e^{i \oint c^{(2)}}$, and $\mathcal{L}_{a\hat{a}}$ is the pure gauge theory of this symmetry. Conversely, the $\mathcal{L}_{a\hat{a}}$ theory has a $\mathbb{Z}_N$ one-form global symmetry generated by, for example, $e^{i \oint a^{(1)}}$, and $\mathcal{L}_{\phi c}$ is the pure gauge theory of this symmetry.

More generally, let $\mathcal{L}$ be the Lagrangian of a theory with global symmetry group $G$. It is common to gauge this global symmetry by coupling $\mathcal{L}$ to gauge fields of $G$. Denote the resulting theory as $\mathcal{L}^G$. A related theory is the pure gauge theory of $G$ without matter fields. Denote it by $\mathcal{L}'$. Often, the pure gauge theory $\mathcal{L}'$ can be obtained from the gauge theory with matter $\mathcal{L}^G$ by taking an appropriate limit of the parameters, e.g., making the matter fields heavy and decoupling them.

Using this notation, the Lagrangian $\mathcal{L}_{\phi c}$ (1.6) has a $G = \mathbb{Z}_N$ zero-form global symmetry. If we simply gauge it, we find a trivial theory. One way to do it is by writing it as a $U(1)$

theory $\mathcal{L}_{\phi c}^G = \frac{N}{2\pi}c^{(2)}(d\phi^{(0)} - a^{(1)})$. Instead, the pure $G = \mathbb{Z}_N$ gauge theory can be written as (1.7) $\mathcal{L}'_{\phi c} = \frac{1}{2\pi}b^{(2)}(d\Phi^{(0)} - Na^{(1)})$ or as in (1.8) $\mathcal{L}'_{\phi c} = \frac{N}{2\pi}a^{(1)}d\hat{a}^{(1)}$.

# 2 The $\varphi$-Theory

## 2.1 The XY-Cube Model

Consider a three-dimensional spatial, cubic lattice with periodic boundary conditions and place a phase variable $e^{i\varphi_s}$ at every site $s = (\hat{x}, \hat{y}, \hat{z})$. Let $L^x, L^y, L^z$ be the number of sites in the $x, y, z$ directions, respectively. When we later take the continuum limit, we will use $x^i = a\hat{x}^i$ $(i = 1, 2, 3)$, where $a$ is the lattice spacing, to label the coordinates. We will also use $\ell^i = aL^i$ to denote the physical size of the system.

The variable $\varphi_s$ is $2\pi$-periodic at each site, $\varphi_s \sim \varphi_s + 2\pi$. Let $\pi_s$ be the conjugate momentum of $\varphi_s$. They obey the commutation relation $[\varphi_s, \pi_{s'}] = i\delta_{s,s'}$. The $2\pi$-periodicity of $\varphi_s$ implies that the eigenvalues of $\pi_s$ are integers. The Hamiltonian is

$$H = \frac{u}{2}\sum_s \pi_s^2 - K\sum_s \cos(\Delta_{xyz}\varphi_s)$$

$$\Delta_{xyz}\varphi_s = \varphi_{s+(1,1,1)} - \varphi_{s+(0,1,1)} - \varphi_{s+(1,0,1)} - \varphi_{s+(1,1,0)} + \varphi_{s+(1,0,0)} + \varphi_{s+(0,1,0)} + \varphi_{s+(0,0,1)} - \varphi_s .$$

$$(2.1)$$

Since the interaction is around a cube, we will refer to this model as the XY-cube model. The system has a large number of $U(1)$ global symmetries, which grows quadratically in the size of the system. For every point $(\hat{x}_0, \hat{y}_0)$ in the $xy$-plane, there is a $U(1)$ global symmetry that acts as

$$U(1)_{\hat{x}_0\hat{y}_0} : \quad \varphi_s \to \varphi_s + \vartheta, \quad \forall s = (\hat{x}, \hat{y}, \hat{z}) \text{ with } (\hat{x}, \hat{y}) = (\hat{x}_0, \hat{y}_0) , \qquad (2.2)$$

where $\vartheta \in (0, 2\pi]$. Similar, there are $U(1)_{\hat{y}_0\hat{z}_0}$ and $U(1)_{\hat{x}_0\hat{z}_0}$ global symmetries associated to the sites in the $yz$-plane and $xz$-plane. There are $L_x + L_y + L_z - 1$ relations among these global symmetries. The composition of all the $U(1)_{\hat{x}_0\hat{y}_0}$ transformation with the same $\hat{x}_0$ is the same as the composition of all the $U(1)_{\hat{x}_0\hat{z}_0}$ with the same $\hat{x}_0$. This leads to $L_x$ relations. Similarly, there are $L_y, L_z$ relations associated to the $y, z$ directions. These relations are not all independent. This reduces the number of relations by 1. In total there are $L_xL_y + L_yL_z + L_xL_z - L_x - L_y - L_z + 1$ independent $U(1)$ global symmetries.

## 2.2 Continuum Lagrangian

The continuum limit of the lattice model is a real scalar field theory with Lagrangian [7]

$$\mathcal{L} = \frac{\mu_0}{2}(\partial_0\varphi)^2 - \frac{1}{2\mu}(\partial_x\partial_y\partial_z\varphi)^2 \ , \tag{2.3}$$

where $\mu_0$ and $\mu$ have mass dimension 2. This theory is a special case of the general class of theories in [16]. The equation of motion is

$$\mu_0\partial_0^2\varphi = \frac{1}{\mu}\partial_x^2\partial_y^2\partial_z^2\varphi \ . \tag{2.4}$$

As in the exotic theories in [3–5], discontinuous field configurations play an important role even in the continuum field theory. More specifically, we will discuss field configurations that are discontinuous in two of the three coordinates, but not in all three coordinates. These configurations are more continuous than a typical lattice configuration. See [3–5] for more discussion on this issue.

In the continuum field theory, this implies that the field $\varphi$ is locally subject to the discrete gauge symmetry

$$\varphi(t,x,y,z) \sim \varphi(t,x,y,z) + 2\pi w^{xy}(x,y) + 2\pi w^{yz}(y,z) + 2\pi w^{xz}(x,z) \ , \tag{2.5}$$

where $w^{ij}(x^i,x^j) \in \mathbb{Z}$. Because of the gauge symmetry, the operators $\partial_i\varphi$, $\partial_i\partial_j\varphi$ are not gauge invariant, while $e^{i\varphi}$ and $\partial_x\partial_y\partial_z\varphi$ are well-defined operators. The gauge symmetry allows nontrivial twisted configurations on a spatial 3-torus, for instance,

$$\begin{aligned}
2\pi &\left[ \frac{xyz}{\ell^x\ell^y\ell^z} - \frac{yz}{\ell^y\ell^z}\Theta(x-x_0) - \frac{xz}{\ell^x\ell^z}\Theta(y-y_0) - \frac{xy}{\ell^x\ell^y}\Theta(z-z_0) \right.\\
&\left. + \frac{x}{\ell^x}\Theta(y-y_0)\Theta(z-z_0) + \frac{y}{\ell^y}\Theta(x-x_0)\Theta(z-z_0) + \frac{z}{\ell^z}\Theta(x-x_0)\Theta(y-y_0) \right] \ .
\end{aligned} \tag{2.6}$$

## 2.3 Global Symmetries

We now discuss the exotic global symmetries of the continuum field theory.

### 2.3.1 Momentum Quadrupole Symmetry

The equation of motion

$$\partial_0 J_0 = \partial_x\partial_y\partial_z J^{xyz} \ , \tag{2.7}$$

implies a symmetry with currents

$$J_0 = \mu_0 \partial_0 \varphi \ ,$$
$$J^{xyz} = \frac{1}{\mu} \partial_x \partial_y \partial_z \varphi \ . \tag{2.8}$$

The currents $(J_0, J^{xyz})$ are in the $(\mathbf{1}, \mathbf{1}')$ representation of $S_4$. The corresponding symmetry is a quadrupole global symmetry [16].

The conserved charges are

$$Q^{ij}(x^i, x^j) = \oint dx^k \ J_0 \ . \tag{2.9}$$

These charges satisfy constraints

$$\oint dy \ Q^{xy}(x, y) = \oint dz \ Q^{xz}(x, z) \ ,$$
$$\oint dx \ Q^{xy}(x, y) = \oint dz \ Q^{yz}(y, z) \ , \tag{2.10}$$
$$\oint dy \ Q^{yz}(y, z) = \oint dx \ Q^{xz}(x, z) \ .$$

On the lattice, these correspond to $L_x L_y + L_y L_z + L_x L_z - L_x - L_y - L_z + 1$ linearly independent charges. They implement

$$\varphi(t, x, y, z) \rightarrow \varphi(t, x, y, z) + f^{xy}(x, y) + f^{yz}(y, z) + f^{xz}(x, z) \ . \tag{2.11}$$

The $\mathbb{Z}$ part of the quadrupole symmetry is gauged, so the global form of the symmetry is $U(1)$ as opposed to $\mathbb{R}$. This global quadrupole symmetry is the continuum limit of the $U(1)_{\hat{x}_0 \hat{y}_0}, U(1)_{\hat{y}_0 \hat{z}_0}, U(1)_{\hat{x}_0 \hat{z}_0}$ symmetries on the lattice. We will refer to this symmetry as *momentum quadrupole symmetry*.[3]

### 2.3.2 Winding Quadrupole Symmetry

The continuum theory also has a *winding quadrupole symmetry* with the conservation equation

$$\partial_0 J_0^{xyz} = \partial_x \partial_y \partial_z J \ . \tag{2.12}$$

---

[3]Note that the "momentum" here is momentum in the target space, not in spacetime.

The conserved currents

$$J_0^{xyz} = \frac{1}{2\pi}\partial_x\partial_y\partial_z\varphi \ ,$$

$$J = \frac{1}{2\pi}\partial_0\varphi \ . \tag{2.13}$$

are in the $(\mathbf{1'}, \mathbf{1})$ representation of $S_4$. The conserved charges are

$$Q_k^{xyz}(x^i, x^j) = \oint dx^k J_0^{xyz} \ . \tag{2.14}$$

Similar to the momentum qudrupole symmetry, there are $L_x L_y + L_y L_z + L_x L_z - L_x - L_y - L_z + 1$ independent charges. The winding quadrupole symmetry is absent on the lattice.

## 2.4 Momentum Mode

We start by analyzing the plane wave solutions in $\mathbb{R}^{3,1}$:

$$\varphi = C e^{i\omega + ik_i x^i} \ . \tag{2.15}$$

The equation of motion gives the dispersion relation

$$\omega^2 = \frac{1}{\mu\mu_0}k_x^2 k_y^2 k_z^2 \ . \tag{2.16}$$

For generic spacetime momenta $k_i$, the quantization is straightforward and leads to a gapless mode, albeit with a nonstandard dispersion relation.

The zero-energy solutions $\omega = 0$ are those modes with one of the three $k_i$'s vanishing. The momentum quadrupole symmetry maps one such zero-energy solution to another. Therefore, we will refer to these modes as the momentum modes. Classically, the momentum quadruple symmetry appears to be spontaneously broken, but this will turn out to be incorrect in the quantum theory.

Let us quantize the momentum modes of $\varphi$:

$$\varphi(t, x, y, z) = \varphi^{xy}(t, x, y) + \varphi^{yz}(t, y, z) + \varphi^{xz}(t, x, z) \ . \tag{2.17}$$

All the three fields on the RHS are point-wise $2\pi$ periodic by (2.5),

$$\varphi^{xy}(t, x, y) \sim \varphi^{xy}(t, x, y) + 2\pi w^{xy}(x, y) \ ,$$
$$\varphi^{yz}(t, y, z) \sim \varphi^{yz}(t, y, z) + 2\pi w^{yz}(y, z) \ , \tag{2.18}$$
$$\varphi^{xz}(t, x, z) \sim \varphi^{xz}(t, x, z) + 2\pi w^{xz}(x, z) \ .$$

They share common zero modes, which implies the following gauge symmetry parametrized

by $c^x(t,x), c^y(t,y), c^z(t,z)$,

$$\varphi^{xy}(t,x,y) \sim \varphi^{xy}(t,x,y) + c^x(t,x) - c^y(t,y) \ ,$$
$$\varphi^{yz}(t,y,z) \sim \varphi^{yz}(t,y,z) + c^y(t,y) - c^z(t,z) \ , \qquad (2.19)$$
$$\varphi^{xz}(t,x,z) \sim \varphi^{xz}(t,x,z) + c^z(t,z) - c^x(t,x) \ .$$

Note that shifting all $c^i(x^i)$ by the same zero mode does not contribute to the above gauge symmetry.

The Lagrangian of these momentum modes is

$$L = \frac{\mu_0}{2} \left[ \ell^z \oint dxdy \ (\dot\varphi^{xy})^2 + \ell^x \oint dydz \ (\dot\varphi^{yz})^2 + \ell^y \oint dxdz \ (\dot\varphi^{xz})^2 \right.$$
$$\left. +2 \oint dxdydz \ (\dot\varphi^{xy}\dot\varphi^{yz} + \dot\varphi^{yz}\dot\varphi^{xz} + \dot\varphi^{xz}\dot\varphi^{xy}) \right] \qquad (2.20)$$

The conjugate momenta are

$$\pi^{xy}(t,x,y) = \mu_0 \left( \ell^z \dot\varphi^{xy}(t,x,y) + \oint dz \ [\dot\varphi^{yz}(t,y,z) + \dot\varphi^{xz}(t,x,z)] \right) \ ,$$

$$\pi^{yz}(t,y,z) = \mu_0 \left( \ell^x \dot\varphi^{yz}(t,y,z) + \oint dx \ [\dot\varphi^{xy}(t,x,y) + \dot\varphi^{xz}(t,x,z)] \right) \ , \qquad (2.21)$$

$$\pi^{xz}(t,x,z) = \mu_0 \left( \ell^y \dot\varphi^{xz}(t,x,z) + \oint dy \ [\dot\varphi^{yz}(t,y,z) + \dot\varphi^{xy}(t,x,y)] \right) \ .$$

They are subject to the constraints

$$\oint dy \ \pi^{xy}(x,y) = \oint dz \ \pi^{xz}(x,z) \ ,$$
$$\oint dx \ \pi^{xy}(x,y) = \oint dz \ \pi^{yz}(y,z) \ , \qquad (2.22)$$
$$\oint dy \ \pi^{yz}(y,z) = \oint dx \ \pi^{xz}(x,z) \ ,$$

which can be thought of as Gauss laws from the gauge symmetry (2.19). In fact, the momenta are the charges of the momentum quadrupole symmetry, $Q^{ij}(x^i,x^j) = \pi^{ij}(x^i,x^j)$.

The point-wise periodicity of $\varphi^{ij}$ implies that their conjugate momenta $\pi^{ij}$ are linear

combination of delta functions with integer coefficients:

$$Q^{xy}(x, y) = \pi^{xy}(x, y) = \sum_{\alpha\beta} N^{xy}_{\alpha\beta}\delta(x - x_\alpha)\delta(y - y_\beta) \ ,$$

$$Q^{yz}(y, z) = \pi^{yz}(y, z) = \sum_{\beta\gamma} N^{yz}_{\beta\gamma}\delta(y - y_\beta)\delta(z - z_\gamma) \ , \qquad (2.23)$$

$$Q^{xz}(x, z) = \pi^{xz}(x, z) = \sum_{\alpha\gamma} N^{xz}_{\alpha\gamma}\delta(x - x_\alpha)\delta(z - z_\gamma) \ ,$$

where $N^{xy}_{\alpha\beta}, N^{yz}_{\beta\gamma}, N^{xz}_{\alpha\gamma} \in \mathbb{Z}$ satisfy the constraints (2.22)

$$N^x_\alpha \equiv \sum_\beta N^{xy}_{\alpha\beta} = \sum_\gamma N^{xz}_{\alpha\gamma} \ ,$$

$$N^y_\beta \equiv \sum_\alpha N^{xy}_{\alpha\beta} = \sum_\gamma N^{yz}_{\beta\gamma} \ ,$$

$$\qquad\qquad (2.24)$$

$$N^z_\gamma \equiv \sum_\alpha N^{xz}_{\alpha\gamma} = \sum_\beta N^{yz}_{\beta\gamma} \ ,$$

$$N \equiv \sum_\alpha N^x_\alpha = \sum_\beta N^y_\beta = \sum_\gamma N^z_\gamma \ .$$

Here $\{x_\alpha\}$, $\{y_\beta\}$, and $\{z_\gamma\}$ are finite sets of points along $x$, $y$, and $z$ axes respectively.

The Hamiltonian can be found to be

$$H = \oint dxdy \ \pi^{xy}\dot{\varphi}^{xy} + \oint dydz \ \pi^{yz}\dot{\varphi}^{yz} + \oint dxdz \ \pi^{xz}\dot{\varphi}^{xz} - L \ ,$$

$$= \frac{1}{2\mu_0 \ell^x \ell^y \ell^z}\left[ \ell^x \ell^y \oint dxdy \ (\pi^{xy})^2 + \ell^y \ell^z \oint dydz \ (\pi^{yz})^2 + \ell^x \ell^z \oint dxdz \ (\pi^{xz})^2 \right. \qquad (2.25)$$

$$\left. - \oint dxdydz \ (\ell^x \pi^{xz}\pi^{xy} + \ell^y \pi^{xy}\pi^{yz} + \ell^z \pi^{yz}\pi^{xz}) + \left(\oint dxdy \ \pi^{xy}\right)^2 \right] \ .$$

The configurations with the lowest nonzero momentum quadrupole symmetry charges are

$$\pi^{xy} = \delta(x - x_0)\delta(y - y_0) \ , \quad \pi^{yz} = \delta(y - y_0)\delta(z - z_0) \ , \quad \pi^{xz} = \delta(x - x_0)\delta(z - z_0) \ , \quad (2.26)$$

for any $x_0$, $y_0$, and $z_0$. Their energy is

$$H = \frac{1}{2\mu_0 \ell^x \ell^y \ell^z}\left[(\ell^x \ell^y + \ell^y \ell^z + \ell^x \ell^z)\delta(0)^2 - (\ell^y + \ell^z + \ell^x)\delta(0) + 1\right] \ . \qquad (2.27)$$

The meaning of $\delta(0)$ here and elsewhere in this paper is as in [3–5]. On a lattice, $\delta(0)$

stands for $\frac{1}{a}$, where $a$ is the lattice spacing. So these modes have energy of order $\frac{1}{a^2}$ and they becomes infinite in the continuum limit.

More generally, the energy of the generic momentum mode (2.23) is

$$H = \frac{1}{2\mu_0 \ell^x \ell^y \ell^z} \left[ \ell^x \ell^y \sum_{\alpha\beta} (N_{\alpha\beta}^{xy})^2 \delta(0)^2 + \ell^y \ell^z \sum_{\beta\gamma} (N_{\beta\gamma}^{yz})^2 \delta(0)^2 + \ell^x \ell^z \sum_{\alpha\gamma} (N_{\alpha\gamma}^{xz})^2 \delta(0)^2 \right.$$
$$\left. - \ell^x \sum_{\alpha} (N_{\alpha}^x)^2 \delta(0) - \ell^y \sum_{\beta} (N_{\beta}^y)^2 \delta(0) - \ell^z \sum_{\gamma} (N_{\gamma}^z)^2 \delta(0) + N^2 \right] . \tag{2.28}$$

We conclude that in the quantum theory, the energy of all the momentum modes is infinite. The momentum symmetry, which is spontaneously broken classically, is restored at quantum level. Even more so, the energy of the charged states is infinite.

On a lattice with lattice spacing $a$, this infinity can be regularized to get an energy of the order of

$$\frac{1}{\mu_0} \frac{1}{\ell a^2} . \tag{2.29}$$

In contrast, the energy of typical fluctuations on the lattice is of the order of $1/(\mu_0 a^3)$, which is parametrically larger than the energy of these momentum modes. In the continuum limit $a \to 0$, the momentum modes are much heavier than generic modes in (2.16) with $k_x \neq 0$, $k_y \neq 0$, and $k_z \neq 0$ whose energy scale is $1/(\mu_0 \ell^3)$. In finite volume, the ground state is the unique momentum mode where the momentum quadrupole charges vanish, and the classically zero-energy configurations with $k_x = 0$, or $k_y = 0$, or $k_z = 0$ are lifted quantum mechanically. In infinite volume $\ell \to \infty$, the energy of the modes with generic $k_i$ is gapless, but the states charged under the momentum quadrupole symmetry still have infinite energy in the continuum limit.

Similar to the discussions in [3–5], the quantitative results for the energy of the momentum modes are not universal, but their qualitative scaling in $1/a^2$ is. For example, let us consider adding

$$g(\partial_x \partial_0 \varphi)^2 \tag{2.30}$$

to the minimal Lagrangian (2.3), with the coupling $g$ of order $a^2$. The term shifts the energy of the momentum modes by an amount of order $g/a^4 \sim 1/a^2$.

## 2.5   Winding Mode

The most general winding configuration can be obtained by taking linear combination of (2.6)

$$
\varphi(t,x,y,z) = 2\pi \left[ W \frac{xyz}{\ell^x \ell^y \ell^z} - \frac{yz}{\ell^y \ell^z} \sum_\alpha W_\alpha^x \Theta_\alpha(x) - \frac{xz}{\ell^x \ell^z} \sum_\beta W_\beta^y \Theta_\beta(y) - \frac{xy}{\ell^x \ell^y} \sum_\gamma W_\gamma^z \Theta_\gamma(z) \right.
$$
$$
\left. + \frac{x}{\ell^x} \sum_{\beta\gamma} W_{\beta\gamma}^{yz} \Theta_\beta(y)\Theta_\gamma(z) + \frac{y}{\ell^y} \sum_{\alpha\gamma} W_{\alpha\gamma}^{xz} \Theta_\alpha(x)\Theta_\gamma(z) + \frac{z}{\ell^z} \sum_{\alpha\beta} W_{\alpha\beta}^{xy} \Theta_\alpha(x)\Theta_\beta(y) \right] ,
$$

$$(2.31)$$

where $\Theta_\alpha(x) \equiv \Theta(x - x_\alpha)$. All the coefficients are integers and they are related by

$$
W_\alpha^x = \sum_\beta W_{\alpha\beta}^{xy} = \sum_\gamma W_{\alpha\gamma}^{xz} ,
$$
$$
W_\beta^y = \sum_\alpha W_{\alpha\beta}^{xy} = \sum_\gamma W_{\beta\gamma}^{yz} ,
$$
$$
W_\gamma^z = \sum_\beta W_{\beta\gamma}^{yz} = \sum_\alpha W_{\alpha\gamma}^{xz} ,
$$
$$
W = \sum_\alpha W_\alpha^x = \sum_\beta W_\beta^y = \sum_\gamma W_\gamma^z .
$$

$$(2.32)$$

The winding quadrupole charge density of this configuration is

$$
J_0^{xyz} = W \frac{1}{\ell^x \ell^y \ell^z} - \frac{1}{\ell^y \ell^z} \sum_\alpha W_\alpha^x \delta_\alpha(x) - \frac{1}{\ell^x \ell^z} \sum_\beta W_\beta^y \delta_\beta(y) - \frac{1}{\ell^x \ell^y} \sum_\gamma W_\gamma^z \delta_\gamma(z)
$$
$$
+ \frac{1}{\ell^x} \sum_{\beta\gamma} W_{\beta\gamma}^{yz} \delta_\beta(y)\delta_\gamma(z) + \frac{1}{\ell^y} \sum_{\alpha\gamma} W_{\alpha\gamma}^{xz} \delta_\alpha(x)\delta_\gamma(z) + \frac{1}{\ell^z} \sum_{\alpha\beta} W_{\alpha\beta}^{xy} \delta_\alpha(x)\delta_\beta(y) ,
$$

$$(2.33)$$

where $\delta_\alpha(x) \equiv \delta(x - x_\alpha)$. The winding configuration has quadrupole charges

$$
Q_x^{xyz}(y,z) = \frac{1}{2\pi} \oint dx \, J_0^{xyz} = \sum_{\beta\gamma} W_{\beta\gamma}^{yz} \delta_\beta(y)\delta_\gamma(z) ,
$$
$$
Q_y^{xyz}(x,z) = \frac{1}{2\pi} \oint dy \, J_0^{xyz} = \sum_{\alpha\gamma} W_{\alpha\gamma}^{xz} \delta_\alpha(x)\delta_\gamma(z) ,
$$
$$
Q_z^{xyz}(x,y) = \frac{1}{2\pi} \oint dz \, J_0^{xyz} = \sum_{\alpha\beta} W_{\alpha\beta}^{xy} \delta_\alpha(x)\delta_\beta(y) .
$$

$$(2.34)$$

The energy of the winding configuration is

$$
\begin{aligned}
H &= \frac{1}{2\mu} \oint dxdydz \ (\partial_x \partial_y \partial_z \varphi)^2 \\
&= \frac{2\pi^2}{\mu \ell^x \ell^y \ell^z} \Bigg[ \ell^x \ell^y \sum_{\alpha\beta} (W^{xy}_{\alpha\beta})^2 \delta(0)^2 + \ell^y \ell^z \sum_{\beta\gamma} (W^{yz}_{\beta\gamma})^2 \delta(0)^2 + \ell^x \ell^z \sum_{\alpha\gamma} (W^{xz}_{\alpha\gamma})^2 \delta(0)^2 \\
&\qquad - \ell^x \sum_{\alpha} (W^x_{\alpha})^2 \delta(0) - \ell^y \sum_{\beta} (W^y_{\beta})^2 \delta(0) - \ell^z \sum_{\gamma} (W^z_{\gamma})^2 \delta(0) + W^2 \Bigg] \ .
\end{aligned}
\tag{2.35}
$$

So, the energy of winding modes with nonzero winding quadrupole charges is infinite. Placing the theory on a lattice with lattice spacing $a$, the energy of such winding modes scales as $1/(\mu \ell a^2)$.

Similar to the discussions for the momentum modes, the quantitative results for the energy of the winding modes are not universal, but their qualitative scaling in $1/a^2$ is. For example, let us consider adding

$$
g(\partial_x^2 \partial_y \partial_z \varphi)^2
\tag{2.36}
$$

to the minimal Lagrangian (2.3), with the coupling $g$ of order $a^2$.[4] The term shifts the energy of the winding modes by an amount of order $g/a^4 \sim 1/a^2$.

## 2.6 Self-Duality

The Euclidean Lagrangian of the theory can be rewritten as

$$
\mathcal{L}_E = \frac{\mu_0}{2} B^2 + \frac{1}{2\mu} E_{xyz} E^{xyz} + \frac{i}{2\pi} \widetilde{B}^{xyz} (\partial_x \partial_y \partial_z \varphi - E_{xyz}) + \frac{i}{2\pi} \widetilde{E}(\partial_\tau \varphi - B) \ ,
\tag{2.37}
$$

where $B, E_{xyz}, \widetilde{E}, \widetilde{B}^{xyz}$ are independent fields. If we integrate out these fields, we recover the original Lagrangian.

Instead, we integrate out only $B, E_{xyz}$

$$
\mathcal{L}_E = \frac{1}{8\pi^2 \mu_0} \widetilde{E}^2 + \frac{\mu}{8\pi^2} \widetilde{B}_{xyz} \widetilde{B}^{xyz} + \frac{i}{2\pi} \widetilde{B}^{xy} \partial_x \partial_y \partial_z \varphi + \frac{i}{2\pi} \widetilde{E} \partial_\tau \varphi \ .
\tag{2.38}
$$

Next, we integrate out $\varphi$ to a find a constraint

$$
\partial_\tau \widetilde{E} = \partial_x \partial_y \partial_z \widetilde{B}^{xyz} \ .
\tag{2.39}
$$

---

[4]Under the duality in Section 2.6, this higher derivative term is dual to the term $g(\partial_0 \partial_x \varphi)^2$.

This can be solved locally in terms of a field $\varphi^{xyz}$ in the $\mathbf{1}'$ of $S_4$:

$$\begin{aligned}
\widetilde{E} &= \partial_x \partial_y \partial_z \varphi^{xyz} \ , \\
\widetilde{B}^{xyz} &= \partial_\tau \varphi^{xyz} \ ,
\end{aligned} \tag{2.40}$$

The Lagrangian becomes

$$\mathcal{L}_E = \frac{\widetilde{\mu}_0}{2} (\partial_\tau \varphi^{xyz})^2 + \frac{1}{2\widetilde{\mu}} (\partial_x \partial_y \partial_z \varphi^{xyz})^2 \ , \tag{2.41}$$

where

$$\widetilde{\mu}_0 = \frac{\mu}{4\pi^2}, \quad \widetilde{\mu} = 4\pi^2 \mu_0 \ . \tag{2.42}$$

Hence the $\varphi$ theory is dual to the $\varphi^{xyz}$ theory. The momentum and the winding quadrupole symmetries as well as their charged states are interchanged under the duality. This duality is analogous to the T-duality of the ordinary $1+1$-dimensional compact scalar theory.

We refer to this duality between the $\varphi$ and the $\varphi^{xyz}$ theories as a self-duality for the following reason. Since the spatial rotation symmetry $S_4$ is discrete, it can be redefined by discrete internal global symmetries. Both theories have a charge conjugation symmetry, $C : \varphi \to -\varphi, C : \varphi^{xyz} \to -\varphi^{xyz}$. The total unitary global symmetry is therefore $\mathbb{Z}_2^C \times S_4$. Let $R^i$ be the 90 degree rotations around the $x^i$-axis of the spatial $S_4$ rotation, $(R^i)^4 = 1$. The fields $\varphi$ and $\varphi^{xyz}$ are in the $\mathbf{1}$ and $\mathbf{1}'$ of this $S_4$, respectively. However, we can consider a different $S_4$ subgroup of $\mathbb{Z}_2^C \times S_4$ generated by $R^i C$. We denote this new subgroup by $S_4^C$. Then the fields $\varphi$ and $\varphi^{xyz}$ are in the $\mathbf{1}'$ and $\mathbf{1}$ of $S_4^C$, respectively. Said differently, the representations for $\varphi$ and $\varphi^{xyz}$ are related by an outer automorphism of $\mathbb{Z}_2^C \times S_4$. Such a nontrivial map of representations by outer automorphisms is common in dualities. (See a related discussion in [3] and in [17].)

# 3 $U(1)$ Tensor Gauge Theory of $B$

We can gauge the momentum quadrupole symmetry by coupling the currents to the tensor gauge field $(B_0, B_{xyz})$:[5]

$$J_0 B_0 + J^{xyz} B_{xyz} \ . \tag{3.1}$$

The current conservation equation $\partial_0 J_0 = \partial_x \partial_y \partial_z J^{xyz}$ implies the gauge transformation

$$\begin{aligned}
B_0 &\sim B_0 + \partial_0 \alpha \ , \\
B_{xyz} &\sim B_{xyz} + \partial_x \partial_y \partial_z \alpha \ .
\end{aligned} \tag{3.2}$$

---

[5]Since there is no magnetic field in this gauge theory, we hope it does not cause any confusion to use $B$ to denote the gauge fields.

The gauge invariant electric field is

$$E_{xyz} = \partial_0 B_{xyz} - \partial_x \partial_y \partial_z B_0 \ , \tag{3.3}$$

while there is no magnetic field.

## 3.1 Lattice Tensor Gauge Theory

Let us discuss the lattice version of the $U(1)$ tensor gauge theory without matter. We have a $U(1)$ phase variable $U_c = e^{ia^3 B_c}$ and its conjugate variable $E_c$ at every cube $c$. The gauge transformation $e^{i\alpha_s}$ is a $U(1)$ phase associated with each site $s$. Under the gauge transformation,

$$U_c \sim U_c \, e^{i\Delta_{xyz}\alpha_s} \ , \tag{3.4}$$

where $\Delta_{xyz}\alpha_s$ is a linear combination of $\alpha_s$ around the cube $c$ defined in (2.1).

There are two types of gauge invariant operators. The first type is an operator $E_c$ at a single cube. The second type is a product of $U_c$'s along $x$ direction at a fixed $y, z$, or similar operators along $y$ or $z$ direction.

The Hamiltonian is

$$H = \frac{1}{g^2} \sum_c E_c^2 \tag{3.5}$$

and the physical states satisfy Gauss law

$$G_s \equiv \sum_{c \ni s} \epsilon_c E_c = 0 \ , \tag{3.6}$$

where the sum over $c$ is an oriented sum ($\epsilon_c = \pm 1$) over the 8 cubes that share a common site $s$.

The lattice model has an electric tensor symmetry whose conserved charge is proportional to $E_c$. The electric tensor symmetry rotates the phase of $U_c$ at a single cube, $U_c \rightarrow e^{i\varphi}U_c$. Using Gauss law (3.6), the dependence of the conserved charge $E_c$ on $c$ is a function of $(\hat{x}, \hat{y})$ plus a function of $(\hat{y}, \hat{z})$ plus a function of $(\hat{x}, \hat{z})$.

## 3.2 Continuum Lagrangian

The Lorentzian Lagrangian of the pure tensor gauge theory is [7,8]

$$\mathcal{L} = \frac{1}{g_e^2} E_{xyz}^2 + \frac{\theta}{2\pi} E_{xyz} \ . \tag{3.7}$$

We will soon show that the total electric flux in Euclidean space is quantized $\oint d\tau dx dy dz E_{xyz} \in 2\pi\mathbb{Z}$, and therefore the theta angle is $2\pi$ periodic, $\theta \sim \theta + 2\pi$. The equations of motion are

$$\partial_0 E_{xyz} = 0 \ , \quad \partial_x \partial_y \partial_z E_{xyz} = 0 \ , \tag{3.8}$$

where the second equation is the Gauss law.

If $\theta = 0, \pi$, the global symmetry includes $\mathbb{Z}_2^C \times S_4$, where $\mathbb{Z}_2^C$ is a charge conjugation symmetry that flips the sign of $B_0, B_{xyz}$. Other values of $\theta$ break the $\mathbb{Z}_2^C \times S_4$ to $S_4$. In addition, for every value of $\theta$ there are parity and time reversal symmetries, under which $E_{xyz}$ is invariant.

## 3.3  Fluxes

We place the theory on a Euclidean 4-torus with lengths $\ell^x$, $\ell^y$, $\ell^z$, $\ell^\tau$, and explore its bundles. For that, we need to understand the possible nontrivial transition functions.

Let us take the transition function as we shift $\tau \to \tau + \ell^\tau$ to be the gauge transformation (2.6),[6]

$$g_{(\tau)}(x, y, z) = 2\pi \left[ \frac{x}{\ell^x}\Theta(y - y_0)\Theta(z - z_0) + \frac{y}{\ell^y}\Theta(x - x_0)\Theta(z - z_0) + \frac{z}{\ell^z}\Theta(x - x_0)\Theta(y - y_0) \right.$$
$$\left. - \frac{yz}{\ell^y \ell^z}\Theta(x - x_0) - \frac{xz}{\ell^x \ell^z}\Theta(y - y_0) - \frac{xy}{\ell^x \ell^y}\Theta(z - z_0) + \frac{xyz}{\ell^x \ell^y \ell^z} \right] \ . \tag{3.9}$$

i.e.,

$$B_{xyz}(\tau + \ell^\tau, x, y, z) = B_{xyz}(\tau, x, y, z) + \partial_x \partial_y \partial_z g_{(\tau)}(x, y, z) \tag{3.10}$$

For example, we can have

$$B_{xyz}(\tau, x, y, z) = 2\pi \frac{\tau}{\ell^\tau} \left[ \frac{1}{\ell^x}\delta(y - y_0)\delta(z - z_0) + \frac{1}{\ell^y}\delta(x - x_0)\delta(z - z_0) + \frac{1}{\ell^z}\delta(x - x_0)\delta(y - y_0) \right.$$
$$\left. - \frac{1}{\ell^y \ell^z}\delta(x - x_0) - \frac{1}{\ell^x \ell^z}\delta(y - y_0) - \frac{1}{\ell^x \ell^y}\delta(z - z_0) + \frac{1}{\ell^x \ell^y \ell^z} \right] \ . \tag{3.11}$$

---

[6]As in all our theories, we allow certain singular configurations provided the terms in the Lagrangian are not too singular (see [3–5]). Here we follow the same rules as in [3,4]. It would be nice to understand better the precise rules controlling these singularities.

Such a configuration gives rise to electric flux

$$e_{(xy)}(x,y) = \oint d\tau \oint dz \, E_{xyz} = 2\pi\delta(x - x_0)\delta(y - y_0) \,,$$

$$e_{(yz)}(y,z) = \oint d\tau \oint dx \, E_{xyz} = 2\pi\delta(y - y_0)\delta(z - z_0) \,, \tag{3.12}$$

$$e_{(xz)}(x,z) = \oint d\tau \oint dy \, E_{xyz} = 2\pi\delta(x - x_0)\delta(z - z_0) \,.$$

With more general twists, we can have

$$e_{(xy)}(x,y) = \oint d\tau \oint dz \, E_{xyz} = 2\pi \sum_{\alpha\beta} n_{\alpha\beta}^{xy}\delta(x - x_\alpha)\delta(y - y_\beta) \,,$$

$$e_{(yz)}(y,z) = \oint d\tau \oint dx \, E_{xyz} = 2\pi \sum_{\beta\gamma} n_{\beta\gamma}^{yz}\delta(y - y_\beta)\delta(z - z_\gamma) \,, \tag{3.13}$$

$$e_{(xz)}(x,z) = \oint d\tau \oint dy \, E_{xyz} = 2\pi \sum_{\alpha\gamma} n_{\alpha\gamma}^{xz}\delta(x - x_\alpha)\delta(z - z_\gamma) \,,$$

where $n_{\alpha\beta}^{xy}, n_{\beta\gamma}^{yz}, n_{\alpha\gamma}^{xz} \in \mathbb{Z}$ satisfy

$$\sum_\beta n_{\alpha\beta}^{xy} = \sum_\gamma n_{\alpha\gamma}^{xz} \,, \quad \sum_\alpha n_{\alpha\beta}^{xy} = \sum_\gamma n_{\beta\gamma}^{yz} \,, \quad \sum_\alpha n_{\alpha\gamma}^{xz} = \sum_\beta n_{\beta\gamma}^{yz} \,. \tag{3.14}$$

We can also write the above fluxes in the integrated form

$$e_{(xy)}(x_1, x_2, y_1, y_2) \equiv \oint d\tau \int_{x_1}^{x_2} dx \int_{y_1}^{y_2} dy \oint dz \, E_{xyz} \in 2\pi\mathbb{Z} \,,$$

$$e_{(yz)}(y_1, y_2, z_1, z_2) \equiv \oint d\tau \oint dx \int_{y_1}^{y_2} dy \int_{z_1}^{z_2} dz \, E_{xyz} \in 2\pi\mathbb{Z} \,, \tag{3.15}$$

$$e_{(xz)}(x_1, x_2, z_1, z_2) \equiv \oint d\tau \int_{x_1}^{x_2} dx \oint dy \int_{z_1}^{z_2} dz \, E_{xyz} \in 2\pi\mathbb{Z} \,.$$

## 3.4 Global Symmetry

The equations of motion can be interpreted as the current conservation equation and a differential condition for an *electric tensor symmetry*

$$\partial_0 J_0^{xyz} = 0 \,, \quad \partial_x\partial_y\partial_z J_0^{xyz} = 0 \,, \tag{3.16}$$

with the current

$$J_0^{xyz} = \frac{2}{g_e^2}E_{xyz} + \frac{\theta}{2\pi} \,. \tag{3.17}$$

We define the current with a shift by $\theta/2\pi$ such that the conserved charge is quantized to be an integer (see (3.34)).

There is an integer conserved charge at every point in space, which coincides with the current itself:

$$Q(x, y, z) = J_0^{xyz} = N^{xy}(x, y) + N^{yz}(y, z) + N^{xz}(x, z) , \tag{3.18}$$

where $N^{ij}(x^i, x^j) \in \mathbb{Z}$. The differential condition $\partial_x \partial_y \partial_z J_0^{xyz} = 0$ constrains the charge $Q$ to have the above form. This conserved charge exists also on the lattice.

Up to a gauge transformation, the electric tensor symmetry acts on the gauge fields as

$$B_{xyz} \to B_{xyz} + c^{xy}(x, y) + c^{yz}(y, z) + c^{xz}(x, z) . \tag{3.19}$$

Note that it leaves the electric field invariant.

The charge objects under this electric global symmetry are the gauge-invariant extended operators defined at a fixed time:

$$
\begin{aligned}
W_{(xy)}(x_1, x_2; y_1, y_2) &= \exp\left[i \int_{x_1}^{x_2} dx \int_{y_1}^{y_2} dy \oint dz \; B_{xyz}\right] , \\
W_{(yz)}(y_1, y_2; z_1, z_2) &= \exp\left[i \int_{y_1}^{y_2} dy \int_{z_1}^{z_2} dz \oint dx \; B_{xyz}\right] , \\
W_{(xz)}(x_1, x_2; z_1, z_2) &= \exp\left[i \int_{x_1}^{x_2} dx \int_{z_1}^{z_2} dz \oint dy \; B_{xyz}\right] .
\end{aligned}
\tag{3.20}
$$

Only integer powers of these operators are invariant under the large gauge transformation (3.9). We can refer to such operators as Wilson tubes. These tube operators are the continuum version of the lattice operators constructed as products of $U_c$ along a line. The symmetry operator $\mathcal{U}(\beta; x, y, z) = e^{i\beta Q(x,y,z)}$ transforms the basic Wilson tube by $e^{i\beta}$ if the symmetry operator intersects the Wilson tube. Otherwise, the symmetry operator commutes with the Wilson tube.

## 3.5 Defects as Fractons

We now discuss defects that are extended in the time direction. The simplest kind of such a defect is

$$\exp\left[i \int_{-\infty}^{\infty} dt \; B_0\right] . \tag{3.21}$$

Its exponent is quantized by imposing invariance under a large gauge transformation. This describes a single static charged particle. A single particle cannot move in space because of the gauge symmetry.

However, four of them that form a quadrupole can move collectively. Consider four

particles in this configuration: $+1$ charge at $(x_1, y_1)$ and $(x_2, y_2)$, and $-1$ charge at $(x_1, y_2)$ and $(x_2, y_1)$. The configuration can move in time along a curve in the $(z, t)$ plane, $z(t)$. Their motion is described by the gauge-invariant defect

$$W(x_1, x_2, y_1, y_2, \mathcal{C}) = \exp\left[ i \int_{x_1}^{x_2} dx \int_{y_1}^{y_2} dy \int_{\mathcal{C}} (dt\ \partial_x \partial_y B_0 + dz\ B_{xyz}) \right] . \tag{3.22}$$

The operator is gauge-invariant for any curve $\mathcal{C}$ without endpoints. Similarly, we can have quadrupole moving collectively in the $x$ and $y$ directions.

## 3.6   Electric Modes

We place the system on a spatial torus with lengths $\ell^x, \ell^y, \ell^z$ and study its spectrum.

We pick the temporal gauge $B_0 = 0$. Using the Gauss law

$$\partial_x \partial_y \partial_z E_{xyz} = 0 , \tag{3.23}$$

up to time independent gauge transformations, we have

$$B_{xyz} = \frac{1}{\ell^z} f^{xy}(t, x, y) + \frac{1}{\ell^x} f^{yz}(t, y, z) + \frac{1}{\ell^y} f^{xz}(t, x, z) , \tag{3.24}$$

where we picked the normalization for later convenience. Note that there are no modes with nontrivial momentum in all the three directions.

Only the sum of the zero modes of $\ell^z f^{xy}(t, z, y)$, etc., is physical. So, there is a gauge symmetry:

$$\begin{aligned}
f^{xy}(t, x, y) &\sim f^{xy}(t, x, y) + \ell^z c^x(t, x) - \ell^z c^y(t, y) , \\
f^{yz}(t, y, z) &\sim f^{yz}(t, y, z) + \ell^x c^y(t, y) - \ell^x c^z(t, z) , \\
f^{xz}(t, x, z) &\sim f^{xz}(t, x, z) + \ell^y c^z(t, z) - \ell^y c^x(t, x) .
\end{aligned} \tag{3.25}$$

Note that shifting all $c^i(x^i)$ by the same zero mode does not contribute to the above gauge symmetry. To remove the gauge ambiguity, we define the gauge invariant variables

$$\begin{aligned}
\bar{f}^{xy}(t, x, y) &= f^{xy}(t, x, y) + \frac{1}{\ell^y} \oint dz\ f^{xz}(t, x, z) + \frac{1}{\ell^x} \oint dz\ f^{yz}(t, y, z) , \\
\bar{f}^{yz}(t, y, z) &= f^{yz}(t, y, z) + \frac{1}{\ell^z} \oint dx\ f^{xy}(t, x, y) + \frac{1}{\ell^y} \oint dx\ f^{xz}(t, x, z) , \\
\bar{f}^{xz}(t, x, z) &= f^{xz}(t, x, z) + \frac{1}{\ell^x} \oint dy\ f^{yz}(t, y, z) + \frac{1}{\ell^z} \oint dy\ f^{xy}(t, x, y) .
\end{aligned} \tag{3.26}$$

However, these variables are not all independent; they are subject to the constraints

$$\oint dy \ \bar{f}^{xy}(t,x,y) = \oint dz \ \bar{f}^{xz}(t,x,z) \ ,$$

$$\oint dz \ \bar{f}^{yz}(t,y,z) = \oint dx \ \bar{f}^{xy}(t,x,y) \ , \tag{3.27}$$

$$\oint dx \ \bar{f}^{xz}(t,x,z) = \oint dy \ \bar{f}^{yz}(t,y,z) \ .$$

The gauge transformation (3.9) implies the following identifications of $\bar{f}^{ij}$:

$$\bar{f}^{xy}(t,x,y) \sim \bar{f}^{xy}(t,x,y) + 2\pi\delta(x-x_0)\delta(y-y_0) \ ,$$
$$\bar{f}^{yz}(t,y,z) \sim \bar{f}^{yz}(t,y,z) + 2\pi\delta(y-y_0)\delta(z-z_0) \ , \tag{3.28}$$
$$\bar{f}^{xz}(t,x,z) \sim \bar{f}^{xz}(t,x,z) + 2\pi\delta(x-x_0)\delta(z-z_0) \ ,$$

for each $x_0$, $y_0$, and $z_0$. On a lattice with $L^i$ sites in $x^i$ direction, it may seem like there are $L^x L^y L^z$ identifications. However, there are only $L^x L^y + L^y L^z + L^x L^z - L^x - L^y - L^z + 1$ identifications. To see this, consider $z_0 = 0$,

$$\bar{f}^{xy}(t,x,y) \sim \bar{f}^{xy}(t,x,y) + 2\pi\delta(x-x_0)\delta(y-y_0) \ ,$$
$$\bar{f}^{yz}(t,y,z) \sim \bar{f}^{yz}(t,y,z) + 2\pi\delta(y-y_0)\delta(z) \ , \tag{3.29}$$
$$\bar{f}^{xz}(t,x,z) \sim \bar{f}^{xz}(t,x,z) + 2\pi\delta(x-x_0)\delta(z) \ .$$

There are $L^x L^y$ such identifications. Next, consider $x_0 = 0$,

$$\bar{f}^{xy}(t,x,y) \sim \bar{f}^{xy}(t,x,y) \ ,$$
$$\bar{f}^{yz}(t,y,z) \sim \bar{f}^{yz}(t,y,z) + 2\pi\delta(y-y_0)\delta(z-z_0) - 2\pi\delta(y-y_0)\delta(z) \ , \tag{3.30}$$
$$\bar{f}^{xz}(t,x,z) \sim \bar{f}^{xz}(t,x,z) + 2\pi\delta(x)\delta(z-z_0) - 2\pi\delta(x)\delta(z) \ .$$

There are $L^y(L^z - 1)$ such identifications. Finally, consider $y_0 = 0$,

$$\bar{f}^{xy}(t,x,y) \sim \bar{f}^{xy}(t,x,y) \ ,$$
$$\bar{f}^{yz}(t,y,z) \sim \bar{f}^{yz}(t,y,z) \ ,$$
$$\bar{f}^{xz}(t,x,z) \sim \bar{f}^{xz}(t,x,z) + 2\pi\delta(x-x_0)\delta(z-z_0) - 2\pi\delta(x-x_0)\delta(z) \tag{3.31}$$
$$- 2\pi\delta(x)\delta(z-z_0) + 2\pi\delta(x)\delta(z) \ .$$

There are $(L^x - 1)(L^z - 1)$ such identifications.

On a lattice with sites labelled as $(\hat{x}, \hat{y}, \hat{z})$, using the constraints (3.27), we can solve for $\bar{f}^{xz}(t, \hat{x} = 1, \hat{z})$, $\bar{f}^{xz}(t, \hat{x} \neq 1, \hat{z} = 1)$, and $\bar{f}^{yz}(t, \hat{y}, \hat{z} = 1)$ in terms of the other $\bar{f}^{ij}$, and then the remaining $L^x L^y + L^y L^z + L^x L^z - L^x - L^y - L^z + 1$ $\bar{f}$'s have periodicities $\bar{f} \sim \bar{f} + \frac{2\pi}{a^2}$.

The Lagrangian for these modes is

$$L = \frac{1}{g_e^2 \ell^x \ell^y \ell^z} \left[ \ell^x \ell^y \oint dx dy \ (\dot{\bar{f}}^{xy})^2 + \ell^y \ell^z \oint dy dz \ (\dot{\bar{f}}^{yz})^2 + \ell^x \ell^z \oint dx dz \ (\dot{\bar{f}}^{xz})^2 \right.$$
$$- \oint dx dy dz \ \left( \ell^x \dot{\bar{f}}^{xz} \dot{\bar{f}}^{xy} + \ell^y \dot{\bar{f}}^{xy} \dot{\bar{f}}^{yz} + \ell^z \dot{\bar{f}}^{yz} \dot{\bar{f}}^{xz} \right) \tag{3.32}$$
$$\left. + \left( \oint dx dy \ \dot{\bar{f}}^{xy} \right)^2 \right] + \frac{\theta}{2\pi} \oint dx dy \ \dot{\bar{f}}^{xy} \ .$$

Let $\bar{\Pi}^{ij}(x^i, x^j)$ be the conjugate momenta of $\bar{f}^{ij}$. The delta function periodicities of (3.29), (3.30), (3.31) imply that $\bar{\Pi}^{ij}(x^i, x^j)$ have independent integer eigenvalues at each $x^i$ and $x^j$. Due to the constraints (3.27), the momenta are subject to a gauge ambiguity:

$$\bar{\Pi}^{xy}(x, y) \sim \bar{\Pi}^{xy}(x, y) + n^x(x) - n^y(y) \ ,$$
$$\bar{\Pi}^{yz}(y, z) \sim \bar{\Pi}^{yz}(y, z) + n^y(y) - n^z(z) \ , \tag{3.33}$$
$$\bar{\Pi}^{xz}(x, z) \sim \bar{\Pi}^{xz}(x, z) + n^z(z) - n^x(x) \ ,$$

where $n^i(x^i) \in \mathbb{Z}$. Note that shifting all $n^i(x^i)$ by the same zero mode does not contribute to the above gauge symmetry. The charge of the electric tensor symmetry (3.18) can be expressed in terms of the conjugate momenta as

$$Q(x, y, z) = \frac{2}{g_e^2} E_{xyz} + \frac{\theta}{2\pi} = \bar{\Pi}^{xy}(x, y) + \bar{\Pi}^{yz}(y, z) + \bar{\Pi}^{xz}(x, z) \ . \tag{3.34}$$

The Hamiltonian is

$$H = \frac{g_e^2}{4} \left[ \ell^z \oint dx dy \ \left( \bar{\Pi}^{xy} - \frac{\theta^{xy}}{2\pi} \right)^2 + \ell^x \oint dy dz \ \left( \bar{\Pi}^{yz} - \frac{\theta^{yz}}{2\pi} \right)^2 + \ell^y \oint dx dz \ \left( \bar{\Pi}^{xz} - \frac{\theta^{xz}}{2\pi} \right)^2 \right.$$
$$+ 2 \oint dx dy dz \ \left( \bar{\Pi}^{xy} - \frac{\theta^{xy}}{2\pi} \right) \left( \bar{\Pi}^{yz} - \frac{\theta^{yz}}{2\pi} \right) + 2 \oint dx dy dz \ \left( \bar{\Pi}^{yz} - \frac{\theta^{yz}}{2\pi} \right) \left( \bar{\Pi}^{xz} - \frac{\theta^{xz}}{2\pi} \right)$$
$$\left. + 2 \oint dx dy dz \ \left( \bar{\Pi}^{xz} - \frac{\theta^{xz}}{2\pi} \right) \left( \bar{\Pi}^{xy} - \frac{\theta^{xy}}{2\pi} \right) \right] \ ,$$
$$\tag{3.35}$$

where $\theta^{xy} + \theta^{yz} + \theta^{xz} = \theta$. Here, $\theta^{ij}$ can depend on $x^i, x^j$ but not $x^k$. The Hamiltonian depends only on the sum of $\theta^{ij}$.

Consider regularizing the above Hamiltonian on a lattice with lattice spacing $a$, and sites labelled as $(\hat{x}, \hat{y}, \hat{z})$, where $\hat{x}^i = 1, \dots, L^i$. The conjugate momenta $\bar{\Pi}^{ij}(\hat{x}^i, \hat{x}^j)$ have integer eigenvalues at each $\hat{x}^i$ and $\hat{x}^j$. States with finitely many nonzero $\bar{\Pi}^{ij}(\hat{x}^i, \hat{x}^j)$ have small energies of the order of $a^2$ which vanish in the continuum limit. This is in contrast to the $\varphi$-theory where the classically zero energy states are lifted to infinite energies quantum

mechanically. There are states with order $L$ nonzero momenta $\bar{\Pi}^{ij}(\hat{x}^i, \hat{x}^j)$. For example, $\bar{\Pi}^{xy}(\hat{x}, \hat{y}) = [\Theta(\hat{x} - \hat{x}_1) - \Theta(\hat{x} - \hat{x}_2)] \delta_{\hat{y}, \hat{y}_0}$ with $\hat{x}_1 < \hat{x}_2$. The energies of such states are of order $a$, and they vanish in the continuum limit. Finally, there are also states with order $L^2$ nonzero $\bar{\Pi}^{ij}(\hat{x}^i, \hat{x}^j)$. For example, $\bar{\Pi}^{xy}(x, y) = [\Theta(x - x_1) - \Theta(x - x_2)] [\Theta(y - y_1) - \Theta(y - y_2)]$ with $x_1 < x_2$ and $y_1 < y_2$. The energies of such states are of order.

Similar to the discussions in [3–5], we now discuss the effect of higher derivative terms on the states in the gauge theory. For example, consider

$$g(\partial_x E_{xyz})^2 \tag{3.36}$$

with the coupling $g$ of order $a^2$. As we discussed above, the states with finitely many nonzero $\bar{\Pi}^{ij}$ have energy of order $a^2$. The higher derivative term shifts their energy by an amount of order $g \sim a^2$. The states with order $1/a$ nonzero momenta $\bar{\Pi}^{ij}$ have energy of order $a$ and their energy is shifted by an amount of order $g/a \sim a$. Therefore, the energy of these two types of states remain zero in the continuum theory. The energy of the states with order $1/a^2$ nonzero momenta $\bar{\Pi}^{ij}$ is of order one and their energy is shifted by an amount of order one. To conclude, while the zero-energy states are not lifted by these higher derivative terms, the finite energy states do receive quantitative corrections leaving the qualitative scaling with $a$ invariant.

There are no relevant operators that violate the electric tensor symmetry. Hence, we conclude that the electric tensor symmetry is robust in the $(3+1)$-dimensional tensor gauge theory.

# 4 $\mathbb{Z}_N$ Tensor Gauge Theory of $B$

In this section, we discuss a $\mathbb{Z}_N$ version of the tensor gauge theory. The theory can be obtained by coupling the $U(1)$ theory to a scalar field $\varphi$ with charge $N$ that Higgses it to $\mathbb{Z}_N$

## 4.1 Lagrangian

The Euclidean Lagrangian is

$$\mathcal{L}_E = \frac{i}{2\pi} \hat{E}^{xyz} (\partial_x \partial_y \partial_z \varphi - N B_{xyz}) + \frac{i}{2\pi} \hat{B} (\partial_\tau \varphi - N B_\tau) , \tag{4.1}$$

where $(B_\tau, B_{xyz})$ are the $U(1)$ tensor gauge fields and $\varphi$ is a $2\pi$-periodic scalar field that Higgses the $U(1)$ gauge symmetry to $\mathbb{Z}_N$. The gauge transformations are

$$
\begin{aligned}
\varphi &\sim \varphi + N\alpha \ , \\
B_\tau &\sim B_\tau + \partial_\tau \alpha \ , \\
B_{xyz} &\sim B_{xyz} + \partial_x \partial_y \partial_z \alpha \ .
\end{aligned}
\tag{4.2}
$$

The fields $\hat{E}^{xyz}$ and $\hat{B}$ are Lagrangian multipliers that constrain $(B_\tau, B_{xyz})$ to be $\mathbb{Z}_N$ gauge fields.

We dualize the Euclidean action by integrating out $\varphi$. This leads to the constraint

$$
\partial_x \partial_y \partial_z \hat{E}^{xyz} + \partial_\tau \hat{B} = 0 \ ,
\tag{4.3}
$$

which is solved locally in terms of a field $\varphi^{xyz}$ in the $\mathbf{1'}$ representation of $S_4$

$$
\hat{E}^{xyz} = \partial_\tau \varphi^{xyz} \ , \quad \hat{B} = -\partial_x \partial_y \partial_z \varphi^{xyz} \ .
\tag{4.4}
$$

The winding modes of $\varphi$ mean that the periods of $\hat{E}^{xyz}$ and of $\hat{B}$ are quantized, corresponding to $\varphi^{xyz} \sim \varphi^{xyz} + 2\pi$. The Euclidean action becomes

$$
\mathcal{L}_E = \frac{iN}{2\pi} \varphi^{xyz} (\partial_\tau B_{xyz} - \partial_x \partial_y \partial_z B_\tau) = \frac{iN}{2\pi} \varphi^{xyz} E_{xyz} \ .
\tag{4.5}
$$

## 4.2  Global Symmetry

The theory has a $\mathbb{Z}_N$ electric global symmetry generated by the gauge-invariant local operator

$$
e^{i\varphi^{xyz}} \ .
\tag{4.6}
$$

As we will discuss soon, these local operators can be added to the Lagrangian and destabilize the theory. There is also a $\mathbb{Z}_N$ magnetic global symmetry generated by the Wilson tubes

$$
\begin{aligned}
W_{(xy)}(x_1, x_2; y_1, y_2) &= \exp\left[ i \int_{x_1}^{x_2} dx \int_{y_1}^{y_2} dy \oint dz \ B_{xyz} \right] \ , \\
W_{(yz)}(y_1, y_2; z_1, z_2) &= \exp\left[ i \oint dx \int_{y_1}^{y_2} dy \int_{z_1}^{z_2} dz \ B_{xyz} \right] \ , \\
W_{(xz)}(x_1, x_2; z_1, z_2) &= \exp\left[ i \int_{x_1}^{x_2} dx \oint dy \int_{z_1}^{z_2} dz \ B_{xyz} \right] \ .
\end{aligned}
\tag{4.7}
$$

This magnetic symmetry is the $\mathbb{Z}_N$ version of the quadruple symmetry discussed in Section 2.3.

The exponents of these symmetry operators are quantized to be integers and

$$e^{iN\varphi^{xyz}} = W_{(ij)}^N = 1 \ . \tag{4.8}$$

Therefore they are $\mathbb{Z}_N$ operators. The operators (4.6) and (4.7) do not commute if they intersect:

$$
\begin{aligned}
e^{i\varphi^{xyz}(x,y,z)} W_{(xy)}(x_1, x_2; y_1, y_2) &= e^{2\pi i/N} W_{(xy)}(x_1, x_2; y_1, y_2) e^{i\varphi^{xyz}(x,y,z)} \ , \\
e^{i\varphi^{xyz}(x,y,z)} W_{(yz)}(y_1, y_2; z_1, z_2) &= e^{2\pi i/N} W_{(yz)}(y_1, y_2; z_1, z_2) e^{i\varphi^{xyz}(x,y,z)} \ , \\
e^{i\varphi^{xyz}(x,y,z)} W_{(xz)}(x_1, x_2; z_1, z_2) &= e^{2\pi i/N} W_{(xz)}(x_1, x_2; z_1, z_2) e^{i\varphi^{xyz}(x,y,z)} \ .
\end{aligned}
\tag{4.9}
$$

As we will see, the spectrum is in a minimal representation of this algebra.

## 4.3 Defects as Fractons

The defects of the $\mathbb{Z}_N$ tensor gauge theory are similar to those of the $U(1)$ tensor gauge theory studied in Section 3.5. The simplest defect is a single static charged particle

$$\exp\left[i \int_{-\infty}^{\infty} dt \ B_0\right] \ . \tag{4.10}$$

A single particle cannot move in space because of the gauge symmetry. However, four of them that form a quadrupole can move collectively. Their motion is described by the gauge-invariant defect

$$\exp\left[i \int_{x_1}^{x_2} dx \int_{y_1}^{y_2} dy \int_{\mathcal{C}} (dt \ \partial_x \partial_y B_0 + dz \ B_{xyz})\right] \ , \tag{4.11}$$

where $\mathcal{C}$ is a curve in the $(z, t)$ plane. Similarly, we can have quadrupole moving collectively in the $x$ and $y$ directions. In the special case where $\mathcal{C}$ is fixed in time, these operators are the generators of the $\mathbb{Z}_N$ quadrupole symmetry (4.7).

## 4.4 Cube Ising Model and a Lattice Tensor Gauge Theory

The $\mathbb{Z}_N$ tensor gauge theory arises as the continuum limits of the two different lattice theories, the $\mathbb{Z}_N$ lattice tensor gauge theory and the $\mathbb{Z}_N$ cube Ising model. In this sense, the two lattice modes are dual to each other at long distance.

### 4.4.1 Cube Ising Model

The $\mathbb{Z}_N$ cube Ising model is the $\mathbb{Z}_N$ version of the XY-cube model. There is a $\mathbb{Z}_N$ phase $U_s$ and its conjugate momentum $V_s$ at each site. They obey the commutation relation $U_s V_s = e^{2\pi i/N} V_s U_s$. The Hamiltonian includes the cube interaction and a transverse field term:

$$
\begin{aligned}
H = - & h \sum_s (V_s + c.c.) \\
- & K \sum_{\hat{x},\hat{y},\hat{z}} (U_{\hat{x},\hat{y},\hat{z}} U^{-1}_{\hat{x}+1,\hat{y},\hat{z}} U^{-1}_{\hat{x},\hat{y}+1,\hat{z}} U^{-1}_{\hat{x},\hat{y},\hat{z}+1} U_{\hat{x}+1,\hat{y}+1,\hat{z}} U_{\hat{x}+1,\hat{y},\hat{z}+1} U_{\hat{x},\hat{y}+1,\hat{z}+1} U^{-1}_{\hat{x}+1,\hat{y}+1,\hat{z}+1} + c.c.) \, .
\end{aligned}
$$
(4.12)

We will assume $h$ to be small. The classical spin system with interaction around a cube has appeared in [18].

The lattice theory has conserved charge operators:

$$
W_{(xy)}(\hat{x}, \hat{y}) = \prod_{\hat{z}=1}^{L^z} V_{\hat{x},\hat{y},\hat{z}} \, ,
$$
(4.13)

and similar operators along $y$ and $z$ direction. In the continuum, they become the quadrupole global symmetry operator (4.7). The $\mathbb{Z}_N$ electric tensor symmetry of the continuum theory is broken in the lattice theory.

If we only impose the $\mathbb{Z}_N$ quadrupole symmetry on the lattice, there is no symmetry-preserving relevant operator at long distances that violates the emergent $\mathbb{Z}_N$ electric tensor symmetry. Hence the emergent $\mathbb{Z}_N$ electric tensor symmetry is robust.

### 4.4.2 Lattice Tensor Gauge Theory

The $\mathbb{Z}_N$ lattice tensor gauge theory has a $\mathbb{Z}_N$ phase variable $U_c$ and its conjugate variable $V_c$ on every cube $c$. They obey $U_c V_c = e^{2\pi i/N} V_c U_c$. The gauge transformation $\eta_s$ is a $\mathbb{Z}_N$ phase associated with each site. Under the gauge transformation,

$$
U_c \sim U_c \eta_{\hat{x},\hat{y},\hat{z}} \eta^{-1}_{\hat{x}+1,\hat{y},\hat{z}} \eta^{-1}_{\hat{x},\hat{y}+1,\hat{z}} \eta^{-1}_{\hat{x},\hat{y},\hat{z}+1} \eta_{\hat{x}+1,\hat{y}+1,\hat{z}} \eta_{\hat{x}+1,\hat{y},\hat{z}+1} \eta_{\hat{x},\hat{y}+1,\hat{z}+1} \eta^{-1}_{\hat{x}+1,\hat{y}+1,\hat{z}+1} \, ,
$$
(4.14)

where the product is over the eight sites around the cube $c$.

Gauss law sets

$$
G_s \equiv \prod_{c \ni s} (V_c)^{\epsilon_c} = 1
$$
(4.15)

where the product is an oriented product ($\epsilon_c = \pm 1$) over the eight cubes $c$ that share a

common site $s$. The Hamiltonian is

$$H = -\widetilde{h} \sum_c (V_c + c.c.) \,, \tag{4.16}$$

and we impose Gauss law as an operator equation. Alternatively, we can impose Gauss law energetically by adding a term to the Hamiltonian

$$H = -K \sum_s (G_s + c.c) - \widetilde{h} \sum_c (V_c + c.c.) \,. \tag{4.17}$$

The lattice theory has conserved charges $V_c$ at each cube. They become the $\mathbb{Z}_N$ electric tensor symmetry generateor (4.6) in the continuum. The $\mathbb{Z}_N$ quadrupole symmetry of the continuum theory is broken in the lattice theory.

When $h$ and $\widetilde{h}$ are both zero, the Hamiltonian (4.17) becomes the Hamiltonian (4.12) of the cube Ising model if we dualize the lattice and identify $U_c \leftrightarrow V_s^{-1}$, $V_c \leftrightarrow U_s^{-1}$. At long distances, they both flow to the $\mathbb{Z}_N$ tensor gauge theory (4.5).

If we only impose the $\mathbb{Z}_N$ electric tensor symmetry on the lattice, one can break the emergent $\mathbb{Z}_N$ quadrupole symmetry by adding operators such as $e^{i\varphi^{xyz}}$ to the Lagrangian. Hence, the emergent $\mathbb{Z}_N$ quadrupole symmetry is not robust.

## 4.5   Ground State Degeneracy

The ground state degeneracy of this system on a lattice is $N^{L^xL^y+L^yL^z+L^xL^z-L^x-L^y-L^z+1}$. One way to see this is to study the ground states of the Lagrangian (4.1). Using the equations of motion (4.2), we can solve for all the fields in terms of $\varphi$, so the solution space is

$$\{\varphi \mid \varphi \sim \varphi + N\alpha\} \,. \tag{4.18}$$

This identification removes almost all configurations of $\varphi$ except for the winding modes (2.31)

$$\varphi(t,x,y,z) = 2\pi \left[ W \frac{xyz}{\ell^x \ell^y \ell^z} - \frac{yz}{\ell^y \ell^z} \sum_\alpha W_\alpha^x \Theta_\alpha(x) - \frac{xz}{\ell^x \ell^z} \sum_\beta W_\beta^y \Theta_\beta(y) - \frac{xy}{\ell^x \ell^y} \sum_\gamma W_\gamma^z \Theta_\gamma(z) \right.$$

$$\left. + \frac{x}{\ell^x} \sum_{\beta\gamma} W_{\beta\gamma}^{yz} \Theta_\beta(y)\Theta_\gamma(z) + \frac{y}{\ell^y} \sum_{\alpha\gamma} W_{\alpha\gamma}^{xz} \Theta_\alpha(x)\Theta_\gamma(z) + \frac{z}{\ell^z} \sum_{\alpha\beta} W_{\alpha\beta}^{xy} \Theta_\alpha(x)\Theta_\beta(y) \right] \,,$$

$$\tag{4.19}$$

where $\Theta_\alpha(x) = \Theta(x - x_\alpha)$, and all the coefficients are integers and they are related by

$$
\begin{aligned}
W_\alpha^x &= \sum_\beta W_{\alpha\beta}^{xy} = \sum_\gamma W_{\alpha\gamma}^{xz} \ , \\
W_\beta^y &= \sum_\alpha W_{\alpha\beta}^{xy} = \sum_\gamma W_{\beta\gamma}^{yz} \ , \\
W_\gamma^z &= \sum_\beta W_{\beta\gamma}^{yz} = \sum_\alpha W_{\alpha\gamma}^{xz} \ , \\
W &= \sum_\alpha W_\alpha^x = \sum_\beta W_\beta^y = \sum_\gamma W_\gamma^z \ .
\end{aligned}
\tag{4.20}
$$

If we regularize the theory on a lattice, the above winding modes are labelled by $L^x L^y + L^y L^z + L^x L^z - L^x - L^y - L^z + 1$ integers. The gauge parameter $\alpha$ also has similar winding modes, so the ones that cannot be gauged away have winding charges $W_{\alpha\beta}^{xy}$, $W_{\beta\gamma}^{yz}$ and $W_{\alpha\gamma}^{xz}$ that are valued in $\mathbb{Z}_N$, i.e., only charges modulo $N$ are physical. This leads to $N^{L^x L^y + L^y L^z + L^x L^z - L^x - L^y - L^z + 1}$ ground states as advertised above.

Another way to see the ground state degeneracy is from the $\mathbb{Z}_N$ global symmetries. On a lattice, the set of commutation relations between $\mathbb{Z}_N$ quadrupole and electric global symmetries (4.9) is isomorphic to $L^x L^y + L^y L^z + L^x L^z - L^x - L^y - L^z + 1$ copies of $\mathbb{Z}_N$ Heisenberg algebra: $AB = e^{2\pi i/N} BA$ and $A^N = B^N = 1$. The isomorphism is given by

$$
\begin{aligned}
A_{\hat{x},\hat{y}} &= e^{i\varphi^{xyz}(\hat{x},\hat{y},1)} \ , & B_{\hat{x},\hat{y}} &= W_{(xy)}(\hat{x},\hat{y}) \ , \\
A_{\hat{y},\hat{z}} &= e^{i\varphi^{xyz}(1,\hat{y},\hat{z})-i\varphi^{xyz}(1,\hat{y},1)} \ , & B_{\hat{y},\hat{z}} &= W_{(yz)}(\hat{y},\hat{z}) \ , \\
A_{\hat{x},\hat{z}} &= e^{i\varphi^{xyz}(\hat{x},1,\hat{z})-i\varphi^{xyz}(\hat{x},1,1)-i\varphi^{xyz}(1,1,\hat{z})+i\varphi^{xyz}(1,1,1)} \ , & B_{\hat{x},\hat{z}} &= W_{(xz)}(\hat{x},\hat{z}) \ ,
\end{aligned}
\tag{4.21}
$$

where $W_{(xy)}(\hat{x},\hat{y}) = \exp\left[ia^3 \sum_{\hat{z}=1}^{L^z} B_{xyz}(\hat{x},\hat{y},\hat{z})\right]$ is a tube operator along the $z$ direction with cross section area $a^2$, and similarly for other directions. The minimal representation of the $\mathbb{Z}_N$ Heisenberg algebra is $N$-dimensional. Therefore, the nontrivial algebra (4.9) forces the ground state degeneracy to be $N^{L^x L^y + L^y L^z + L^x L^z - L^x - L^y - L^z + 1}$.

# 5 $U(1)$ Tensor Gauge Theory of $C$

Consider a $(\mathbf{R}_{\text{time}}, \mathbf{R}_{\text{space}}) = (\mathbf{3'}, \mathbf{2})$ tensor global symmetry generated by currents $(J_0^{ij}, J^{[ij]k})$ [4]. The currents obey the conservation equation:

$$
\partial_0 J_0^{ij} = \partial_k (J^{[ki]j} + J^{[kj]i}) \ ,
\tag{5.1}
$$

and a differential condition

$$
\partial_i \partial_j J_0^{ij} = 0 \ .
\tag{5.2}
$$

We will study the pure gauge theory without matter obtained by gauging this $(\mathbf{3'}, \mathbf{2})$ tensor global symmetry.

We can gauge the $(\mathbf{3'}, \mathbf{2})$ tensor global symmetry by coupling the currents to the tensor gauge field $(C_0^{ij}, C^{[ij]k})$:

$$\frac{1}{2} J_0^{ij} C_0^{ij} + \frac{1}{2} J^{[ij]k} C^{[ij]k} \ . \tag{5.3}$$

The current conservation equation (5.1) and the differential condition (5.2) imply the gauge transformation

$$\begin{aligned} C_0^{ij} &\sim C_0^{ij} + \partial_0 \alpha^{ij} - \partial^i \partial^j \alpha_0 \\ C^{[ij]k} &\sim C^{[ij]k} - \partial^i \alpha^{jk} + \partial^j \alpha^{ik} \ . \end{aligned} \tag{5.4}$$

The gauge parameters $(\alpha_0, \alpha^{ij})$ are also gauge fields themselves in the $(\mathbf{1}, \mathbf{3'})$ representation of $S_4$. They have their own gauge transformation

$$\alpha_0 \sim \alpha_0 + \partial_0 \gamma \ , \qquad \alpha^{ij} \sim \alpha^{ij} + \partial^i \partial^j \gamma \ , \tag{5.5}$$

where $\gamma$ is in the $\mathbf{1}$ of $S_4$. The gauge transformation (5.5) of $(\alpha_0, \alpha^{ij})$ does not affect the gauge transformation (5.4) of $(C_0^{ij}, C^{[ij]k})$. The gauge-invariant electric field is

$$E^{[ij]k} = \partial_0 C^{[ij]k} + \partial^i C_0^{jk} - \partial^j C_0^{ik} \ , \tag{5.6}$$

which is in $\mathbf{2}$ of $S_4$, while there is no magnetic field.

It is interesting that the gauge parameters $(\alpha_0, \alpha^{ij})$ are similar to the gauge fields $(A_0, A_{ij})$ of the $A$-theory of [4], with their corresponding gauge symmetry (5.5). The $(C_0^{ij}, C^{[ij]k})$ gauge fields are similar to the field strengths $(E_{ij}, B_{[ij]k})$ of $(A_0, A_{ij})$. Furthermore, the electric field $E^{[ij]k}$ of $(C_0^{ij}, C^{[ij]k})$ is similar to the Bianchi identity of the $A$-theory. Physically, it means that the gauge fields $(A_0, A_{ij})$ are the Higgs fields of $(C_0^{ij}, C^{[ij]k})$ (see Section 6.2). More generally, this analogy is the same as the relation between higher form gauge theories of different degrees, where the gauge fields of one theory are similar to the gauge parameters for the higher form gauge theory.

## 5.1  The Lattice Model

In this subsection, we will discuss the $U(1)$ lattice tensor gauge theory of $C$. We will present both the Lagrangian and Hamiltonian presentations of this lattice model.

We will consider a Euclidean lattice with lattice spacing $a$. The gauge parameters are placed on the temporal links $\eta_\tau$ (in $\mathbf{1}$ of $S_4$) and on the spatial plaquettes $\eta^{xy}$, $\eta^{yz}$ and $\eta^{xz}$ (in $\mathbf{3'}$ of $S_4$). We also write $\eta_\tau = e^{ia\alpha_\tau}$, and $\eta^{ij} = e^{ia^2\alpha^{ij}}$. The gauge parameters of the gauge parameters are placed on the sites $\lambda = e^{i\gamma}$ (in $\mathbf{1}$ of $S_4$).

The gauge fields are placed on the cubes. On each temporal cube along $\tau xy$, there

is a gauge field $U_\tau^{xy} = e^{ia^3 C_\tau^{xy}}$ (in $\mathbf{3'}$ of $S_4$), and similarly along $\tau yz$ and $\tau zx$. On each spatial cube, there are three gauge fields $U^{[ij]k} = e^{ia^3 C^{[ij]k}}$ (in $\mathbf{2}$ of $S_4$), which satisfy $U^{[xy]z} U^{[yz]x} U^{[zx]y} = 1$.

The gauge transformations act on the gauge fields as

$$
\begin{aligned}
U_\tau^{xy}(\hat{\tau}, \hat{x}, \hat{y}, \hat{z}) &\sim U_\tau^{xy}(\hat{\tau}, \hat{x}, \hat{y}, \hat{z}) \eta^{xy}(\hat{\tau}, \hat{x}, \hat{y}, \hat{z})^{-1} \eta^{xy}(\hat{\tau}+1, \hat{x}, \hat{y}, \hat{z}) \\
&\quad \times \eta_\tau(\hat{\tau}, \hat{x}, \hat{y}, \hat{z})^{-1} \eta_\tau(\hat{\tau}, \hat{x}+1, \hat{y}+1, \hat{z})^{-1} \eta_\tau(\hat{\tau}, \hat{x}+1, \hat{y}, \hat{z}) \eta_\tau(\hat{\tau}, \hat{x}, \hat{y}+1, \hat{z}) , \\
U^{[xy]z}(\hat{\tau}, \hat{x}, \hat{y}, \hat{z}) &\sim U^{[xy]z}(\hat{\tau}, \hat{x}, \hat{y}, \hat{z}) \eta^{xz}(\hat{\tau}, \hat{x}, \hat{y}+1, \hat{z}) \eta^{xz}(\hat{\tau}, \hat{x}, \hat{y}, \hat{z})^{-1} \\
&\quad \times \eta^{yz}(\hat{\tau}, \hat{x}+1, \hat{y}, \hat{z})^{-1} \eta^{yz}(\hat{\tau}, \hat{x}, \hat{y}, \hat{z}) ,
\end{aligned}
\tag{5.7}
$$

and similarly for $U_\tau^{yz}$, $U_\tau^{zx}$, $U^{[yz]x}$ and $U^{[zx]y}$. The gauge transformations themselves have gauge transformations given by

$$
\begin{aligned}
\eta_\tau(\hat{\tau}, \hat{x}, \hat{y}, \hat{z}) &\sim \eta_\tau(\hat{\tau}, \hat{x}, \hat{y}, \hat{z}) \lambda(\hat{\tau}+1, \hat{x}, \hat{y}, \hat{z}) \lambda(\hat{\tau}, \hat{x}, \hat{y}, \hat{z})^{-1} , \\
\eta^{xy}(\hat{\tau}, \hat{x}, \hat{y}, \hat{z}) &\sim \eta^{xy}(\hat{\tau}, \hat{x}, \hat{y}, \hat{z}) \lambda(\hat{\tau}, \hat{x}, \hat{y}, \hat{z}) \lambda(\hat{\tau}, \hat{x}+1, \hat{y}+1, \hat{z}) \\
&\quad \times \lambda(\hat{\tau}, \hat{x}+1, \hat{y}, \hat{z})^{-1} \lambda(\hat{\tau}, \hat{x}, \hat{y}+1, \hat{z})^{-1} ,
\end{aligned}
\tag{5.8}
$$

and similarly for $\eta^{yz}$ and $\eta^{zx}$.

Let us discuss the gauge-invariant local terms in the action. There are three kinds of terms on each spacetime hyper-cube:

$$
\begin{aligned}
L^{[xy]z}(\hat{\tau}, \hat{x}, \hat{y}, \hat{z}) &= U_\tau^{xz}(\hat{\tau}, \hat{x}, \hat{y}+1, \hat{z})^{-1} U_\tau^{xz}(\hat{\tau}, \hat{x}, \hat{y}, \hat{z}) U_\tau^{yz}(\hat{\tau}, \hat{x}+1, \hat{y}, \hat{z}) U_\tau^{yz}(\hat{\tau}, \hat{x}, \hat{y}, \hat{z})^{-1} \\
&\quad \times U^{[xy]z}(\hat{\tau}, \hat{x}, \hat{y}, \hat{z})^{-1} U^{[xy]z}(\hat{\tau}+1, \hat{x}, \hat{y}, \hat{z}) ,
\end{aligned}
\tag{5.9}
$$

and similarly $L^{[yz]x}$ and $L^{[zx]y}$. These terms together with their complex conjugate become the squares of the electric fields in the continuum limit.

In addition to the local, gauge invariant operators above, there are non-local, extended ones. For example, we have a *slab operator* along the $xy$ plane:

$$
\prod_{\hat{x}=1}^{L^x} \prod_{\hat{y}=1}^{L^y} U^{[xy]z}(\hat{x}, \hat{y}, \hat{z}_0) .
\tag{5.10}
$$

Similarly, we have slab operators along $yz$ and $zx$ planes.

In the Hamiltonian formulation, we choose the temporal gauge to set all $U_\tau^{ij} = 1$. On each spatial cube, we introduce the electric field $E^{[ij]k}$ such that $\frac{2}{g_e^2} E^{[ij]k}$ is conjugate to the phase variable $U^{[ij]k}$ with $g_e$ the electric coupling constant. This definition of the lattice electric field differs from the continuum definition by a power of the lattice spacing, which can be easily restored by dimension analysis.

Since $U^{[xy]z}U^{[yz]x}U^{[zx]y} = 1$, the electric fields have a gauge ambiguity $E^{[xy]z} \sim E^{[xy]z} + c$, $E^{[yz]x} \sim E^{[yz]x} + c$, $E^{[zx]y} \sim E^{[zx]y} + c$, at every given cube. We also define $E^{k(ij)} = E^{[ki]j} + E^{[kj]i}$, which satisfy $E^{x(yz)} + E^{y(zx)} + E^{z(xy)} = 0$ and do not have any gauge ambiguity. Note that both $E^{[ij]k}$ and $E^{i(jk)}$ have the same number of degrees of freedom.

On every spatial plaquette in the $xy$ direction, we impose the Gauss law

$$G^{xy}(\hat{x}, \hat{y}, \hat{z}) = E^{[zx]y}(\hat{x}, \hat{y}, \hat{z} + 1) - E^{[zx]y}(\hat{x}, \hat{y}, \hat{z}) - E^{[yz]x}(\hat{x}, \hat{y}, \hat{z} + 1) + E^{[yz]x}(\hat{x}, \hat{y}, \hat{z})$$
$$= E^{z(xy)}(\hat{x}, \hat{y}, \hat{z} + 1) - E^{z(xy)}(\hat{x}, \hat{y}, \hat{z}) = 0 ,$$
(5.11)

and similarly in the $yz$ and $xz$ directions. So there are three Gauss laws $G^{ij} = 0$.

The Hamiltonian is the sum of $(E^{i(jk)})^2$ over all the cubes, with the three Gauss laws imposed as operator equations. Alternatively, we can impose the Gauss laws energetically by adding a term $\sum_{\text{plaquettes}}(G^{ij})^2$ to the Hamiltonian.

The lattice model has an *electric tensor symmetry* whose conserved charges are proportional to

$$E^{i(jk)}(\hat{x}_0, \hat{y}_0, \hat{z}_0) .$$
(5.12)

They trivially commute with the Hamiltonian. The electric tensor symmetry generated by $E^{x(yz)}$ rotates the phase of $U^{[ij]k}$ at a single spatial cube:

$$U^{[xy]z} \to e^{i\alpha}U^{[xy]z} , \quad U^{[yz]x} \to U^{[yz]x} , \quad U^{[zx]y} \to e^{-i\alpha}U^{[zx]y} .$$
(5.13)

Using Gauss law (5.11), the conserved charge $E^{x(yz)}$ is independent of $\hat{x}$. Its $\hat{y}$ and $\hat{z}$ dependence is further restricted by the constraint $E^{x(yz)} + E^{y(zx)} + E^{z(xy)} = 0$ (see Section 5.4).

## 5.2   Continuum Lagrangian

The Lorentzian Lagrangian of the pure tensor gauge theory is

$$\mathcal{L} = \frac{1}{2g_e^2}E_{[ij]k}E^{[ij]k} = \frac{1}{2g_e^2}E_{i(jk)}E^{i(jk)} ,$$
(5.14)

where $E^{i(jk)} = 3E_{i(jk)}$ (see Appendix A). The equations of motion are

$$\frac{2}{g_e^2}\partial_0 E^{k(ij)} = 0 , \qquad \partial_k E^{k(ij)} = 0 .$$
(5.15)

where the second equation is the Gauss law.

In many ways, this theory is similar to the $U(1)$ gauge theory in $2 + 1$ dimensions in [3] and the gauge theory of Section 3. They have only electric field, but no magnetic field, they

have quantized fluxes (see below), and no local excitations (see below). But unlike these theories, here the electric field is not invariant under the cubic group, and therefore we do not add a $\theta$-parameter.

## 5.3  Fluxes

Let us put the theory on a Euclidean 4-torus of lengths $\ell^\tau$, $\ell^x$, $\ell^y$, $\ell^z$. Consider the gauge field configurations with nontrivial a transition function at time $\tau = \ell^\tau$:[7]

$$
\begin{aligned}
g^{yz}_{(\tau)}(x,y,z) &= 2\pi \frac{x}{\ell^x}\left[\frac{1}{\ell^z}\delta(y - y_0) + \frac{1}{\ell^y}\delta(z - z_0) - \frac{1}{\ell^y\ell^z}\right] \,, \\
g^{xz}_{(\tau)}(x,y,z) &= \frac{\pi}{\ell^x\ell^z}\left[\Theta(y - y_0) - \frac{y}{\ell^y}\right] \,, \\
g^{xy}_{(\tau)}(x,y,z) &= \frac{\pi}{\ell^x\ell^y}\left[\Theta(z - z_0) - \frac{z}{\ell^z}\right] \,,
\end{aligned}
\tag{5.16}
$$

which has its own transition function at $x = \ell^x$,

$$
h_{(x)}(y,z) = 2\pi\left[\frac{z}{\ell^z}\Theta(y - y_0) + \frac{y}{\ell^y}\Theta(z - z_0) - \frac{yz}{\ell^y\ell^z}\right] \,.
\tag{5.17}
$$

We have $C^{[ij]k}(\tau + \ell^\tau, x, y, z) = C^{[ij]k}(\tau, x, y, z) + \partial_j g^{ki}_{(\tau)}(x, y, z) - \partial_i g^{jk}_{(\tau)}(x, y, z)$. We also have $g^{ij}_{(\tau)}(x + \ell^x, y, z) = g^{ij}_{(\tau)}(x, y, z) + \partial_i\partial_j h_{(x)}(y, z)$.

A gauge field configuration with these transition functions is

$$
\begin{aligned}
C^{[xy]z} &= -\frac{2\pi\tau}{\ell^\tau\ell^x}\left[\frac{1}{\ell^y}\delta(z - z_0) + \frac{1}{2\ell^z}\delta(y - y_0) - \frac{1}{2\ell^y\ell^z}\right] \,, \\
C^{[zx]y} &= \frac{2\pi\tau}{\ell^\tau\ell^x}\left[\frac{1}{\ell^z}\delta(y - y_0) + \frac{1}{2\ell^y}\delta(z - z_0) - \frac{1}{2\ell^y\ell^z}\right] \,, \\
C^{[yz]x} &= \frac{2\pi\tau}{\ell^\tau\ell^x}\left[\frac{1}{2\ell^y}\delta(z - z_0) - \frac{1}{2\ell^z}\delta(y - y_0)\right] \,,
\end{aligned}
\tag{5.18}
$$

---

[7]As in all our theories, we allow certain singular configurations provided the terms in the Lagrangian are not too singular (see [3–5]). Here we follow the same rules as in [4]. It would be nice to understand better the precise rules controlling these singularities.

with the electric fields

$$E^{[xy]z} = -\frac{2\pi}{\ell^\tau \ell^x} \left[ \frac{1}{\ell^y} \delta(z - z_0) + \frac{1}{2\ell^z} \delta(y - y_0) - \frac{1}{2\ell^y \ell^z} \right] ,$$

$$E^{[zx]y} = \frac{2\pi}{\ell^\tau \ell^x} \left[ \frac{1}{\ell^z} \delta(y - y_0) + \frac{1}{2\ell^y} \delta(z - z_0) - \frac{1}{2\ell^y \ell^z} \right] , \qquad (5.19)$$

$$E^{[yz]x} = \frac{2\pi}{\ell^\tau \ell^x} \left[ \frac{1}{2\ell^y} \delta(z - z_0) - \frac{1}{2\ell^z} \delta(y - y_0) \right] ,$$

Since the electric field $E^{[ij]k}$ has mass dimension 4, it is allowed to have delta function singularities following the rules in [3, 4]. We have

$$\oint d\tau \oint dx \oint dy \; E^{[xy]z} = -2\pi\delta(z - z_0) ,$$

$$\oint d\tau \oint dx \oint dz \; E^{[zx]y} = 2\pi\delta(y - y_0) . \qquad (5.20)$$

$$\oint d\tau \oint dy \oint dz \; E^{[yz]x} = 0 .$$

By taking linear combinations of similar bundles, we can realize a general electric flux

$$e^{[xy]z}(z_1, z_2) = \oint d\tau \oint dx \oint dy \int_{z_1}^{z_2} dz \; E^{[xy]z} \in 2\pi\mathbb{Z} , \qquad (5.21)$$

and similarly $e^{[zx]y}(y_1, y_2)$ and $e^{[yz]x}(x_1, x_2)$. In particular, the fluxes can be nontrivial when integrated over all spacetime, but they satisfy

$$e^{[xy]z}(0, \ell^z) + e^{[zx]y}(0, \ell^y) + e^{[yz]x}(0, \ell^x) = 0 . \qquad (5.22)$$

On the lattice, these nontrivial fluxes correspond to the products

$$\prod_{\hat\tau, \hat x, \hat y} L^{[xy]z} = 1 , \quad \prod_{\hat\tau, \hat y, \hat z} L^{[yz]x} = 1 , \quad \prod_{\hat\tau, \hat x, \hat z} L^{[zx]y} = 1 . \qquad (5.23)$$

## 5.4   Global Symmetries and Their Charges

We now discuss the global symmetries of the $C$ theory.

The equation of motion (5.15) is identified as the current conservation equation

$$\partial_0 J_0^{i(jk)} = 0 , \qquad (5.24)$$

with current
$$J_0^{i(jk)} = \frac{2}{g_e^2} E^{i(jk)} . \tag{5.25}$$

The second equation of (5.15) is an additional differential equation imposed on $J_0$:
$$\partial_k J_0^{k(ij)} = 0 . \tag{5.26}$$

We will refer to this current as *electric tensor symmetry*. This is the continuum version of the lattice symmetry (5.12). Note that there is no spatial component of the current.

The charges are
$$Q^{i(jk)}(x, y, z) = J_0^{i(jk)}(x, y, z) , \tag{5.27}$$

at every point in space. The differential condition (5.26) means that the charges satisfy
$$\begin{aligned}
Q^{z(xy)}(x, y, z) &= Q^{xy}(x, y) , \\
Q^{x(yz)}(x, y, z) &= Q^{yz}(y, z) , \\
Q^{y(zx)}(x, y, z) &= Q^{zx}(x, z) ,
\end{aligned} \tag{5.28}$$

where $Q^{ij}(x^i, x^j) \in \mathbb{Z}$. Since the charges satisfy the constraint $Q^{z(xy)} + Q^{x(yz)} + Q^{y(zx)} = 0$, we can write them as
$$\begin{aligned}
Q^{z(xy)}(x, y, z) &= Q^y(y) - Q^x(x) , \\
Q^{x(yz)}(x, y, z) &= Q^z(z) - Q^y(y) , \\
Q^{y(zx)}(x, y, z) &= Q^x(x) - Q^z(z) ,
\end{aligned} \tag{5.29}$$

where $Q^i(x^i) \in \mathbb{Z}$. Note that shifting $Q^i(x^i) \to Q^i(x^i) + n$ by any $n \in \mathbb{Z}$ does not change the charges. So, on a lattice, there are $L^x + L^y + L^z - 1$ charges.

The symmetry operators are
$$\mathcal{U}^{i(jk)}(\beta; x, y, z) = \exp\left[i\frac{2\beta}{g_e^2} E^{i(jk)}(x, y, z)\right] . \tag{5.30}$$

The electric tensor symmetry acts on the gauge fields as
$$\begin{aligned}
C^{[xy]z}(x, y, z) &\to C^{[xy]z}(x, y, z) + c^z(z) - \frac{1}{2}c^x(x) - \frac{1}{2}c^y(y) , \\
C^{[yz]x}(x, y, z) &\to C^{[yz]x}(x, y, z) + c^x(x) - \frac{1}{2}c^y(y) - \frac{1}{2}c^z(z) , \\
C^{[zx]y}(x, y, z) &\to C^{[zx]y}(x, y, z) + c^y(y) - \frac{1}{2}c^z(z) - \frac{1}{2}c^x(x) ,
\end{aligned} \tag{5.31}$$

up to time-independent gauge transformations.

The operators charged under this electric tensor symmetry are *slab operators*:

$$W_x(x_1, x_2) = \exp\left( i \int_{x_1}^{x_2} dx \oint dy \oint dz \; C^{[yz]x} \right) ,$$

$$W_y(y_1, y_2) = \exp\left( i \oint dx \int_{y_1}^{y_2} dy \oint dz \; C^{[zx]y} \right) , \qquad (5.32)$$

$$W_z(z_1, z_2) = \exp\left( i \oint dx \oint dy \int_{z_1}^{z_2} dz \; C^{[xy]z} \right) .$$

These correspond to the slab operators (5.10) on the lattice. The symmetry operator (5.30) and charged operators (5.32) satisfy the commutation relation

$$\mathcal{U}^{z(xy)}(\beta; x, y, z)W_x(x_1, x_2) = e^{-i\beta}W_x(x_1, x_2)\mathcal{U}^{z(xy)}(\beta; x, y, z) , \quad \text{if} \quad x_1 < x < x_2 ,$$

$$\mathcal{U}^{z(xy)}(\beta; x, y, z)W_y(y_1, y_2) = e^{i\beta}W_y(y_1, y_2)\mathcal{U}^{z(xy)}(\beta; x, y, z) , \quad \text{if} \quad y_1 < y < y_2 , \qquad (5.33)$$

$$\mathcal{U}^{z(xy)}(\beta; x, y, z)W_z(z_1, z_2) = W_z(z_1, z_2)\mathcal{U}^{z(xy)}(\beta; x, y, z) , \quad \text{if} \quad z_1 < z < z_2 ,$$

and they commute otherwise. The commutation relations for $\mathcal{U}^{y(zx)}$ and $\mathcal{U}^{x(yz)}$ are similar.

Only integer powers of $W_i$ are invariant under the large gauge transformations such as (5.16). It then follows that $\beta$ is $2\pi$-periodic. Therefore, the global structure of the electric tensor symmetry is $U(1)$, rather than $\mathbb{R}$.

## 5.5 Defects as Fractonic Strips

There are no particles in the model, however, there are *strips* fixed between two parallel planes, whose *fibers* are oriented normal to the plane and cannot bend. For example, a charge $+1$, static strip extended along the $y$ direction with fibers along the $z$ direction (fixed between two $xy$ planes at $z = z_1$ and $z = z_2$) is described by the defect[8]

$$\tilde{W}_z(z_1, z_2) = \exp\left[ i \int_{-\infty}^{\infty} dt \oint dy \int_{z_1}^{z_2} dz \; C_0^{yz} \right] . \qquad (5.34)$$

More generally, we can have a charge $+1$ static closed strip along a closed loop $\mathcal{C}^{xy}$ in the $xy$ plane with fibers along the $z$ direction described by the defect

$$\tilde{W}_z(z_1, z_2, \mathcal{C}^{xy}) = \exp\left[ i \int_{-\infty}^{\infty} dt \int_{z_1}^{z_2} dz \oint_{\mathcal{C}^{xy}} (C_0^{zx}dx + C_0^{yz}dy) \right] . \qquad (5.35)$$

---

[8]In the Euclidean version of this defect, the charge is quantized by invariance under the large gauge transformation generated by $\alpha^{yz} = 2\pi \frac{\tau}{\ell^\tau}\left[\frac{1}{\ell^z}\delta(y - y_0) + \frac{1}{\ell^y}\delta(z - z_0) - \frac{1}{\ell^y\ell^z}\right]$, where $z_1 < z_0 < z_2$, and $\alpha^{xy} = \alpha^{xz} = 0$. This large gauge transformation has a transition function at $\tau = \ell^\tau$ given by (5.17).

Here, the strip is given by $\mathcal{C}^{xy} \times [z_1, z_2]$. Each fiber of such a strip is oriented along the $z$ direction and cannot bend away from that. Similarly, there are static strips in the $yz$ and the $xz$ planes.

The strips can also move within their planes, but their fibers still cannot bend. For example, a charge $+1$, closed strip can move by itself in the $xy$ plane, but its fibers cannot bend away from the $z$ direction. The corresponding defect is

$$\tilde{W}_z(z_1, z_2, \mathcal{S}^{xyt}) = \exp\left[i \int_{z_1}^{z_2} dz \oint_{\mathcal{S}^{xyt}} \left(C_0^{zx} dxdt + C_0^{yz} dydt + C^{[xy]z} dxdy\right)\right] , \qquad (5.36)$$

where $\mathcal{S}^{xyt}$ is the world-sheet of the boundary of the strip at, say, $z_1$. For a static strip, $\mathcal{S}^{xyt} = \mathcal{C}^{xy} \times \mathbb{R}_t$. Similarly, there are moving strips in $yz$ and $xz$ planes.

At fixed time, with $\mathcal{S}^{xyt} = xy$ plane, we recover the slab operator $W_z(z_1, z_2)$ defined in (5.32). Similarly, we can recover the other slab operators.

## 5.6 Electric Modes

We place the system on a spatial 3-torus with lengths $\ell^x, \ell^y, \ell^z$. We pick the temporal gauge $C_0^{ij} = 0$, and then the Gauss law $\partial_k E^{k(ij)} = 0$ implies that up to a (time-independent) gauge transformation the spatial gauge fields take the form

$$C^{[xy]z}(t, x, y, z) = \frac{1}{\ell^x \ell^y} f^z(t, z) - \frac{1}{2\ell^y \ell^z} f^x(t, x) - \frac{1}{2\ell^z \ell^x} f^y(t, y) ,$$

$$C^{[yz]x}(t, x, y, z) = \frac{1}{\ell^y \ell^z} f^x(t, x) - \frac{1}{2\ell^z \ell^x} f^y(t, y) - \frac{1}{2\ell^x \ell^y} f^z(t, z) , \qquad (5.37)$$

$$C^{[zx]y}(t, x, y, z) = \frac{1}{\ell^z \ell^x} f^y(t, y) - \frac{1}{2\ell^x \ell^y} f^z(t, z) - \frac{1}{2\ell^y \ell^z} f^x(t, x) .$$

Note that there is no mode with nontrivial momenta in all the $x$, $y$, $z$ directions, therefore the theory has no propagating degrees of freedom.[9]

The three zero modes of $f^i(t, x^i)$ are not all physical. Indeed, the shift

$$f^i(t, x^i) \sim f^i(t, x^i) + \ell^j \ell^k c(t) , \qquad (5.38)$$

leaves gauge fields $C^{[ij]k}$ invariant. To remove this gauge ambiguity, we define gauge invari-

---

[9]Let us check this by counting the on-shell local degrees of freedom. Locally, we can use the freedom in $\gamma$ to set $\alpha_0 = 0$. The remaining gauge freedom is in $\alpha^{ij}$. We use it to fix the temporal gauge $C_0^{ij} = 0$. We are left with the two spatial degrees of freedom $C^{[ij]k}$, or equivalently $E^{[ij]k}$ (in **2**) and we need to impose Gauss law in **3'**. So we are left with no local degrees of freedom. This counting is similar to the counting in the ordinary two-form gauge theory in $2 + 1$ dimensions.

ant variables

$$\bar{f}^i(t, x^i) = f^i(t, x^i) - \frac{1}{2\ell^i} \oint dx^j \ f^j(t, x^j) - \frac{1}{2\ell^i} \oint dx^k \ f^k(t, x^k) \ . \tag{5.39}$$

However, these variables are not all independent; they satisfy a constraint

$$\oint dx \ \bar{f}^x(t, x) + \oint dy \ \bar{f}^y(t, y) + \oint dz \ \bar{f}^z(t, z) = 0 \ . \tag{5.40}$$

Large gauge transformations of the form (5.16) imply the following identifications in $\bar{f}^i$:

$$\bar{f}^i(t, x^i) \sim \bar{f}^i(t, x^i) + 2\pi\delta(x^i - x_0^i) \ . \tag{5.41}$$

Naively there appear to be $L^x L^y L^z$ identifications on a lattice. However, there are only $L^x + L^y + L^z - 1$ identifications. The constraint (5.40) implies that $\bar{f}^z(t, \hat{z} = 1)$ can be solved in terms of the other $\bar{f}$'s. The remaining $L^x + L^y + L^z - 1$ $\bar{f}$'s have periodicities $\bar{f} \sim \bar{f} + \frac{2\pi}{a}$.

The Lagrangian for these modes is

$$L = \frac{3}{2g_e^2 \ell^x \ell^y \ell^z} \left[ \ell^x \oint dx \ (\dot{\bar{f}}^x)^2 + \ell^y \oint dy \ (\dot{\bar{f}}^y)^2 + \ell^z \oint dz \ (\dot{\bar{f}}^z)^2 \right.$$
$$\left. + \frac{2}{3} \oint dxdy \ \dot{\bar{f}}^x \dot{\bar{f}}^y + \frac{2}{3} \oint dydz \ \dot{\bar{f}}^y \dot{\bar{f}}^z + \frac{2}{3} \oint dxdz \ \dot{\bar{f}}^x \dot{\bar{f}}^z \right] \ . \tag{5.42}$$

Let $\bar{\Pi}^i(x^i)$ be the conjugate momenta of $\bar{f}^i(x^i)$. Because of the delta functions periodicities of (5.41), the momentum $\bar{\Pi}^i(x^i)$ has independent integer eigenvalues at each $x^i$. Due to the constraint (5.40), the momenta are subject to a gauge ambiguity:

$$\bar{\Pi}^i(x^i) \sim \bar{\Pi}^i(x^i) + n \ , \tag{5.43}$$

for any integer $n \in \mathbb{Z}$. The charges of the electric tensor symmetry (5.27) can be written in terms of the momenta as

$$Q^{i(jk)}(x, y, z) = \frac{2}{g_e^2} E^{i(jk)} = \bar{\Pi}^k(x^k) - \bar{\Pi}^j(x^j) \ , \tag{5.44}$$

which agrees with (5.29). Therefore, on a lattice with $L^i$ sites in $x^i$ direction, there are $L^x + L^y + L^z - 1$ charges.

The Hamiltonian for these modes is

$$
\begin{aligned}
H = \frac{g_e^2}{6} \Bigg[ &\ell^y \ell^z \oint dx \, (\bar{\Pi}^x)^2 + \ell^x \ell^z \oint dy \, (\bar{\Pi}^y)^2 + \ell^x \ell^y \oint dz \, (\bar{\Pi}^z)^2 \\
&- \ell^z \oint dx dy \, \bar{\Pi}^x \bar{\Pi}^y - \ell^x \oint dy dz \, \bar{\Pi}^y \bar{\Pi}^z - \ell^y \oint dx dz \, \bar{\Pi}^x \bar{\Pi}^z \Bigg] \; .
\end{aligned}
\tag{5.45}
$$

On a lattice with lattice spacing $a$, since $\bar{\Pi}^i(\hat{x}^i)$ have integer eigenvalues at each point $\hat{x}^i$, an electric mode with a finite number of nonzero charges has energy of order $g_e^2 \ell^2 a$, which goes to zero in the continuum limit $a \to 0$ when $\ell$ is kept finite. The energy of the electric modes become order one if they have order $1/a$ number of nonzero charges. Similar to the discussions for the electric modes in $B$-theory, the zero energy modes are not lifted by higher derivative terms while the modes with energy of order one can received quantitative corrections of order one. Nevertheless, the qualitative scaling with $a$ remains universal.

# 6  $\mathbb{Z}_N$ Tensor Gauge Theory of $C$

## 6.1  $\mathbb{Z}_N$ Version of the Lattice Model of $\hat{\phi}$ and $\mathbb{Z}_N$ Version of the $C$ Tensor Gauge Theory

We consider two lattice models that have the same continuum limit, the lattice $\mathbb{Z}_N$ version of the $\hat{\phi}$-theory of [4] and the $\mathbb{Z}_N$ lattice tensor gauge theory of $C$. The two lattice models are dual to each other at long distance.

### 6.1.1  $\mathbb{Z}_N$ Lattice Model of $\hat{\phi}$

The first lattice model is the $\mathbb{Z}_N$ version of the $\hat{\phi}^{i(jk)}$ lattice model in Section 4.1 of [4]. There are three $\mathbb{Z}_N$ phase variables $\hat{U}_s^{i(jk)}$ and their conjugate momenta $\hat{V}_{i(jk)}^s$ at every site $s = (\hat{x}, \hat{y}, \hat{z})$, which satisfy $\hat{U}_s^{x(yz)} \hat{U}_s^{y(zx)} \hat{U}_s^{z(xy)} = 1$ and $\hat{V}_{i(jk)}^s \sim \hat{\eta}_s \hat{V}_{i(jk)}^s$, where $\hat{\eta}_s$ is an arbitrary $\mathbb{Z}_N$ phase. They obey the canonical commutation relations $\hat{U}_s^{i(jk)} \hat{V}_{i(jk)}^s = e^{2\pi i/N} \hat{V}_{i(jk)}^s \hat{U}_s^{i(jk)}$. We also define $\hat{V}_s^{[ij]k} = \hat{V}_{i(jk)}^s (\hat{V}_{j(ki)}^s)^{-1}$, which satisfy $\hat{V}_s^{[xy]z} \hat{V}_s^{[yz]x} \hat{V}_s^{[zx]y} = 1$. Note that both $\hat{V}_{i(jk)}^s$ and $\hat{V}_s^{[ij]k}$ have the same number of degrees of freedom.

The Hamiltonian is

$$
\begin{aligned}
&H = -\hat{K} \sum_s (\hat{L}_{xy} + \hat{L}_{yz} + \hat{L}_{zx}) - \hat{h} \sum_s (\hat{V}_s^{[xy]z} + \hat{V}_s^{[yz]x} + \hat{V}_s^{[zx]y}) + \text{c.c.} \; , \\
&\hat{L}_{xy} = \hat{U}_{\hat{x}, \hat{y}, \hat{z}+1}^{z(xy)} (\hat{U}_{\hat{x}, \hat{y}, \hat{z}}^{z(xy)})^{-1} \; .
\end{aligned}
\tag{6.1}
$$

We will assume $\hat{h}$ to be small.

There are symmetry operators that are products of $\hat{V}_s^{[ij]k}$ along $ij$ plane. For example, the symmetry operator along $xy$ plane is

$$\prod_{\hat{x}=1}^{L^x}\prod_{\hat{y}=1}^{L^y}\hat{V}_{\hat{x},\hat{y},\hat{z}_0}^{[xy]z} \, , \tag{6.2}$$

and similarly along $yz$ and $xz$ planes. There are $L^x + L^y + L^z - 1$ such operators on the lattice [4]. In the continuum, they become the $\mathbb{Z}_N$ $(\mathbf{2}, \mathbf{3'})$ tensor symmetry.

### 6.1.2   Lattice Tensor Gauge Theory of $C$

The second lattice model is the $\mathbb{Z}_N$ lattice tensor gauge theory of $C$. There are three $\mathbb{Z}_N$ phase variables $U_c^{[ij]k}$, and their conjugate momenta $V_c^{[ij]k}$ on every cube $c$, which satisfy the constraint $U_c^{[xy]z}U_c^{[yz]x}U_c^{[zx]y} = 1$ and $V_c^{[ij]k} \sim \eta_c V_c^{[ij]k}$, where $\eta_c$ is an arbitrary $\mathbb{Z}_N$ phase, and $i, j, k$ are cyclically ordered. They obey the canonical commutation relations $U_c^{[ij]k}V_c^{[ij]k} = e^{2\pi i/N}V_c^{[ij]k}U_c^{[ij]k}$. We also define $V_c^{i(jk)} = V_c^{[ki]j}V_c^{[kj]i}$, which satisfy $V_c^{x(yz)}V_c^{y(zx)}V_c^{z(xy)} = 1$. Note that both $V_c^{[ij]k}$ and $V_c^{i(jk)}$ have the same number of degrees of freedom.

For each spatial plaquette along the $ij$ direction, there is a a $\mathbb{Z}_N$ gauge parameter $\eta_{\hat{x},\hat{y},\hat{z}}^{ij}$. Under the gauge transformation,

$$U_c^{[xy]z} \sim U_c^{[xy]z}\eta_{\hat{x},\hat{y}+1,\hat{z}}^{xz}(\eta_{\hat{x},\hat{y},\hat{z}}^{xz})^{-1}(\eta_{\hat{x}+1,\hat{y},\hat{z}}^{yz})^{-1}\eta_{\hat{x},\hat{y},\hat{z}}^{yz} \, , \tag{6.3}$$

where the product is over four plaquettes in $xz$ and $yz$ planes around the cube $c$, and similarly for $U^{[yz]x}$ and $U^{[zx]y}$.

Gauss law sets

$$\begin{aligned}
G^{xy}(\hat{x}, \hat{y}, \hat{z}) &\equiv V_{\hat{x},\hat{y},\hat{z}+1}^{[zx]y}(V_{\hat{x},\hat{y},\hat{z}}^{[zx]y})^{-1}(V_{\hat{x},\hat{y},\hat{z}+1}^{[yz]x})^{-1}V_{\hat{x},\hat{y},\hat{z}}^{[yz]x} \\
&= V_{\hat{x},\hat{y},\hat{z}+1}^{z(xy)}(V_{\hat{x},\hat{y},\hat{z}}^{z(xy)})^{-1} = \prod_{c \ni p_{xy}}(V_c^{z(xy)})^{\epsilon_c} = 1 \, ,
\end{aligned} \tag{6.4}$$

where the product is an oriented product ($\epsilon_c = \pm 1$) over the two cubes $c$ that share a common plaquette $p_{xy}$ in the $xy$ direction. Similarly, there are two more Gauss laws in $yz$ and $xz$ directions. The Hamiltonian is

$$H = -\widetilde{h}\sum_c(V_c^{x(yz)} + V_c^{y(zx)} + V_c^{z(xy)}) + \text{c.c.} \, , \tag{6.5}$$

with Gauss law imposed as operator equations.

The symmetry operators are $V_c^{i(jk)}$ at each cube $c$. Because of the Gauss laws, there are $L^x + L^y + L^z - 1$ such operators. They become the $\mathbb{Z}_N$ electric tensor symmetry generators

in the continuum.

Alternatively, we can relax the Gauss and impose it energetically by adding a term to the Hamiltonian

$$H = -\hat{K} \sum_p G_p - \widetilde{h} \sum_c (V_c^{x(yz)} + V_c^{y(zx)} + V_c^{z(xy)}) + \text{c.c.} \ . \tag{6.6}$$

When both $\hat{h}$ of (6.1) and $\widetilde{h}$ of (6.6) vanish, the Hamiltonian (6.6) becomes the Hamiltonian (6.1) of the plaquette model if we dualize the lattice and identify $U_c^{[ij]k} \leftrightarrow (\hat{V}_s^{[ij]k})^{-1}$, $V_c^{i(jk)} \leftrightarrow \hat{U}_s^{i(jk)}$.

## 6.2  Continuum Lagrangian

We can obtain a continuum description of the $\mathbb{Z}_N$ theory by coupling the $U(1)$ $C$-theory to an $A$-theory with charge $N$ that Higgses it to $\mathbb{Z}_N$. The Euclidean Lagrangian is

$$\mathcal{L}_E = \frac{i}{2(2\pi)} \widetilde{B}_{ij} \left( (\partial_0 A^{ij} - \partial^i \partial^j A_0) - N C_0^{ij} \right) + \frac{i}{2(2\pi)} \widetilde{E}_{[ij]k} \left( (\partial^j A^{ki} - \partial^i A^{jk}) - N C^{[ij]k} \right) \ , \tag{6.7}$$

where $(C_0^{ij}, C^{[ij]k})$ are the $U(1)$ tensor gauge field in the $(\mathbf{3'}, \mathbf{2})$ representation of $S_4$, and $(A_0, A^{ij})$ are the $U(1)$ gauge field in the $(\mathbf{1}, \mathbf{3'})$ representation of $S_4$ that Higgses gauge symmetry of $(C_0^{ij}, C^{[ij]k})$ to $\mathbb{Z}_N$. The gauge transformations are

$$\begin{aligned}
A_0 &\sim A_0 + \partial_0 \beta + N\alpha_0 \ , \\
A^{ij} &\sim A^{ij} + \partial^i \partial^j \beta + N\alpha^{ij} \ , \\
C_0^{ij} &\sim C_0^{ij} + \partial_0 \alpha^{ij} - \partial^i \partial^j \alpha_0 \\
C^{[ij]k} &\sim C^{[ij]k} - \partial^i \alpha^{jk} + \partial^j \alpha^{ki} \ .
\end{aligned} \tag{6.8}$$

The equations of motion are

$$\partial_0 A^{ij} - \partial^i \partial^j A_0 - N C_0^{ij} = 0 \ , \quad \partial^j A^{ki} - \partial^i A^{jk} - N C^{[ij]k} = 0 \ , \quad \widetilde{B}_{ij} = 0 \ , \quad \widetilde{E}_{[ij]k} = 0 \ . \tag{6.9}$$

We can dualize the Euclidean action by integrating out $(A_0, A^{ij})$. We rewrite the Lagrangian (6.7) as

$$\mathcal{L}_E = -\frac{i}{2(2\pi)} A_0 \partial^i \partial^j \widetilde{B}_{ij} - \frac{i}{2(2\pi)} A^{ij} \left( \partial_0 \widetilde{B}_{ij} - 3\partial^k \widetilde{E}_{k(ij)} \right) - \frac{iN}{2(2\pi)} \left( \widetilde{B}_{ij} C_0^{ij} + \widetilde{E}_{[ij]k} C^{[ij]k} \right) \ , \tag{6.10}$$

We now interpret the Higgs fields $(A_0, A^{ij})$ as Lagrangian multipliers implementing the

constraints

$$\partial^i \partial^j \widetilde{B}_{ij} = 0 \; , \qquad \partial_0 \widetilde{B}_{ij} = 3 \partial^k \widetilde{E}_{k(ij)} \; , \tag{6.11}$$

Locally, these constraints can be solved by a field $\hat{\phi}_{i(jk)}$ in $\mathbf{2}$:

$$\widetilde{E}_{k(ij)} = \partial_0 \hat{\phi}_{k(ij)} \; , \qquad \widetilde{B}_{ij} = 3 \partial^k \hat{\phi}_{k(ij)} \; . \tag{6.12}$$

The Euclidean Lagrangian (6.7) then becomes

$$\mathcal{L}_E = \frac{iN}{2(2\pi)} \hat{\phi}_{[ij]k} \left( \partial_0 C^{[ij]k} + \partial^i C_0^{jk} - \partial^j C_0^{ik} \right) = \frac{iN}{2(2\pi)} \hat{\phi}_{[ij]k} E^{[ij]k} \; . \tag{6.13}$$

The equations of motion are

$$E^{[ij]k} = 0 \; , \qquad \partial_0 \hat{\phi}_{[ij]k} = 0 \; , \qquad \partial^k \hat{\phi}_{k(ij)} = 0 \; . \tag{6.14}$$

To see that the value of $N$ is quantized, let us place the theory on a Euclidean 4-torus. Under the (large gauge) transformation $\hat{\phi}^{x(yz)} \sim \hat{\phi}^{x(yz)} + 2\pi$, $\hat{\phi}^{y(zx)} \sim \hat{\phi}^{y(zx)} - 2\pi$ and $\hat{\phi}^{z(xy)} \sim \hat{\phi}^{z(xy)}$, the action of (6.13) shifts by

$$\frac{iN}{2\pi} \oint d\tau dx dy dz \left( \frac{4\pi}{3} E^{[xy]z} - \frac{2\pi}{3} E^{[yz]x} - \frac{2\pi}{3} E^{[zx]y} \right) = iN \oint d\tau dx dy dz \; E^{[xy]z} \; . \tag{6.15}$$

Since the fluxes are quantized (see Section 5.3), for the path integral to be invariant under this (large gauge) transformation, we need

$$N \in \mathbb{Z} \; . \tag{6.16}$$

## 6.3   Global Symmetries

Let us study the global symmetries of the $\mathbb{Z}_N$ tensor gauge theory. They are summarized in Figure 2. Since the $U(1)$ electric tensor symmetry of the $A$ theory is gauged, it turns the $U(1)$ magnetic tensor symmetry to $\mathbb{Z}_N$. Recall that this symmetry is dual to the $(\mathbf{2}, \mathbf{3}')$ momentum tensor symmetry of the $\hat{\phi}^{i(jk)}$ theory [4]. In addition, coupling the *matter* field $A$ to the pure gauge $C$ theory breaks the $U(1)$ electric tensor symmetry of the $C$ theory to $\mathbb{Z}_N$.

The $\mathbb{Z}_N$ electric tensor symmetry is generated by the local operators $e^{i\hat{\phi}^{i(jk)}}$, and the $\mathbb{Z}_N$ $(\mathbf{2}, \mathbf{3}')$ tensor symmetry is generated by the slab operators $W_i(x_1^i, x_2^i)$ of (5.32). They are

both charged under each other, and satisfy the commutation relations

$$e^{i\hat{\phi}^{z(xy)}(x,y,z)}W_x(x_1,x_2) = e^{-2\pi i/N}W_x(x_1,x_2)e^{i\hat{\phi}^{z(xy)}(x,y,z)}\ ,\quad\text{if}\quad x_1 < x < x_2\ ,$$

$$e^{i\hat{\phi}^{z(xy)}(x,y,z)}W_y(y_1,y_2) = e^{2\pi i/N}W_y(y_1,y_2)e^{i\hat{\phi}^{z(xy)}(x,y,z)}\ ,\quad\text{if}\quad y_1 < y < y_2\ ,\qquad(6.17)$$

$$e^{i\hat{\phi}^{z(xy)}(x,y,z)}W_z(z_1,z_2) = W_z(z_1,z_2)e^{i\hat{\phi}^{z(xy)}(x,y,z)}\ ,\quad\text{if}\quad z_1 < z < z_2\ ,$$

and they commute otherwise. The commutation relations of $e^{i\hat{\phi}^{y(zx)}}$ and $e^{i\hat{\phi}^{x(yz)}}$ are similar.

Because of the second equation of motion of $\hat{\phi}^{i(jk)}$ in (6.14), the electric tensor symmetry operator factorizes, for cyclically ordered $i, j, k$, into

$$e^{i\hat{\phi}^{i(jk)}(x,y,z)} = e^{i\hat{\phi}^j(x^j)-i\hat{\phi}^k(x^k)}\ ,\qquad(6.18)$$

where $e^{i\hat{\phi}^i(x^i)}$ have a gauge ambiguity, $e^{i\hat{\phi}^i(x^i)} \sim \hat{\eta}e^{i\hat{\phi}^i(x^i)}$, where $\hat{\eta}$ is an arbitrary $\mathbb{Z}_N$ phase.

Depending on the global symmetry we impose in the microscopic model, the local operator $e^{i\hat{\phi}^{i(jk)}}$ of the continuum theory may or may not be added to the Lagrangian to destabilize the theory. Let us demonstrate it in the two microscopic lattice models of Section 6.1.

In the $\mathbb{Z}_N$ plaquette lattice model discussed in Section 6.1.1, there is a microscopic $(\mathbf{2}, \mathbf{3}')$ tensor symmetry (6.2). So its continuum limit is robust since there are no relevant local operators that are invariant under this symmetry.

On the other hand, the $(\mathbf{2}, \mathbf{3}')$ tensor symmetry is absent in the lattice tensor gauge theory discussed in Section 6.1.2. In this lattice gauge theory, only the electric tensor symmetry, which is generated by $V_c^{i(jk)}$, is manifest. Therefore we can deform the short-distance theory by adding local operators $e^{i\hat{\phi}^{i(jk)}}$, which are charged under the $(\mathbf{2}, \mathbf{3}')$ tensor symmetry. This will generically lift the ground state degeneracy discussed in Section 6.4, and break the $\mathbb{Z}_N$ $(\mathbf{2}, \mathbf{3}')$ tensor symmetry.

## 6.4   Ground State Degeneracy

In the presentation (6.7), all the fields can be solved in terms of the gauge fields $(A_0, A^{ij})$, and the solution space reduces to

$$\left\{A_0, A^{ij}\ \middle|\ A_0 \sim A_0 + \partial_0\beta + N\alpha_0\ ,\ A^{ij} \sim A^{ij} + \partial^i\partial^j\beta + N\alpha^{ij}\right\}\ .\qquad(6.19)$$

The only modes that survive after gauging are the magnetic modes of the $A$ theory. If we regularize the theory on a lattice, these magnetic modes are labelled by $L^x + L^y + L^z - 1$ integers [4]. Large gauge transformations of $(\alpha_0, \alpha^{ij})$ identify these integers modulo $N$. Therefore, there are $N^{L^x+L^y+L^z-1}$ nontrivial magnetic modes and this is the ground state degeneracy.

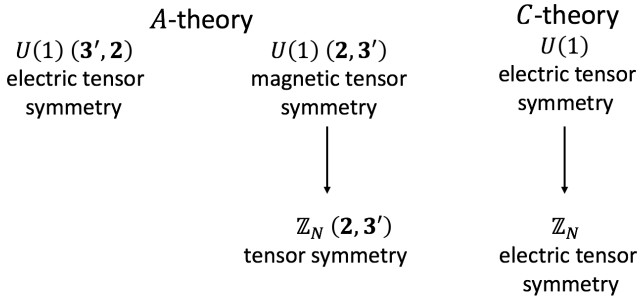

Figure 2: The global symmetries of the $U(1)$ $A$-theory, the $U(1)$ $C$-theory, the $\mathbb{Z}_N$ $C$-theory, and their relations. The electric tensor symmetry of the $A$-theory is gauged and therefore it is absent in the $\mathbb{Z}_N$ gauge theory. Note that the $U(1)$ $C$-theory does not have a magnetic symmetry.

There are other ways to see this. One can start with the $BF$-type presentation (6.13), find the solution space of the equations of motion (6.14) in the temporal gauge $C_0^{ij} = 0$, and then quantize these modes on a lattice. Yet another way to see this is by studying the symmetry operators on the lattice:

$$e^{i\hat{\phi}^{i(jk)}(\hat{x},\hat{y},\hat{z})} \ , \quad W_i(\hat{x}^i) \ . \tag{6.20}$$

The former factorizes, for cyclically ordered $i, j, k$, into

$$e^{i\hat{\phi}^{i(jk)}(\hat{x},\hat{y},\hat{z})} = e^{i\hat{\phi}^j(\hat{x}^j) - i\hat{\phi}^k(\hat{x}^k)} \ , \quad \forall \ \hat{x}, \hat{y}, \hat{z} \ , \tag{6.21}$$

because of the equation of motion (6.14) of $\hat{\phi}^{i(jk)}$ (or the Gauss law (6.4) of the $\mathbb{Z}_N$ electric tensor symmetry), where $e^{i\hat{\phi}^i(\hat{x}^i)}$ have a gauge ambiguity,

$$e^{i\hat{\phi}^i(\hat{x}^i)} \sim \hat{\eta} e^{i\hat{\phi}^i(\hat{x}^i)} \ , \tag{6.22}$$

where $\hat{\eta}$ is an arbitrary $\mathbb{Z}_N$ phase. In addition, we have the constraint

$$\prod_{\hat{x}=1}^{L^x} W_x(\hat{x}) \prod_{\hat{y}=1}^{L^y} W_y(\hat{y}) \prod_{\hat{z}=1}^{L^z} W_z(\hat{z}) = 1 \ . \tag{6.23}$$

Using the gauge ambiguity (6.22), we can fix $e^{i\hat{\phi}^z(\hat{z}=1)} = 1$, and using the constraint (6.23), we can solve for $W_z(\hat{z} = 1)$ in terms of other $W_i(\hat{x}^i)$. Therefore, there are $L^x + L^y + L^z - 1$ operators of each kind.

The set of commutation relations (6.17) of these operators is isomorphic to $L^x + L^y + L^z - 1$ copies of Heisenberg algebra, $AB = e^{2\pi i/N} BA$ and $A^N = B^N = 1$. The isomorphism is given by

$$
\begin{aligned}
A_{\hat{x}} &= e^{i\hat{\phi}^{y(zx)}(\hat{x},1,1)} , & B_{\hat{x}} &= W_x(\hat{x}) , & \hat{x} &= 1, \dots, L^x , \\
A_{\hat{y}} &= e^{-i\hat{\phi}^{x(yz)}(1,\hat{y},1)} , & B_{\hat{y}} &= W_y(\hat{y}) , & \hat{y} &= 1, \dots, L^y , \\
A_{\hat{z}} &= e^{i\hat{\phi}^{x(yz)}(1,1,\hat{z})-i\hat{\phi}^{x(yz)}(1,1,1)} , & B_{\hat{z}} &= W_z(\hat{z}) , & \hat{z} &= 2, \dots, L^z ,
\end{aligned}
\tag{6.24}
$$

These commutation relations force the ground state degeneracy to be $N^{L^x + L^y + L^z - 1}$.

# 7 $U(1)$ Tensor Gauge Theory of $\hat{C}$

Consider a $(\mathbf{R}_{\text{time}}, \mathbf{R}_{\text{space}}) = (\mathbf{3'}, \mathbf{1})$ dipole global symmetry generated by currents $(J_0^{ij}, J)$ [4]. The currents obey the conservation equation:

$$
\partial_0 J_0^{ij} = \partial^i \partial^j J ,
\tag{7.1}
$$

and a differential condition

$$
\partial^i J_0^{jk} = \partial^j J_0^{ik} .
\tag{7.2}
$$

We will study the pure gauge theory without matter obtained by gauging this $(\mathbf{3'}, \mathbf{1})$ dipole global symmetry.

We can gauge the $(\mathbf{3'}, \mathbf{1})$ dipole global symmetry by coupling the currents to the tensor gauge field $(\hat{C}_0^{ij}, \hat{C})$:

$$
\frac{1}{2} J_0^{ij} \hat{C}_0^{ij} + J\hat{C} .
\tag{7.3}
$$

The current conservation equation (7.1) and the differential condition (7.2) imply the gauge transformation

$$
\begin{aligned}
\hat{C}_0^{ij} &\sim \hat{C}_0^{ij} + \partial_0 \hat{\alpha}^{ij} - \partial_k \hat{\alpha}_0^{k(ij)} \\
\hat{C} &\sim \hat{C} + \frac{1}{2} \partial_i \partial_j \hat{\alpha}^{ij} ,
\end{aligned}
\tag{7.4}
$$

where $\hat{\alpha}_0^{i(jk)}$ satisfies a constraint $\hat{\alpha}_0^{x(yz)} + \hat{\alpha}_0^{y(zx)} + \hat{\alpha}_0^{z(xy)} = 0$. The gauge parameters $(\hat{\alpha}_0^{i(jk)}, \hat{\alpha}^{ij})$ are also gauge fields themselves in the $(\mathbf{2}, \mathbf{3'})$ representation of $S_4$. They have their own gauge transformation

$$
\begin{aligned}
\hat{\alpha}_0^{i(jk)} &\sim \hat{\alpha}_0^{i(jk)} + \partial_0 \hat{\gamma}^{i(jk)} , \\
\hat{\alpha}^{ij} &\sim \hat{\alpha}^{ij} + \partial_k \hat{\gamma}^{k(ij)} ,
\end{aligned}
\tag{7.5}
$$

where $\hat{\gamma}^{i(jk)}$ is in the $\mathbf{2}$ of $S_4$. The gauge transformation (7.5) of $(\hat{\alpha}_0^{i(jk)}, \hat{\alpha}^{ij})$ does not affect

the gauge transformation (7.4) of $(\hat{C}_0^{ij}, \hat{C})$. The gauge-invariant electric field is

$$\hat{E} = \partial_0 \hat{C} - \frac{1}{2}\partial_i\partial_j \hat{C}_0^{ij} \ , \tag{7.6}$$

which is in $\mathbf{1}$ of $S_4$, while there is no magnetic field.

Similar to the discussion in Section 5, we see that the gauge parameters $(\hat{\alpha}_0^{i(jk)}, \hat{\alpha}^{ij})$ are similar to the gauge fields $(\hat{A}_0^{i(jk)}, \hat{A}^{ij})$ of the $\hat{A}$-theory of [4], with their corresponding gauge symmetry (7.5). The $(\hat{C}_0^{ij}, \hat{C})$ gauge fields are similar to the field strengths $(\hat{E}^{ij}, \hat{B})$ of $(\hat{A}_0^{i(jk)}, \hat{A}^{ij})$. Furthermore, the electric field $\hat{E}$ of $(\hat{C}_0^{ij}, \hat{C})$ is similar to the Bianchi identity of the $\hat{A}$-theory. Physically, it means that the gauge fields $(\hat{A}_0^{i(jk)}, \hat{A}^{ij})$ are the Higgs fields of $(\hat{C}_0^{ij}, \hat{C})$ (see Section 8.2). Again, this analogy is the same as the relation between higher form gauge theories of different degrees.

## 7.1 The Lattice Model

In this subsection, we will discuss the $U(1)$ lattice tensor gauge theory of $\hat{C}$. We will present both the Lagrangian and Hamiltonian presentations of this lattice model.

We will consider a Euclidean lattice with lattice spacing $a$. The gauge parameters are placed on the links. On each spatial link along $k$ direction of the Euclidean lattice, there is a gauge parameter $\hat{\eta}^{ij} = e^{ia\hat{\alpha}^{ij}}$ in $\mathbf{3}'$ of $S_4$.[10] On each temporal link, there are three gauge parameters $\hat{\eta}_\tau^{i(jk)} = e^{ia\hat{\alpha}_\tau^{i(jk)}}$ in $\mathbf{2}$ of $S_4$ satisfying $\hat{\eta}_\tau^{i(jk)}\hat{\eta}_\tau^{j(ki)}\hat{\eta}_\tau^{k(ij)} = 1$. The gauge parameters of the gauge parameters are placed on the sites. On each site, there are three such parameters $\hat{\lambda}^{i(jk)} = e^{i\hat{\gamma}^{i(jk)}}$ in $\mathbf{2}$ of $S_4$ satisfying $\hat{\lambda}^{i(jk)}\hat{\lambda}^{j(ki)}\hat{\lambda}^{k(ij)} = 1$.

The gauge fields are placed on the cubes and plaquettes. On each spatial cube, there is a gauge field $\hat{U} = e^{ia^3\hat{C}}$ in $\mathbf{1}$ of $S_4$. On each plaquette in $\tau z$ direction, there is a gauge field $\hat{U}_\tau^{xy} = e^{ia^2\hat{C}_0^{xy}}$, and similarly in the $\tau y$ and $\tau z$ directions. They are in $\mathbf{3}'$ of $S_4$.

---

[10]When the theory has a charge conjugation symmetry $\mathbb{Z}_2^C$, we can label our fields using the representations of $S_4^C$ defined in Section 2.6, instead of the original $S_4$. The two choices are related by an outer automorphism of the global symmetry $\mathbb{Z}_2^C \times S_4$. Then the $S_4^C$ representations for the $\hat{C}$ gauge fields are $(\mathbf{R}_{\text{time}}, \mathbf{R}_{\text{space}}) = (\mathbf{3}, \mathbf{1}')$. This convention is more natural for the lattice models here where the spatial gauge parameters $\hat{\alpha}^{ij}$, which is in the $\mathbf{3}$ of $S_4^C$, are placed on the links.

The gauge transformations act on the gauge fields as

$$
\begin{aligned}
\hat{U}_\tau^{xy}(\hat{\tau},\hat{x},\hat{y},\hat{z}) \sim{}& \hat{U}_\tau^{xy}(\hat{\tau},\hat{x},\hat{y},\hat{z})\hat{\eta}^{xy}(\hat{\tau},\hat{x},\hat{y},\hat{z})^{-1}\hat{\eta}^{xy}(\hat{\tau}+1,\hat{x},\hat{y},\hat{z}) \\
&\times \hat{\eta}_\tau^{z(xy)}(\hat{\tau},\hat{x},\hat{y},\hat{z})\hat{\eta}_\tau^{z(xy)}(\hat{\tau},\hat{x},\hat{y},\hat{z}+1)^{-1} , \\
\hat{U}(\hat{\tau},\hat{x},\hat{y},\hat{z}) \sim{}& \hat{U}(\hat{\tau},\hat{x},\hat{y},\hat{z})\hat{\eta}^{xy}(\hat{\tau},\hat{x}+1,\hat{y}+1,\hat{z})\hat{\eta}^{xy}(\hat{\tau},\hat{x},\hat{y},\hat{z}) \\
&\times \hat{\eta}^{xy}(\hat{\tau},\hat{x}+1,\hat{y},\hat{z})^{-1}\hat{\eta}^{xy}(\hat{\tau},\hat{x},\hat{y}+1,\hat{z})^{-1} \\
&\times \hat{\eta}^{yz}(\hat{\tau},\hat{x},\hat{y}+1,\hat{z}+1)\hat{\eta}^{yz}(\hat{\tau},\hat{x},\hat{y},\hat{z}) \\
&\times \hat{\eta}^{yz}(\hat{\tau},\hat{x},\hat{y},\hat{z}+1)^{-1}\hat{\eta}^{yz}(\hat{\tau},\hat{x},\hat{y}+1,\hat{z})^{-1} \\
&\times \hat{\eta}^{zx}(\hat{\tau},\hat{x}+1,\hat{y},\hat{z}+1)\hat{\eta}^{zx}(\hat{\tau},\hat{x},\hat{y},\hat{z}) \\
&\times \hat{\eta}^{zx}(\hat{\tau},\hat{x}+1,\hat{y},\hat{z})^{-1}\hat{\eta}^{zx}(\hat{\tau},\hat{x},\hat{y},\hat{z}+1)^{-1} ,
\end{aligned}
\tag{7.7}
$$

and similarly for $\hat{U}_\tau^{yz}$ and $\hat{U}_\tau^{zx}$. The gauge transformations themselves have gauge transformations given by

$$
\begin{aligned}
\hat{\eta}_\tau^{i(jk)}(\hat{\tau},\hat{x},\hat{y},\hat{z}) &\sim \hat{\eta}_\tau^{i(jk)}(\hat{\tau},\hat{x},\hat{y},\hat{z})\hat{\lambda}^{i(jk)}(\hat{\tau}+1,\hat{x},\hat{y},\hat{z})\hat{\lambda}^{i(jk)}(\hat{\tau},\hat{x},\hat{y},\hat{z})^{-1} , \\
\hat{\eta}^{xy}(\hat{\tau},\hat{x},\hat{y},\hat{z}) &\sim \hat{\eta}^{xy}(\hat{\tau},\hat{x},\hat{y},\hat{z})\hat{\lambda}^{z(xy)}(\hat{\tau},\hat{x},\hat{y},\hat{z}+1)\hat{\lambda}^{z(xy)}(\hat{\tau},\hat{x},\hat{y},\hat{z})^{-1} ,
\end{aligned}
\tag{7.8}
$$

and similarly for $\hat{\eta}^{yz}$ and $\hat{\eta}^{zx}$.

Let us discuss the gauge-invariant local terms in the action. There is only one kind of term on each spacetime hyper-cube:

$$
\begin{aligned}
\hat{L}(\hat{\tau},\hat{x},\hat{y},\hat{z}) ={}& \hat{U}_\tau^{xy}(\hat{\tau},\hat{x},\hat{y},\hat{z})^{-1}\hat{U}_\tau^{xy}(\hat{\tau},\hat{x}+1,\hat{y}+1,\hat{z})^{-1}\hat{U}_\tau^{xy}(\hat{\tau},\hat{x}+1,\hat{y},\hat{z})\hat{U}_\tau^{xy}(\hat{\tau},\hat{x},\hat{y}+1,\hat{z}) \\
&\times \hat{U}_\tau^{yz}(\hat{\tau},\hat{x},\hat{y},\hat{z})^{-1}\hat{U}_\tau^{yz}(\hat{\tau},\hat{x},\hat{y}+1,\hat{z}+1)^{-1}\hat{U}_\tau^{yz}(\hat{\tau},\hat{x},\hat{y},\hat{z}+1)\hat{U}_\tau^{yz}(\hat{\tau},\hat{x},\hat{y}+1,\hat{z}) \\
&\times \hat{U}_\tau^{zx}(\hat{\tau},\hat{x},\hat{y},\hat{z})^{-1}\hat{U}_\tau^{zx}(\hat{\tau},\hat{x}+1,\hat{y},\hat{z}+1)^{-1}\hat{U}_\tau^{zx}(\hat{\tau},\hat{x}+1,\hat{y},\hat{z})\hat{U}_\tau^{zx}(\hat{\tau},\hat{x},\hat{y},\hat{z}+1) \\
&\times \hat{U}(\hat{\tau},\hat{x},\hat{y},\hat{z})^{-1}\hat{U}(\hat{\tau}+1,\hat{x},\hat{y},\hat{z}) .
\end{aligned}
\tag{7.9}
$$

This term together with its complex conjugate becomes the square of the electric field in the continuum limit.

In addition to the local, gauge-invariant operators above, there are non-local, extended ones. For example, we have a *slab operator* along the $xy$ plane:

$$
\prod_{\hat{x}=1}^{L^x}\prod_{\hat{y}=1}^{L^y}\hat{U}(\hat{x},\hat{y},\hat{z}_0) .
\tag{7.10}
$$

Similarly, we have slab operators along $yz$ and $zx$ planes.

In the Hamiltonian formulation, we choose the temporal gauge to set all $\hat{U}_\tau^{ij}=1$. We introduce the electric field $\hat{E}$ such that $\frac{2}{\hat{g}_e^2}\hat{E}$ is conjugate to the phase of the spatial variable

$\hat{U}$ with $\hat{g}_e$ the electric coupling constant (up to some dimensionful factors of the lattice spacing $a$).

On every spatial link in the $z$ direction, we impose the Gauss law

$$\hat{G}^{xy}(\hat{x}, \hat{y}, \hat{z}) = \hat{E}(\hat{x}+1, \hat{y}+1, \hat{z}) - \hat{E}(\hat{x}+1, \hat{y}, \hat{z}) - \hat{E}(\hat{x}, \hat{y}+1, \hat{z}) + \hat{E}(\hat{x}, \hat{y}, \hat{z}) = 0 , \quad (7.11)$$

and similarly in the $x$ and $y$ directions. So there are three Gauss laws $\hat{G}^{ij} = 0$.

The Hamiltonian is the sum of $\hat{E}^2$ over all cubes, with the three Gauss laws imposed as operator equations. Alternatively, we can impose the Gauss laws energetically by adding a term $\sum_{\text{links}} (\hat{G}^{ij})^2$ to the Hamiltonian.

The lattice model has an *electric symmetry* whose conserved charges are proportional to

$$\hat{E}(\hat{x}_0, \hat{y}_0, \hat{z}_0) . \quad (7.12)$$

They trivially commute with the Hamiltonian. The electric symmetry shifts the phase variable $\hat{U}$ at a single spatial cube:

$$\hat{U} \to e^{i\alpha}\hat{U} . \quad (7.13)$$

Using Gauss law (7.11), the dependence of the conserved charge $\hat{E}$ on the spatial cube is a function of $\hat{x}$ plus a function of $\hat{y}$ plus a function of $\hat{z}$.

## 7.2   Continuum Lagrangian

The Lorentzian Lagrangian of the pure tensor gauge theory is

$$\mathcal{L} = \frac{1}{\hat{g}_e^2}\hat{E}^2 + \frac{\hat{\theta}}{2\pi}\hat{E} , \quad (7.14)$$

where the $\hat{\theta}$ parameter is $2\pi$-periodic due to the quantization of the electric flux (7.21). The equation of motion are

$$\frac{2}{\hat{g}_e^2}\partial_0\hat{E} = 0, \quad \partial_i\partial_j\hat{E} = 0 . \quad (7.15)$$

where the second equation is the Gauss law.

If $\hat{\theta} = 0, \pi$, the global symmetry includes $\mathbb{Z}_2^C \times S_4$, where $\mathbb{Z}_2^C$ is a charge conjugation symmetry that flips the sign of $\hat{C}_0^{ij}, \hat{C}$. Other values of $\hat{\theta}$ break the $\mathbb{Z}_2^C \times S_4$ symmetry to $S_4^C$, where the latter group is defined in Section 2.6. In addition, for every value of $\hat{\theta}$ there are parity and time reversal symmetries, under which $\hat{E}$ is invariant.

## 7.3 Fluxes

Let us put the theory on a Euclidean 4-torus with lengths $\ell^\tau$, $\ell^x$, $\ell^y$, $\ell^z$. Consider gauge field configurations with a nontrivial transition function at $\tau = \ell^\tau$:[11]

$$
\hat{g}^{xy}_{(\tau)} = 2\pi \left[ \frac{xy}{\ell^x \ell^y} \delta(z - z_0) + \frac{x}{\ell^x \ell^z} \Theta(y - y_0) + \frac{y}{\ell^y \ell^z} \Theta(x - x_0) - \frac{2xy}{\ell^x \ell^y \ell^z} \right] ,
$$
$$
\hat{g}^{yz}_{(\tau)} = 0 , \tag{7.16}
$$
$$
\hat{g}^{zx}_{(\tau)} = 0 ,
$$

which has its own transition function at $x = \ell^x$

$$
\hat{h}^{x(yz)}_{(x)} = -2\pi \left[ \frac{z}{\ell^z} \Theta(y - y_0) + \frac{y}{\ell^y} \Theta(z - z_0) - \frac{yz}{\ell^y \ell^z} \right] ,
$$
$$
\hat{h}^{y(zx)}_{(x)} = 0 , \tag{7.17}
$$
$$
\hat{h}^{z(xy)}_{(x)} = -\hat{h}^{x(yz)}_{(x)} ,
$$

and the transition function at $y = \ell^y$

$$
\hat{h}^{x(yz)}_{(y)} = 0 ,
$$
$$
\hat{h}^{y(zx)}_{(y)} = -2\pi \left[ \frac{z}{\ell^z} \Theta(x - x_0) + \frac{x}{\ell^x} \Theta(z - z_0) - \frac{xz}{\ell^x \ell^z} \right] , \tag{7.18}
$$
$$
\hat{h}^{z(xy)}_{(y)} = -\hat{h}^{y(zx)}_{(y)} ,
$$

and there is no nontrivial transition function at $z = \ell^z$. We have $\hat{C}(\tau + \ell^\tau, x, y, z) = \hat{C}(\tau, x, y, z) + \partial_x \partial_y \hat{g}^{xy}_{(\tau)}(x, y, z)$. We also have $\hat{g}^{xy}_{(\tau)}(x + \ell^x, y, z) = \hat{g}^{xy}_{(\tau)}(x, y, z) + \partial_z \hat{h}^{z(xy)}_{(x)}$ and $\hat{g}^{xy}_{(\tau)}(x, y + \ell^y, z) = \hat{g}^{xy}_{(\tau)}(x, y, z) + \partial_z \hat{h}^{z(xy)}_{(y)}$.

A gauge field configuration with the above transition functions is

$$
\hat{C} = \frac{2\pi\tau}{\ell^\tau \ell^x \ell^y \ell^z} \left[ \ell^z \delta(z - z_0) + \ell^y \delta(y - y_0) + \ell^x \delta(x - x_0) - 2 \right] , \tag{7.19}
$$

with the electric field

$$
\hat{E} = \frac{2\pi}{\ell^\tau \ell^x \ell^y \ell^z} \left[ \ell^z \delta(z - z_0) + \ell^y \delta(y - y_0) + \ell^x \delta(x - x_0) - 2 \right] . \tag{7.20}
$$

Since the electric field $\hat{E}$ has mass dimension 4, it is allowed to have delta function singu-

---

[11] As in all our theories, we allow certain singular configurations provided the terms in the Lagrangian are not too singular (see [3–5]). Here we follow the same rules as in [4]. It would be nice to understand better the precise rules controlling these singularities.

larities according to the rules in [3,4]. Such configurations have nontrivial electric flux

$$\hat{e}_{(z)}(z_1, z_2) = \oint d\tau \oint dx \oint dy \int_{z_1}^{z_2} dz \, \hat{E} \in 2\pi\mathbb{Z} \, , \qquad (7.21)$$

and similarly there are fluxes $\hat{e}_{(x)}(x_1, x_2)$ and $\hat{e}_{(y)}(y_1, y_2)$ along $x$ and $y$ direction. In particular when integrated over the whole spacetime, the flux is an integer multiple of $2\pi$.

On the lattice, these nontrivial fluxes correspond to the products

$$\prod_{\hat{\tau},\hat{x},\hat{y}} \hat{L} = 1 \, , \quad \prod_{\hat{\tau},\hat{y},\hat{z}} \hat{L} = 1 \, , \quad \prod_{\hat{\tau},\hat{x},\hat{z}} \hat{L} = 1 \, . \qquad (7.22)$$

## 7.4 Global Symmetries and Their Charges

We now discuss the global symmetries of the $\hat{C}$-theory.

The equation of motion (7.15) is identified as the current conservation equation

$$\partial_0 J_0 = 0 \, , \qquad (7.23)$$

with current

$$J_0 = \frac{2}{\hat{g}_e^2} \hat{E} + \frac{\hat{\theta}}{2\pi} \, . \qquad (7.24)$$

We define the current with a shift by $\frac{\hat{\theta}}{2\pi}$ so that the conserved charge is properly quantized (see (7.47)). The second equation of (7.15) is an additional differential equation imposed on $J_0$:

$$\partial_i \partial_j J_0 = 0 \, . \qquad (7.25)$$

We will refer to this current as *electric symmetry*. This is the continuum version of the lattice symmetry (7.12). Note that the current does not have spatial components.

The charges are

$$Q(x, y, z) = J_0(x, y, z) \, , \qquad (7.26)$$

at every point in space. The differential condition (7.25) means the charge satisfies

$$Q(x, y, z) = Q^x(x) + Q^y(y) + Q^z(z) \, , \qquad (7.27)$$

where $Q^i(x^i) \in \mathbb{Z}$. Only the sum of zero modes of $Q^i(x^i)$ is physical because, the shift

$$Q^x(x) \sim Q^x(x) + n^x \, , \quad Q^y(y) \sim Q^y(y) + n^y \, , \quad Q^z(z) \sim Q^z(z) - n^x - n^y \, , \qquad (7.28)$$

where $n^x, n^y \in \mathbb{Z}$, does not change the charges. So, on a lattice, there are $L^x + L^y + L^z - 2$ charges.

The symmetry operator is

$$\mathcal{U}(\beta; x, y, z) = \exp\left[i\frac{2\beta}{\hat{g}_e^2}\hat{E}(x, y, z)\right] . \tag{7.29}$$

The electric symmetry acts on the gauge fields as

$$\hat{C}(x, y, z) \to \hat{C}(x, y, z) + c^x(x) + c^y(y) + c^z(z) . \tag{7.30}$$

The operators charged under this electric tensor symmetry are *slab operators*:

$$\hat{W}_x(x_1, x_2) = \exp\left(i \int_{x_1}^{x_2} dx \oint dy \oint dz \, \hat{C}\right) ,$$

$$\hat{W}_y(y_1, y_2) = \exp\left(i \oint dx \int_{y_1}^{y_2} dy \oint dz \, \hat{C}\right) , \tag{7.31}$$

$$\hat{W}_z(z_1, z_2) = \exp\left(i \oint dx \oint dy \int_{z_1}^{z_2} dz \, \hat{C}\right) .$$

These correspond to the slab operators (7.10) on the lattice. The symmetry operator (7.29) and charged operators (7.31) satisfy the commutation relation

$$\mathcal{U}(\beta; x, y, z)\hat{W}_x(x_1, x_2) = e^{i\beta}\hat{W}_x(x_1, x_2)\mathcal{U}(\beta; x, y, z) , \quad \text{if} \quad x_1 < x < x_2 , \tag{7.32}$$

and they commute otherwise. Similarly, there are commutation relations for $\hat{W}_y(y_1, y_2)$ and $\hat{W}_z(z_1, z_2)$.

Only integer powers of $\hat{W}_i$ are invariant under the large gauge transformation (7.16). It then follows that $\beta$ is $2\pi$-periodic. Therefore, the global structure of the electric symmetry is $U(1)$, rather than $\mathbb{R}$.

## 7.5 Defects as Fractonic Strings

The theory has no probe particles but it has probe strings. There are three types of strings associated to three spatial directions. A charge $+1$ string associated to the $x^i$ direction can extends only in the $x^i$ direction. For example, the string along the $x$ direction is described by the defect:

$$\exp\left(i \int_{-\infty}^{\infty} dt \oint dx \, \hat{C}_0^{yz}\right) , \tag{7.33}$$

and similarly for the defects along the other directions. We can study the Euclidean version of the surface defects and let it wrap around the Euclidean time. The charge is quantized because of the large gauge transformation $\hat{\alpha}^{yz} = \frac{2\pi\tau}{\ell^x\ell^\tau}$, $\hat{\alpha}^{xy} = \hat{\alpha}^{zx} = 0$. The large gauge

transformation has its own transition function at $\tau = \ell^\tau$, $\hat{\alpha}^{yz}(\tau + \ell^\tau) = \hat{\alpha}^{yz}(\tau) + \partial_x \hat{\gamma}^{x(yz)}$ with $\hat{\gamma}^{x(yz)} = \frac{2\pi x}{\ell^x}$.

The string above cannot move in the $y$ or the $z$ directions, but a pair of them with opposite charges separated in the $z$ direction can move collectively in the $y$ direction. The motion is described by the defect

$$\exp\left[i \int_{z_1}^{z_2} dz \oint dx \int_{\mathcal{C}} \left(\partial_z \hat{C}_0^{yz} dt + \hat{C} dy\right)\right] , \qquad (7.34)$$

where $\mathcal{C}$ is a spacetime curve in $(t, y)$.

More generally, a pair of strings separated in the $z$ direction can form a closed loop in $xy$-plane that can evolve in time:

$$\exp\left[i \int_{z_1}^{z_2} dz \oint_{\mathcal{S}} \left(\partial_z \hat{C}_0^{yz} dxdt - (\partial_z \hat{C}_0^{zx} + \partial_y \hat{C}_0^{xy})dydt + \hat{C} dxdy\right)\right] , \qquad (7.35)$$

where $\mathcal{S}$ is a spacetime sheet in $(t, x, y)$.[12] It is straightforward to check that these defects are gauge invariant. We have similar defects for the other directions.

## 7.6   Electric Modes

We place the system on a spatial 3-torus with lengths $\ell^x, \ell^y, \ell^z$. We pick the temporal gauge $\hat{C}_0^{ij} = 0$ and then Gauss law $\partial_i \partial_j \hat{E} = 0$ states that up to a (time-independent) gauge transformation the field takes the form

$$\hat{C}(t, x, y, z) = \frac{1}{\ell^x \ell^y} \hat{f}^z(t, z) + \frac{1}{\ell^x \ell^z} \hat{f}^y(t, y) + \frac{1}{\ell^y \ell^z} \hat{f}^x(t, x) . \qquad (7.37)$$

Note that there is no mode with nontrivial momenta in all the $x, y, z$ directions, therefore the theory has no propagating degrees of freedom.[13]

Only the sum of the zero modes of $\frac{1}{\ell^x \ell^y} \hat{f}^z(t, z)$, $\frac{1}{\ell^x \ell^z} \hat{f}^y(t, y)$ and $\frac{1}{\ell^y \ell^z} \hat{f}^x(t, x)$ is physical.

___________

[12] The $x$ and $y$ indices of the defect appear to be on different footings in (7.35). This is not the case, however, since we can rewrite it as

$$\exp\left[i \int_{z_1}^{z_2} dz \oint_{\mathcal{S}} \left((\partial_z \hat{C}_0^{yz} + \partial_x \hat{C}_0^{xy})dxdt - \partial_z \hat{C}_0^{zx} dydt + \hat{C} dxdy\right)\right] , \qquad (7.36)$$

using Stokes' theorem $\oint_{\mathcal{S}}(\partial_x \hat{C}_0^{xy} dxdt + \partial_y \hat{C}_0^{xy} dydt) = 0$.

[13] Let us check this by counting the on-shell local degrees of freedom. Locally, we can use the freedom in $\hat{\gamma}^{i(jk)}$ to set $\hat{\alpha}_0^{i(jk)} = 0$. The remaining gauge freedom is in $\hat{\alpha}^{ij}$. We use it to fix the temporal gauge $\hat{C}_0^{ij} = 0$. We are left with one degrees of freedom $\hat{C}$, or equivalently $\hat{E}$ and we need to impose Gauss law in **3′**. So we are left with no local degrees of freedom. This counting is similar to the counting in the ordinary two-form gauge theory in $2+1$ dimensions and in the $C$ theory.

This implies a gauge symmetry:

$$\hat{f}^x(t,x) \sim \hat{f}^x(t,x) + \ell^y \ell^z c_1(t) + \ell^y \ell^z c_2(t)$$
$$\hat{f}^y(t,y) \sim \hat{f}^y(t,y) - \ell^x \ell^z c_1(t) \tag{7.38}$$
$$\hat{f}^z(t,z) \sim \hat{f}^z(t,z) - \ell^x \ell^y c_2(t)$$

To remove this gauge ambiguity, we define the gauge-invariant variables $\bar{f}^i$ as

$$\bar{f}^x(t,x) = \hat{f}^x(t,x) + \frac{1}{\ell^x} \oint dy \, \hat{f}^y(t,y) + \frac{1}{\ell^x} \oint dz \, \hat{f}^z(t,z) \ ,$$
$$\bar{f}^y(t,y) = \hat{f}^y(t,y) + \frac{1}{\ell^y} \oint dz \, \hat{f}^z(t,z) + \frac{1}{\ell^y} \oint dx \, \hat{f}^x(t,x) \tag{7.39}$$
$$\bar{f}^z(t,z) = \hat{f}^z(t,z) + \frac{1}{\ell^z} \oint dx \, \hat{f}^x(t,x) + \frac{1}{\ell^z} \oint dy \, \hat{f}^y(t,y)$$

These variables are subject to a constraint

$$\oint dx \, \bar{f}^x(t,x) = \oint dy \, \bar{f}^y(t,y) = \oint dz \, \bar{f}^z(t,z) \ . \tag{7.40}$$

By performing a large gauge transformation of the form (7.16), we obtain the following identifications on $\bar{f}^i$:

$$\bar{f}^x(t,x) \sim \bar{f}^x(t,x) + 2\pi\delta(x - x_0)$$
$$\bar{f}^y(t,y) \sim \bar{f}^y(t,y) + 2\pi\delta(y) \tag{7.41}$$
$$\bar{f}^z(t,z) \sim \bar{f}^z(t,z) + 2\pi\delta(z)$$

for each $x_0$, and

$$\bar{f}^x(t,x) \sim \bar{f}^x(t,x)$$
$$\bar{f}^y(t,y) \sim \bar{f}^y(t,y) + 2\pi\delta(y - y_0) - 2\pi\delta(y) \tag{7.42}$$
$$\bar{f}^z(t,z) \sim \bar{f}^z(t,z)$$

for each $y_0$, and

$$\bar{f}^x(t,x) \sim \bar{f}^x(t,x)$$
$$\bar{f}^y(t,y) \sim \bar{f}^y(t,y) \tag{7.43}$$
$$\bar{f}^z(t,z) \sim \bar{f}^z(t,z) + 2\pi\delta(z - z_0) - 2\pi\delta(z)$$

for each $z_0$. On a lattice with $L^i$ sites in the $x^i$ direction, we can solve the first $\bar{f}^y(\hat{y} = 1)$ and $\bar{f}^z(\hat{z} = 1)$ in terms of the other coordinates using (7.40), then the remaining $L^x + L^y + L^z - 2$ $\bar{f}$'s have periodicities $\bar{f} \sim \bar{f} + \frac{2\pi}{a}$.

The Lagrangian for these modes is

$$L = \frac{1}{\hat{g}_e^2 \ell^x \ell^y \ell^z} \left[ \ell^x \oint dx (\dot{\bar{f}}^x)^2 + \ell^y \oint dy (\dot{\bar{f}}^y)^2 + \ell^z \oint dz (\dot{\bar{f}}^z)^2 - 2 \left( \oint dx \dot{\bar{f}}^x \right)^2 \right] + \frac{\hat{\theta}}{2\pi} \oint dx \, \dot{\bar{f}}^x \,,$$
(7.44)

Let $\bar{\Pi}^x(x)$, $\bar{\Pi}^y(y)$ and $\bar{\Pi}^z(z)$ be the conjugate momenta of $\bar{f}^i$. The delta function periodicity (7.41), (7.42) and (7.43) imply that $\bar{\Pi}^i(x^i)$ have independent integer eigenvalues at every $x^i$. Due to the constraint (7.40) on $\bar{f}^i$, the conjugate momenta $\bar{\Pi}^i$ are subject to a gauge ambiguity generated by the constraint:

$$\begin{aligned}
\bar{\Pi}^x(x) &\sim \bar{\Pi}^x(x) + 1 \,, \\
\bar{\Pi}^y(y) &\sim \bar{\Pi}^y(y) - 1 \,, \\
\bar{\Pi}^z(z) &\sim \bar{\Pi}^z(z) \,,
\end{aligned}$$
(7.45)

and

$$\begin{aligned}
\bar{\Pi}^x(x) &\sim \bar{\Pi}^x(x) + 1 \,, \\
\bar{\Pi}^y(y) &\sim \bar{\Pi}^y(y) \,, \\
\bar{\Pi}^z(z) &\sim \bar{\Pi}^z(z) - 1 \,.
\end{aligned}$$
(7.46)

The charge of the electric global symmetry (7.26) is expressed in terms of the conjugate momenta as

$$Q(x, y, z) = \frac{2}{\hat{g}_e^2} \hat{E} + \frac{\hat{\theta}}{2\pi} = \bar{\Pi}^x(x) + \bar{\Pi}^y(y) + \bar{\Pi}^z(z) \,.$$
(7.47)

The Hamiltonian is

$$\begin{aligned}
H = \frac{\hat{g}_e^2}{4} &\left[ \ell^y \ell^z \oint dx \left( \bar{\Pi}^x - \frac{\hat{\theta}^x}{2\pi} \right)^2 + \ell^x \ell^z \oint dy \left( \bar{\Pi}^y - \frac{\hat{\theta}^y}{2\pi} \right)^2 + \ell^x \ell^y \oint dz \left( \bar{\Pi}^z - \frac{\hat{\theta}^z}{2\pi} \right)^2 \right. \\
&+ 2\ell^z \oint dxdy \left( \bar{\Pi}^x - \frac{\hat{\theta}^x}{2\pi} \right) \left( \bar{\Pi}^y - \frac{\hat{\theta}^y}{2\pi} \right) + 2\ell^y \oint dxdz \left( \bar{\Pi}^x - \frac{\hat{\theta}^x}{2\pi} \right) \left( \bar{\Pi}^z - \frac{\hat{\theta}^z}{2\pi} \right) \\
&+ \left. 2\ell^x \oint dydz \left( \bar{\Pi}^y - \frac{\hat{\theta}^y}{2\pi} \right) \left( \bar{\Pi}^z - \frac{\hat{\theta}^z}{2\pi} \right) \right] \,.
\end{aligned}$$
(7.48)

where $\hat{\theta}_x + \hat{\theta}_y + \hat{\theta}_z = \hat{\theta}$. One can show that the Hamiltonian depends only on the sum of $\hat{\theta}_x, \hat{\theta}_y, \hat{\theta}_z$. Let us regularize the Hamiltonian on a lattice with lattice spacing $a$. States with finitely many nonzero $\bar{\Pi}^i(x^i)$ have energy of order $\hat{g}_e^2 \ell^2 a$. States with order $1/a$ many nonzero $\bar{\Pi}^i(x^i)$ have energy of order one. Similar to the discussions for the electric modes in $B$-theory, the zero energy modes are not lifted by higher derivative terms while the modes with energy of order one can received quantitative corrections of order one. Nevertheless, the qualitative scaling with $a$ remains universal.

# 8 $\mathbb{Z}_N$ Tensor Gauge Theory of $\hat{C}$

## 8.1 Plaquette Ising Model and Lattice Tensor Gauge Theory

We consider two lattice model that have the same continuum limit, the $\mathbb{Z}_N$ plaquette Ising model and the $\mathbb{Z}_N$ lattice tensor gauge theory of $\hat{C}$. In this sense, the two lattice models are dual to each other at long distance.

### 8.1.1 Plaquette Ising Model

The $\mathbb{Z}_N$ plaquette Ising model is the $\mathbb{Z}_N$ version of the XY-plaquette model. In $3 + 1$ dimensions, it has featured in the construction of the X-cube model [9]. See [10] for a review on this model.

There is a $\mathbb{Z}_N$ phase variable $U_s$ and its conjugate momentum $V_s$ at every site $s = (\hat{x}, \hat{y}, \hat{z})$. They obey the commutation relation $U_s V_s = e^{2\pi i/N} V_s U_s$. The Hamiltonian is

$$H = -K \sum_{\hat{x}, \hat{y}, \hat{z}} (L_{xy} + L_{yz} + L_{xz}) - h \sum_s V_s + \text{c.c.} ,$$

$$L_{xy} = U_{\hat{x}, \hat{y}, \hat{z}} U^{-1}_{\hat{x}+1, \hat{y}, \hat{z}} U^{-1}_{\hat{x}, \hat{y}+1, \hat{z}} U_{\hat{x}+1, \hat{y}+1, \hat{z}} .$$

$$(8.1)$$

We will assume $h$ to be small.

The symmetry operators are products of $V_s$ along any plane — for example, along the $xy$ plane,

$$\prod_{\hat{x}=1}^{L^x} \prod_{\hat{y}=1}^{L^y} V(\hat{x}, \hat{y}, \hat{z}_0) , \qquad (8.2)$$

and similarly along the $yz$ and the $xz$ planes. They become the $(\mathbf{1}, \mathbf{3}')$ dipole symmetry. There are $L^x + L^y + L^z - 2$ such operators on the lattice [4].

### 8.1.2 Lattice Tensor Gauge Theory of $\hat{C}$

The second lattice model is the $\mathbb{Z}_N$ lattice tensor gauge theory of $\hat{C}$. There is a $\mathbb{Z}_N$ phase variable $\hat{U}_c$ and its conjugate variable $\hat{V}_c$ on every cube $c$. They obey $\hat{U}_c \hat{V}_c = e^{2\pi i/N} \hat{V}_c \hat{U}_c$. For each link along the $k$ direction, there is a $\mathbb{Z}_N$ gauge parameter $\hat{\eta}^{ij}(\hat{x}, \hat{y}, \hat{z})$. Under the gauge transformation,

$$\hat{U}_c \sim \hat{U}_c \hat{\eta}^{xy}_{\hat{x}, \hat{y}, \hat{z}} (\hat{\eta}^{xy}_{\hat{x}+1, \hat{y}, \hat{z}})^{-1} (\hat{\eta}^{xy}_{\hat{x}, \hat{y}+1, \hat{z}})^{-1} \hat{\eta}^{xy}_{\hat{x}+1, \hat{y}+1, \hat{z}}$$
$$\hat{\eta}^{yz}_{\hat{x}, \hat{y}, \hat{z}} (\hat{\eta}^{yz}_{\hat{x}, \hat{y}+1, \hat{z}})^{-1} (\hat{\eta}^{yz}_{\hat{x}, \hat{y}, \hat{z}+1})^{-1} \hat{\eta}^{yz}_{\hat{x}, \hat{y}+1, \hat{z}+1} \qquad (8.3)$$
$$\hat{\eta}^{xz}_{\hat{x}, \hat{y}, \hat{z}} (\hat{\eta}^{xz}_{\hat{x}+1, \hat{y}, \hat{z}})^{-1} (\hat{\eta}^{xz}_{\hat{x}, \hat{y}, \hat{z}+1})^{-1} \hat{\eta}^{xz}_{\hat{x}+1, \hat{y}, \hat{z}+1}$$

where the product is over the 12 links around the cube.

Gauss law sets

$$\hat{G}_\ell \equiv \prod_{c \ni \ell} (\hat{V}_c)^{\epsilon_c} = 1 \tag{8.4}$$

where the product is an oriented product ($\epsilon_c = \pm 1$) over the four cubes $c$ that share a common link $\ell$. The Hamiltonian is

$$H = -\widetilde{h} \sum_c \hat{V}_c + \text{c.c.} \ , \tag{8.5}$$

with Gauss law imposed as an operator equation.

The symmetry operators are $\hat{V}_c$ at each cube $c$. Because of the Gauss laws, there are $L^x + L^y + L^z - 2$ such operators. They become the $\mathbb{Z}_N$ electric symmetry generators in the continuum.

Alternatively, we can impose Gauss law energetically by adding a term to the Hamiltonian

$$H = -K \sum_\ell \hat{G}_\ell - \widetilde{h} \sum_c \hat{V}_c + \text{c.c.} \ . \tag{8.6}$$

When $h$ in (8.1) and $\widetilde{h}$ in (8.6) vanish, the Hamiltonian (8.6) becomes the Hamiltonian (8.1) of the plaquette Ising model if we dualize the lattice and identify $\hat{U}_c \leftrightarrow V_s^{-1}$, $\hat{V}_c \leftrightarrow U_s$.

## 8.2   Continuum Lagrangian

We can obtain a continuum description of the $\mathbb{Z}_N$ theory by coupling the $U(1)$ $\hat{C}$-theory to a $\hat{A}$-theory with charge $N$ that Higgses it to $\mathbb{Z}_N$. The Euclidean Lagrangian is

$$\mathcal{L}_E = -\frac{i}{2(2\pi)} \check{B}_{ij} \left( (\partial_0 \hat{A}^{ij} - \partial_k \hat{A}_0^{k(ij)}) - N\hat{C}_0^{ij} \right) - \frac{i}{2\pi} \check{E} \left( \frac{1}{2} \partial_i \partial_j \hat{A}^{ij} - N\hat{C} \right) , \tag{8.7}$$

where $(\hat{C}_0^{(ij)}, \hat{C})$ are the $U(1)$ tensor gauge field in the $(\mathbf{3'}, \mathbf{1})$ representation of $S_4$, and $(\hat{A}_0^{k(ij)}, \hat{A}^{ij})$ are the $U(1)$ gauge field in the $(\mathbf{2}, \mathbf{3'})$ representation of $S_4$. These couplings Higgs the gauge symmetry of $(\hat{C}_0^{(ij)}, \hat{C})$ to $\mathbb{Z}_N$. The gauge transformations are

$$\begin{aligned}
\hat{A}_0^{i(jk)} &\sim \hat{A}_0^{i(jk)} + \partial_0 \hat{\beta}^{i(jk)} + N\hat{\alpha}_0^{i(jk)} \ , \\
\hat{A}^{ij} &\sim \hat{A}^{ij} + \partial_k \hat{\beta}^{k(ij)} + N\hat{\alpha}^{ij} \ , \\
\hat{C}_0^{ij} &\sim \hat{C}_0^{ij} + \partial_0 \hat{\alpha}^{ij} - \partial_k \hat{\alpha}_0^{k(ij)} \ , \\
\hat{C} &\sim \hat{C} + \frac{1}{2} \partial_i \partial_j \hat{\alpha}^{ij} \ ,
\end{aligned} \tag{8.8}$$

The equations of motion are

$$\partial_0 \hat{A}^{ij} - \partial_k \hat{A}_0^{k(ij)} - N\hat{C}_0^{ij} = 0 \ , \quad \frac{1}{2}\partial_i\partial_j \hat{A}^{ij} - N\hat{C} = 0 \ , \quad \check{B}_{ij} = 0 \ , \quad \check{E} = 0 \ . \tag{8.9}$$

We can dualize the Euclidean action by integrating out $(\hat{A}_0^{k(ij)}, \hat{A}^{ij})$. We rewrite the Lagrangian (8.7) as

$$\mathcal{L}_E = \frac{iN}{2\pi}\left(\frac{1}{2}\check{B}_{ij}\hat{C}_0^{ij} + \check{E}\hat{C}\right) + \frac{i}{2(2\pi)}\hat{A}^{ij}\left(\partial_0\check{B}_{ij} - \partial_i\partial_j\check{E}\right) - \frac{i}{6(2\pi)}\hat{A}_0^{k(ij)}(2\partial_k\check{B}_{ij} - \partial_i\check{B}_{jk} - \partial_j\check{B}_{ik}) \ , \tag{8.10}$$

where we have used $\hat{A}_0^{k(ij)}(\partial_k\check{B}_{ij} + \partial_i\check{B}_{jk} + \partial_j\check{B}_{ik}) = 0$. We now interpret the Higgs fields $(\hat{A}_0^{k(ij)}, \hat{A}^{ij})$ as Lagrangian multipliers implementing the constraints

$$2\partial_k\check{B}_{ij} - \partial_i\check{B}_{jk} - \partial_j\check{B}_{ik} = 0 \ , \qquad \partial_0\check{B}_{ij} = \partial_i\partial_j\check{E} \ , \tag{8.11}$$

where the first constraint can also be written as

$$\partial_i\check{B}_{jk} = \partial_j\check{B}_{ik} \ . \tag{8.12}$$

These constraints can be solved locally by a scalar field $\phi$ in **1**:

$$\check{E} = \partial_0\phi \ , \qquad \check{B}_{ij} = \partial_i\partial_j\phi \ . \tag{8.13}$$

The Euclidean Lagrangian (8.7) then becomes

$$\mathcal{L}_E = \frac{iN}{2\pi}\phi\left(\frac{1}{2}\partial_i\partial_j\hat{C}_0^{ij} - \partial_0\hat{C}\right) = -\frac{iN}{2\pi}\phi\hat{E} \ . \tag{8.14}$$

The equations of motion are

$$\hat{E} = 0 \ , \quad \partial_0\phi = 0 \ , \quad \partial_i\partial_j\phi = 0 \ . \tag{8.15}$$

To see that the value of $N$ is quantized, let us place the theory on a Euclidean 4-torus. Since the fluxes of $\hat{E}$ are quantized (see Section 7.3), invariance under $\phi \sim \phi + 2\pi$ leads to

$$N \in \mathbb{Z} \ . \tag{8.16}$$

## 8.3   Global Symmetries

Let us study the global symmetries of the $\mathbb{Z}_N$ tensor gauge theory. They are summarized in Figure 3. Since the $U(1)$ electric tensor symmetry of the $\hat{A}$ theory is gauged, it turns

the $U(1)$ magnetic dipole symmetry to $\mathbb{Z}_N$. Recall that this symmetry is dual to the $(\mathbf{1}, \mathbf{3}')$ momentum dipole symmetry of the $\phi$ theory [4]. In addition, coupling the *matter* field $\hat{A}$ to the pure gauge $\hat{C}$ theory breaks the $U(1)$ electric symmetry of the $\hat{C}$ theory to $\mathbb{Z}_N$.

The $\mathbb{Z}_N$ electric symmetry is generated by the local operators $e^{i\phi}$, and the $\mathbb{Z}_N$ $(\mathbf{1}, \mathbf{3}')$ dipole symmetry is generated by the slab operators $\hat{W}_i(x_1^i, x_2^i)$ in (7.31). They are both charged under each other, and satisfy the commutation relations

$$e^{i\phi(x,y,z)}\hat{W}_i(x_1^i, x_2^i) = e^{2\pi i/N}\hat{W}_i(x_1^i, x_2^i)e^{i\phi(x,y,z)} , \quad \text{if} \quad x_1^i < x^i < x_2^i , \tag{8.17}$$

and they commute otherwise.

Because of the second equation of motion of $\phi$ in (8.15), the electric symmetry operator factorizes into

$$e^{i\phi(x,y,z)} = e^{i\phi^x(x)+i\phi^y(y)+i\phi^z(z)} , \tag{8.18}$$

where $e^{i\phi^i(x^i)}$ have a gauge ambiguity,

$$e^{i\phi^x(x)} \sim \eta^x e^{i\phi^x(x)} , \quad e^{i\phi^y(y)} \sim \eta^y e^{i\phi^y(y)} , \quad e^{i\phi^z(z)} \sim (\eta^x \eta^y)^{-1} e^{i\phi^z(z)} . \tag{8.19}$$

where $\eta^x, \eta^y$ are arbitrary $\mathbb{Z}_N$ phases.

Depending on the global symmetry we impose in the microscopic model, the local operator $e^{i\phi}$ of the continuum theory may or may not be added to the Lagrangian to destabilize the theory. Let us demonstrate it in the two microscopic lattice models of Section 8.1.

In the $\mathbb{Z}_N$ plaquette lattice model discussed in Section 8.1.1, there is a microscopic $(\mathbf{1}, \mathbf{3}')$ dipole symmetry generated by (8.2). So its continuum limit is robust since there are no relevant local operators that are invariant under this symmetry.

On the other hand, the $(\mathbf{1}, \mathbf{3}')$ dipole symmetry is absent in the lattice tensor gauge theory discussed in Section 8.1.2. In this lattice gauge theory, only the electric symmetry, which is generated by $\hat{V}_c$, is manifest. So, we can deform the short-distance theory by adding local operators $e^{i\phi}$, which are charged under the $(\mathbf{1}, \mathbf{3}')$ symmetry. This will generically lift the ground state degeneracy discussed in Section 8.4, and break the $\mathbb{Z}_N$ $(\mathbf{1}, \mathbf{3}')$ tensor symmetry.

## 8.4 Ground State Degeneracy

In the presentation (8.7), all the fields can be solved in terms of the gauge fields $(\hat{A}_0^{k(ij)}, \hat{A}^{ij})$, and the solution space reduces to

$$\left\{ \hat{A}_0^{k(ij)}, \hat{A}^{ij} \mid \hat{A}_0^{i(jk)} \sim \hat{A}_0^{i(jk)} + \partial_0 \hat{\beta}^{i(jk)} + N\hat{\alpha}_0^{i(jk)} , \ \hat{A}^{ij} \sim \hat{A}^{ij} + \partial_k \hat{\beta}^{k(ij)} + N\hat{\alpha}^{ij} \right\} . \tag{8.20}$$

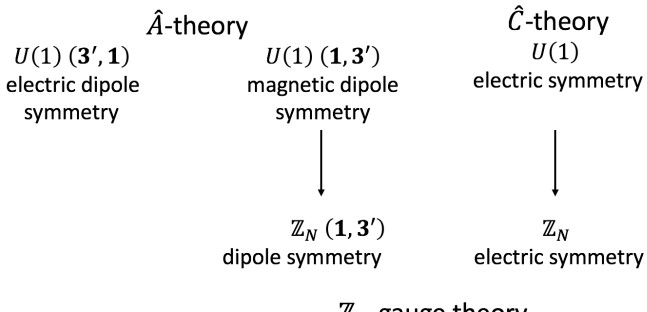

Figure 3: The global symmetries of the $U(1)$ $\hat{A}$-theory, the $U(1)$ $\hat{C}$-theory, the $\mathbb{Z}_N$ $\hat{C}$-theory, and their relations. The electric dipole symmetry of the $\hat{A}$-theory is gauged and therefore it is absent in the $\mathbb{Z}_N$ gauge theory. Note that the $U(1)$ $\hat{C}$-theory does not have a magnetic symmetry.

The only modes that survive after gauging are the magnetic modes of the $\hat{A}$ theory. If we regularize the theory on a lattice, these magnetic modes are labelled by $L^x + L^y + L^z - 2$ integers [4]. Large gauge transformations $(\hat{\alpha}_0^{i(jk)}, \hat{\alpha}^{ij})$ identify these integers modulo $N$. As a result, there are $N^{L^x+L^y+L^z-2}$ nontrivial configurations leading to this ground state degeneracy.

There are other ways to see this. One can start with the $BF$-type presentation (8.14), find the solution space of the equations of motion (8.15) in the temporal gauge $\hat{C}_0^{ij} = 0$, and then quantize these modes on a lattice. Yet another way to see this is by studying the symmetry operators on the lattice:

$$e^{i\phi(\hat{x},\hat{y},\hat{z})} , \quad \hat{W}_i(\hat{x}^i) . \tag{8.21}$$

The former factorizes into

$$e^{i\phi(\hat{x},\hat{y},\hat{z})} = e^{i\phi^x(\hat{x})+i\phi^y(\hat{y})+i\phi^z(\hat{z})} , \quad \forall \; \hat{x}, \hat{y}, \hat{z} , \tag{8.22}$$

because of the equation of motion (8.15) of $\phi$. Here $e^{i\phi^i(\hat{x}^i)}$ have a gauge ambiguity,

$$e^{i\phi^x(\hat{x})} \sim \eta^x e^{i\phi^x(\hat{x})} , \quad e^{i\phi^y(\hat{y})} \sim \eta^y e^{i\phi^y(\hat{y})} , \quad e^{i\phi^z(\hat{z})} \sim (\eta^x\eta^y)^{-1} e^{i\phi^z(\hat{z})} . \tag{8.23}$$

with $\eta^x, \eta^y$ arbitrary $\mathbb{Z}_N$ phases. The latter satisfy two constraints

$$\prod_{\hat{x}=1}^{L^x} \hat{W}_x(\hat{x}) = \prod_{\hat{y}=1}^{L^y} \hat{W}_y(\hat{y}) = \prod_{\hat{z}=1}^{L^z} \hat{W}_z(\hat{z}) . \tag{8.24}$$

Using the gauge ambiguity (8.23), we can fix $e^{i\phi^z(\hat{z}=1)} = 1$ and $e^{i\phi^y(\hat{y}=1)} = 1$, and using the two constraints (8.24), we can solve for $\hat{W}_z(\hat{z} = 1)$ and $\hat{W}_y(\hat{y} = 1)$ in terms of other $\hat{W}_i(\hat{x}^i)$. Therefore, there are $L^x + L^y + L^z - 2$ operators of each kind.

The set of commutation relations (8.17) of these operators is isomorphic to $L^x + L^y + L^z - 2$ copies of Heisenberg algebra, $AB = e^{2\pi i/N}BA$ and $A^N = B^N = 1$. The isomorphism is given by

$$
\begin{aligned}
A_{\hat{x}} &= e^{i\phi(\hat{x},1,1)} \ , & B_{\hat{x}} &= \hat{W}_x(\hat{x}) \ , & \hat{x} &= 1,\ldots,L^x \ , \\
A_{\hat{y}} &= e^{i\phi(1,\hat{y},1)-i\phi(1,1,1)} \ , & B_{\hat{y}} &= \hat{W}_y(\hat{y}) \ , & \hat{y} &= 2,\ldots,L^y \ , \\
A_{\hat{z}} &= e^{i\phi(1,1,\hat{z})-i\phi(1,1,1)} \ , & B_{\hat{z}} &= \hat{W}_z(\hat{z}) \ , & \hat{z} &= 2,\ldots,L^z \ ,
\end{aligned}
\tag{8.25}
$$

These commutation relations force the ground state degeneracy to be $N^{L^x+L^y+L^z-2}$.

# Acknowledgements

We thank M. Hermele for helpful discussions. PG was supported by Physics Department of Princeton University. HTL was supported by a Croucher Scholarship for Doctoral Study, a Centennial Fellowship from Princeton University and Physics Department of Princeton University. The work of NS was supported in part by DOE grant DE$-$SC0009988. NS and SHS were also supported by the Simons Collaboration on Ultra-Quantum Matter, which is a grant from the Simons Foundation (651440, NS). SHS thanks the Department of Physics at National Taiwan University for its hospitality while this work was being completed. Opinions and conclusions expressed here are those of the authors and do not necessarily reflect the views of funding agencies.

# A    Representation Theory of $S_4$

The symmetry group of the cubic lattice (modulo translations) is the *cubic group*, which consists of 48 elements. We will focus on the group of orientation-preserving symmetries of the cube, which is isomorphic to the permutation group of four objects $S_4$.

The irreducible representations of $S_4$ are the trivial representation $\mathbf{1}$, the sign representation $\mathbf{1'}$, a two-dimensional irreducible representation $\mathbf{2}$, the standard representation $\mathbf{3}$, and another three-dimensional irreducible representation $\mathbf{3'}$. The representation $\mathbf{3'}$ is the tensor product of the sign representation and the standard representation, $\mathbf{3'} = \mathbf{1'} \otimes \mathbf{3}$.

It is convenient to embed $S_4 \subset SO(3)$ and decompose the $SO(3)$ irreducible represen-

tations in terms of $S_4$ representations. The first few are

$$
\begin{aligned}
SO(3) &\supset S_4 \\
\mathbf{1} &= \mathbf{1} \\
\mathbf{3} &= \mathbf{3} \\
\mathbf{5} &= \mathbf{2} \oplus \mathbf{3}' \\
\mathbf{7} &= \mathbf{1}' \oplus \mathbf{3} \oplus \mathbf{3}' \\
\mathbf{9} &= \mathbf{1} \oplus \mathbf{2} \oplus \mathbf{3} \oplus \mathbf{3}'
\end{aligned}
\tag{A.1}
$$

We will label the components of $S_4$ representations using $SO(3)$ vector indices as follows. The three-dimensional standard representation $\mathbf{3}$ of $S_4$ carries an $SO(3)$ vector index $i$, or equivalently, an antisymmetric pair of indices $[jk]$.[14] Similarly, the irreducible representations of $S_4$ can be expressed in terms of the following tensors:

$$
\begin{aligned}
\mathbf{1} \quad &: \quad S \\
\mathbf{1}' \quad &: \quad T_{(ijk)} \quad , \quad i \neq j \neq k \\
\mathbf{2} \quad &: \quad B_{[ij]k} \quad , \quad i \neq j \neq k \quad , \quad B_{[ij]k} + B_{[jk]i} + B_{[ki]j} = 0 \\
&\quad\quad B_{i(jk)} \quad , \quad i \neq j \neq k \quad , \quad B_{i(jk)} + B_{j(ki)} + B_{k(ij)} = 0 \\
\mathbf{3} \quad &: \quad V_i \\
\mathbf{3}' \quad &: \quad E_{ij} \quad\quad , \quad i \neq j \quad\quad\quad , \quad E_{ij} = E_{ji}
\end{aligned}
\tag{A.2}
$$

In the above we have two different expressions, $B_{[ij]k}$ and $B_{i(jk)}$, for the irreducible representation $\mathbf{2}$ of $S_4$. In the first expression, $B_{[ij]k}$ is the component of $\mathbf{2}$ in the tensor product $\mathbf{3} \otimes \mathbf{3} = \mathbf{1} \oplus \mathbf{2} \oplus \mathbf{3} \oplus \mathbf{3}'$. In the second expression, $B_{i(jk)}$ is the component of $\mathbf{2}$ in the tensor product $\mathbf{3} \otimes \mathbf{3}' = \mathbf{1}' \oplus \mathbf{2} \oplus \mathbf{3} \oplus \mathbf{3}'$.

In most of this paper, the indices $i, j, k$ in every expression are not equal, $i \neq j \neq k$ (see (A.2) for example). Equivalently, components of a tensor with repeated indices are set to be zero, e.g. $E_{ii} = 0$ and $B_{ijj} = 0$ (no sum). Repeated indices in an expression are summed over unless otherwise stated. For example, $E_{ij}E^{ij} = 2E_{xy}^2 + 2E_{yz}^2 + 2E_{xz}^2$. As in this expression, we will often use $x, y, z$ both as coordinates and as the indices of a tensor.

We generally do not distinguish the upper and lower indices with only one exception for the $\mathbf{2}$ of $S_4$. The upper indexed tensors $B^{[ij]k}$ and $B^{i(jk)}$ are related as

$$
B^{i(jk)} = B^{[ij]k} + B^{[ik]j} \ , \qquad B^{[ij]k} = \frac{1}{3}\left(B^{i(jk)} - B^{j(ik)}\right) \ ,
\tag{A.3}
$$

---

[14]We will adopt the convention that indices in the square brackets are antisymmetrized, whereas indices in the parentheses are symmetrized. For example, $A_{[ij]} = -A_{[ji]}$ and $A_{(ij)} = A_{(ji)}$.

whereas the lower indexed tensors $B_{[ij]k}$ and $B_{i(jk)}$ are related as[15]

$$B_{i(jk)} = \frac{1}{3}\left(B_{[ij]k} + B_{[ik]j}\right) , \qquad B_{[ij]k} = B_{i(jk)} - B_{j(ik)} , \qquad (\text{A.4})$$

The upper and lower indexed tensors are related as

$$B^{i(jk)} = 3B_{i(jk)} , \qquad B^{[ij]k} = B_{[ij]k} . \qquad (\text{A.5})$$

Because of this convention, we can freely raise and lower the $[ij]k$ indices, but not the $i(jk)$ indices. Finally, any two tensors $A$ and $B$ in the **2** of $S_4$ can be contracted in the following ways:

$$
\begin{aligned}
A_{[ij]k}B_{[ij]k} &= A^{[ij]k}B^{[ij]k} = A^{[ij]k}B_{[ij]k} \\
&= A^{i(jk)}B_{i(jk)} = \frac{1}{3}A^{i(jk)}B^{i(jk)} = 3A_{i(jk)}B_{i(jk)} .
\end{aligned}
\qquad (\text{A.6})
$$

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
