# Peer review of "More Exotic Field Theories in 3+1 Dimensions"

_SciPost Physics_

## Round 2 · Referee Report · Anonymous (Referee 1) · 2020-9-21

Report
This paper is a follow up to several previous papers by the same set of authors on the exploration of quantum field theory description of fracton models. The authors made significant progress on this topic by making use of nonstandard quantum fields where discontinuous configurations play an important role. This paper, following the framework established in previous papers, discussed several more examples. It is carefully written and provides nice pedagogical material for anyone who wants to learn the method in detail. I support its publication in Scipost although I think it would be helpful if the authors can explain more about the general logic behind their writing. Right now, the paper is presented as "some more examples". What is the importance of these examples? The authors did comment on the connection of some of them to those discussed in previous papers, but it is not clear why they find these models interesting. In general, I find the series of papers very carefully written and the cases explored in detail, but I would appreciate if the authors can comment more broadly on where they think this effort can lead us. My understanding from reading the series is that first we learn the surprising fact that it is possible to describe the exotic fracton physics using field theory, secondly the key to this description is to consider discontinuous configurations of the field (but still more continuous than typical lattice configurations). What are other important take home messages? How does this help with our understanding of the fracton models? On the other hand, what do we learn about field theory from this process? Addressing these questions is not essential to the publication of this paper, but I would find it very helpful.

---

## Editorial Decision

resubmitted